# Natural-gradient learning for spiking neurons

**Elena Kreutzer[1]\*, Walter Senn[1‡], Mihai A Petrovici[1,2]\*‡**

[1]Department of Physiology, University of Bern, Bern, Switzerland; [2]Kirchhoff-Institute for Physics, Heidelberg University, Heidelberg, Germany

**Abstract** In many normative theories of synaptic plasticity, weight updates implicitly depend on the chosen parametrization of the weights. This problem relates, for example, to neuronal morphology: synapses which are functionally equivalent in terms of their impact on somatic firing can differ substantially in spine size due to their different positions along the dendritic tree. Classical theories based on Euclidean-gradient descent can easily lead to inconsistencies due to such parametrization dependence. The issues are solved in the framework of Riemannian geometry, in which we propose that plasticity instead follows natural-gradient descent. Under this hypothesis, we derive a synaptic learning rule for spiking neurons that couples functional efficiency with the explanation of several well-documented biological phenomena such as dendritic democracy, multiplicative scaling, and heterosynaptic plasticity. We therefore suggest that in its search for functional synaptic plasticity, evolution might have come up with its own version of natural-gradient descent.

### Editor's evaluation

The natural gradient has a long and rich history in machine learning. Here, the authors derive a biologically plausible implementation of natural-gradient-based plasticity for spiking neurons, which renders learning invariant under dendritic transformations. This new synaptic learning rule makes several very interesting experimental predictions with respect to the interplay of homo- and heterosynaptic plasticity, and with regard to the scaling of plasticity by the presynaptic variance.

**\*For correspondence:**
kreutzer@pyl.unibe.ch (EK);
mihai.petrovici@unibe.ch (MAP)

‡Joint senior authorship

**Competing interest:** The authors declare that no competing interests exist.

## Introduction

Understanding the fundamental computational principles underlying synaptic plasticity represents a long-standing goal in neuroscience. To this end, a multitude of top-down computational paradigms have been developed, which derive plasticity rules as gradient descent on a particular objective function of the studied neural network (*Rosenblatt, 1958*; *Rumelhart et al., 1986*; *Pfister et al., 2006*; *D'Souza et al., 2010*; *Friedrich et al., 2011*).

However, the exact physical quantity to which these synaptic weights correspond often remains unspecified. What is frequently simply referred to as $w_{ij}$ (the synaptic weight from neuron $j$ to neuron $i$) might relate to different components of synaptic interaction, such as calcium concentration in the presynaptic axon terminal, neurotransmitter concentration in the synaptic cleft, receptor activation in the postsynaptic dendrite or the postsynaptic potential (PSP) amplitude in the spine, the dendritic shaft or at the soma of the postsynaptic cell. All these biological processes can be linked by transformation rules, but depending on which of them represents the variable with respect to which performance is optimized, the network behavior during training can be markedly different.

As an example we consider the parametrization of the synaptic strength either as PSP amplitude in the soma, $w^s$, or as PSP amplitude in the dendrite, $w^d$ (see also *Figure 1* and Sec. 'The naive Euclidean gradient is not parametrization-invariant'). Reparametrizing the synaptic strength in this way implies

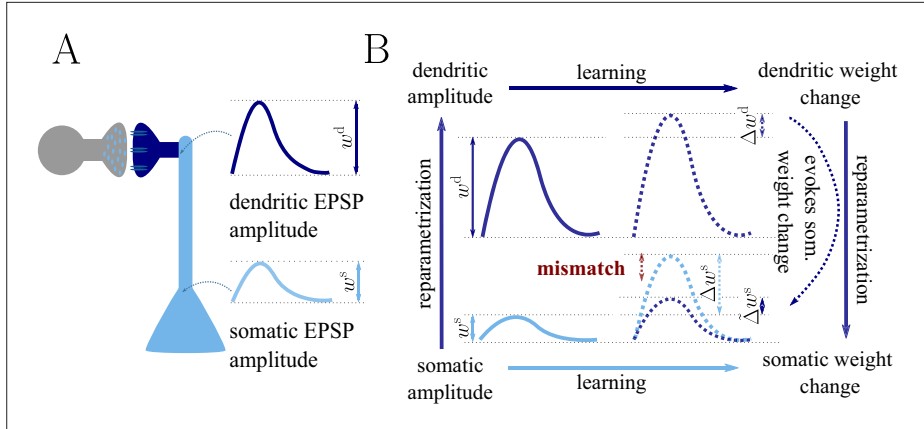

**Figure 1.** Classical gradient descent depends on chosen parametrization. (**A**) The strength of a synapse can be parametrized in various ways, for example, as the EPSP amplitude at either the soma $w^s$ or the dendrite $w^d$. Biological processes such as attenuation govern the relationship between these variables. Depending on the chosen parametrization, Euclidean-gradient descent can yield different results. (**B**) Phenomenological correlates. EPSPs before learning are represented as continuous, after learning as dashed curves. The light blue arrow represents gradient descent on the error as a function of the somatic EPSP $C^s\left[w^s\right]$ (also shown in light blue). The resulting weight change leads to an increase $\Delta w^s$ in the somatic EPSP after learning. The dark blue arrows track the calculation of the same gradient, but with respect to the dendritic EPSP (also shown in dark blue): (1) taking the attenuation into account in order to compute the error as a function of $w^d$, (2) calculating the gradient, followed by (3) deriving the associated change in $\tilde{\Delta}w^s$, again considering attenuation. Due to the attenuation $f(w)$ entering the calculation twice, the synaptic weights updates, as well as the associated evolution of a neuron's output statistics over time, will differ under the two parametrizations.

an attenuation factor for each single synapse, but different factors are assigned across the positions on the dendritic tree. As a consequence, the weight vector will follow a different trajectory during learning depending on whether the somatic or dendritic parametrization of the PSP amplitude was chosen.

It certainly could be the case that evolution has favored a parametrization-dependent learning rule, along with one particular parametrization over all others, but this would necessarily imply sub-optimal convergence for all but a narrow set of neuron morphologies and connectome configurations. An invariant learning rule on the other hand would not only be mathematically unambiguous and therefore more elegant, but could also improve learning, thus increasing fitness.

In some aspects, the question of invariant behavior is related to the principle of relativity in physics, which requires the laws of physics – in our case: the improvement of performance during learning – to be the same in all frames of reference. What if neurons would seek to conserve the way they adapt their behavior regardless of, for example, the specific positioning of synapses along their dendritic tree? Which equations of motion – in our case: synaptic learning rules – are able to fulfill this requirement?

The solution lies in following the path of steepest descent not in relation to a small change in the synaptic weights (Euclidean-gradient descent), but rather with respect to a small change in the input-output distribution (natural-gradient descent). This requires taking the gradient of the error function with respect to a metric defined directly on the space of possible input-output distributions, with coordinates defined by the synaptic weights. First proposed in *Amari, 1998*, but with earlier roots in information geometry (*Amari, 1987*; *Amari and Nagaoka, 2000*), natural-gradient methods (*Yang and Amari, 1998*; *Rattray and Saad, 1999*; *Park et al., 2000*; *Kakade, 2001*) have recently been rediscovered in the context of deep learning (*Pascanu and Bengio, 2013*; *Martens, 2014*; *Ollivier, 2015*; *Amari et al., 2019*; *Bernacchia et al., 2018*). Moreover, *Pascanu and Bengio, 2013* showed that the natural-gradient learning rule is closely related to other machine learning algorithms. However, most of the applications focus on rate-based networks which are not inherently linked to a statistical manifold and have to be equipped with Gaussian noise or a probabilistic output layer interpretation in order to allow an application of the natural gradient. Furthermore, a biologically plausible synaptic plasticity rule needs to make all the required information accessible at the synapse itself, which is usually unnecessary and therefore largely ignored in machine learning.

The stochastic nature of neuronal outputs in-vivo (see, e.g. *Softky and Koch, 1993*) provides a natural setting for plasticity rules based on information geometry. As a model for biological synapses, natural gradient combines the elegance of invariance with the success of gradient-descent-based learning rules. In this manuscript, we derive a closed-form synaptic learning rule based on natural-gradient descent for spiking neurons and explore its implications. Our learning rule equips the synapses with more functionality compared to classical error learning by enabling them to adjust their learning rate to their respective impact on the neuron's output. It naturally takes into account relevant variables such as the statistics of the afferent input or their respective positions on the dendritic tree. This allows a set of predictions which are corroborated by both experimentally observed phenomena such as dendritic democracy and multiplicative weight dynamics and theoretically desirable properties such as Bayesian reasoning (*Marceau-Caron and Ollivier, 2007*). Furthermore, and unlike classical error-learning rules, plasticity based on the natural gradient is able to incorporate both homo- and heterosynaptic phenomena into a unified framework. While theoretically derived heterosynaptic components of learning rules are notoriously difficult for synapses to implement due to their non-locality, we show that in our learning rule they can be approximated by quantities accessible at the locus of plasticity. In line with results from machine learning, the combination of these features also enables faster convergence during supervised learning.

## Results

### The naive Euclidean gradient is not parametrization-invariant

We consider a cost function $C$ on the neuronal level that, in the sense of cortical credit assignment (see e.g. *Sacramento et al., 2017*), can relate to some behavioral cost of the agent that it serves. The output of the neuron depends on the amplitudes of the somatic PSPs elicited by the presynaptic spikes. We denote these 'somatic weights' by $w^{\mathrm{s}}$, and may parametrize the neuronal cost as $C = C^{\mathrm{s}}[w^{\mathrm{s}}]$.

However, dendritic PSP amplitudes $w^{\mathrm{d}}$ can be argued to offer a more unmitigated representation of synaptic weights, so we might rather wish to express the cost as $C = C^{\mathrm{d}}[w^{\mathrm{d}}]$. These two parametrizations are related by an attenuation factor $\alpha$ (between 0 and 1):

$$w^{\mathrm{s}} = \alpha w^{\mathrm{d}}. \tag{1}$$

In general, this attenuation factor depends on the synaptic position and is therefore described by a vector that is multiplied component-wise with the weights. For clarity, we assume a simplified picture in which we neglect, for example, the spread of PSPs as they travel along dendritic cables. However, our observations regarding the effects of reparametrization hold in general.

It may now seem straightforward to switch between the somatic and dendritic representation of the cost by simply substituting variables, for example

$$C^{\mathrm{d}}[w^{\mathrm{d}}] = C^{\mathrm{s}}[w^{\mathrm{s}}] = C^{\mathrm{s}}[\alpha w^{\mathrm{d}}]. \tag{2}$$

To derive a plasticity rule for the somatic and dendritic weights, we might consider gradient descent on the cost:

$$\Delta w^{\mathrm{d}} = -\frac{\partial C^{\mathrm{d}}}{\partial w^{\mathrm{d}}} = -\frac{\partial w^{\mathrm{s}}}{\partial w^{\mathrm{d}}} \frac{\partial C^{\mathrm{s}}}{\partial w^{\mathrm{s}}} = -\alpha \frac{\partial C^{\mathrm{s}}}{\partial w^{\mathrm{s}}} = \alpha \Delta w^{\mathrm{s}} . \tag{3}$$

At first glance, this relation seems reasonable: dendritic weight changes affect the cost more weakly then somatic weight changes, so their respective gradient is more shallow by the factor $\alpha$. However, from a functional perspective, the opposite should be true: dendritic weights should experience a larger change than somatic weights in order to elicit the same effect on the cost. This conflict can be made explicit by considering that somatic weight changes are, themselves, attenuated dendritic weight changes: $\Delta w^{\mathrm{s}} = \alpha \Delta w^{\mathrm{d}}$. Substituting this into *Equation 3* leads to an inconsistency: $\Delta w^{\mathrm{d}} = \alpha^2 \Delta w^{\mathrm{d}}$. To solve this conundrum, we need to shift the focus from changing the synaptic input to changing the neuronal output, while at the same time considering a more rigorous treatment of gradient descent (see also *Surace et al., 2020*).

## The natural-gradient plasticity rule

We consider a neuron with somatic potential $V$ (above a baseline potential $V_{\text{rest}}$) evoked by the spikes $x_i$ of $n$ presynaptic afferents firing at rates $r_i$. The presynaptic spikes of afferent $i$ cause a train of weighted dendritic potentials $w_i^{\text{d}} x_i^{\epsilon}$ locally at the synaptic site. The $x_i$ denotes the unweighted synaptic potential (USP) train elicited by the low-pass-filtered spike train $x_i$. At the soma, each dendritic potential is attenuated by a potentially nonlinear function that depends on the synaptic location:

$$w_i^{\text{s}} = f_i(w_i^{\text{d}}). \tag{4}$$

The somatic voltage above baseline thus reads as

$$V = \sum_{i=1}^{n} w_i^{\text{s}} x_i^{\epsilon} = \sum_{i=1}^{n} f_i(w_i^{\text{d}}) x_i^{\epsilon} . \tag{5}$$

We further assume that the neuron's firing follows an inhomogeneous Poisson process whose rate

$$\phi_t(V) := \phi(V_t) \tag{6}$$

depends on the current membrane potential through a nonlinear transfer function $\phi$. In this case, spiking in a sufficiently short interval $[t, t + \mathrm{d}t]$ is Bernoulli-distributed. The probability of a spike occurring in this interval (denoted as $y_t = 1$) is then given by

$$p_{\boldsymbol{w}} \left( y_t = 1 \big| \boldsymbol{x}_t^{\epsilon} \right) = \phi_t(V) \, \mathrm{d}t \, , \tag{7}$$

which defines our generalized linear neuron model (**Gerstner and Kistler, 2002**). Here, we used $\boldsymbol{x}^{\epsilon}$ as a shorthand notation for the USP vector $(x_i^{\epsilon})$. The full input-output distribution then reads

$$p_{\boldsymbol{w}} \left( y_t, \boldsymbol{x}_t^{\epsilon} \right) = \left( \phi_t \mathrm{d}t \right)^{y_t} \left( 1 - \phi_t \mathrm{d}t \right)^{(1-y_t)} p_{\text{usp}} \left( \boldsymbol{x}_t^{\epsilon} \right) \, , \tag{8}$$

where the probability density of the input $\boldsymbol{x}_t^{\epsilon}$ is independent of synaptic weights (see also Sec. 'Detailed derivation of the natural-gradient learning rule'). In the following, we drop the time indices for better readability.

In view of this stochastic nature of neuronal responses, we consider a neuron that strives to reproduce a target firing distribution $p^*\left(y|\boldsymbol{x}^{\epsilon}\right)$. The required information is received as teacher spike train $Y^*$ that is sampled from $p^*$ (see Sec. 'Sketch for the derivation of the somatic natural-gradient learning rule' for details). In this context, plasticity may follow a supervised-learning paradigm based on gradient descent. The Kullback-Leibler divergence between the neuron's current and its target firing distribution

$$C \left[ p_{\boldsymbol{w}} \right] = D_{\text{KL}}(p^* \| p_{\boldsymbol{w}}) = \mathbb{E} \left[ \log \left( \frac{p^*}{p_{\boldsymbol{w}}} \right) \right]_{p^*} \tag{9}$$

represents a natural cost function which measures the error between the current and the desired output distribution in an information-theoretic sense. Minimizing this cost function is equivalent to maximizing the log-likelihood of teacher spikes, since $p^*$ does not depend on $\boldsymbol{w}$. Using naive Euclidean-gradient descent with respect to the synaptic weights (denoted by $\nabla_{\boldsymbol{w}}^{\text{E}}$) results in the well-known error-correcting rule (**Pfister et al., 2006**),

$$\dot{\boldsymbol{w}}^{\text{s}} = -\eta \nabla_{\boldsymbol{w}}^{E} C = \eta \left[ Y^* - \phi(V) \right] \frac{\phi'(V)}{\phi(V)} \boldsymbol{x}^{\epsilon}, \tag{10}$$

which is a spike-based version of the classical perceptron learning rule (**Rosenblatt, 1958**), whose multilayer version forms the basis of the error-backpropagation algorithm (**Rumelhart et al., 1986**). On the single-neuron level, a possible biological implementation has been suggested by **Urbanczik and Senn, 2014**, who demonstrated how a neuron may exploit its morphology to store errors, an idea that was recently extended to multilayer networks (**Sacramento et al., 2017**; **Haider et al., 2021**).

However, as we argued above, learning based on Euclidean-gradient descent is not unproblematic. It cannot account for synaptic weight (re)parametrization, as caused, for example, by the diversity of synaptic loci on the dendritic tree. *Equation 10* represents the learning rule for a somatic parametrization of synaptic weights. For a local computation of Euclidean gradients using a dendritic parametrization, weight updates decrease with increasing distance towards the soma (*Equation 3*), which

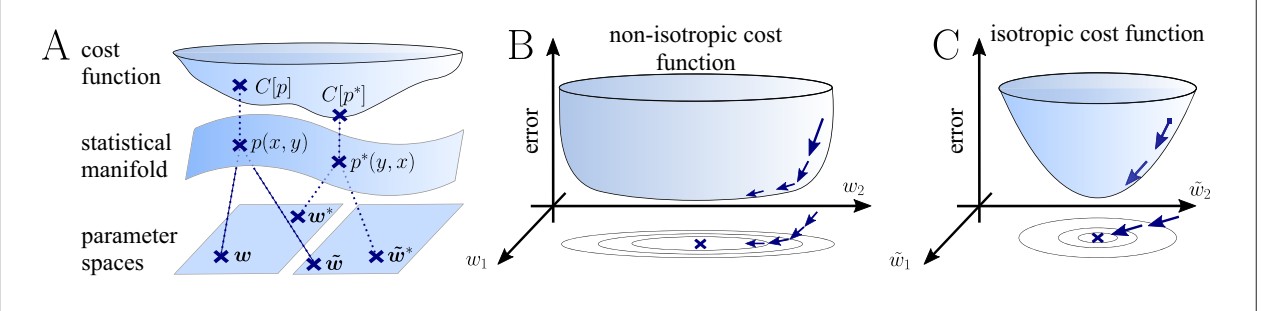

**Figure 2.** The natural gradient represents the true gradient direction on the manifold of neuronal input-output distributions. (**A**) During supervised learning, the error between the current and the target state is measured in terms of a cost function defined on the neuron's output space; in our case, this is the manifold formed by the neuronal output distributions $p(y, x)$. As the output of a neuron is determined by the strength of incoming synapses, the cost $C$ depends indirectly on the afferent weight vector $w$. Since the gradient of a function depends on the distance measure of the underlying space, Euclidean-gradient descent, which follows the gradient of the cost as a function of the synaptic weights $\partial C/\partial w$, is not uniquely defined, but depends on how $w$ is parametrized. If, instead, we follow the gradient on the output manifold itself, it becomes independent of the underlying parametrization. Expressed in a specific parametrization, the resulting natural gradient contains a correction term that accounts for the distance distortion between the synaptic parameter space and the output manifold. (**B–C**) Standard gradient descent learning is suited for isotropic (**C**), rather than for non-isotropic (**B**) cost functions. For example, the magnitude of the gradient decreases in valley regions where the cost function is flat, resulting in slow convergence to the target. A non-optimal choice of parametrization can introduce such artefacts and therefore harm the performance of learning rules based on Euclidean-gradient descent. In contrast, natural-gradient learning will locally correct for distortions arising from non-optimal parametrizations (see also *Figure 3*).

is likely to harm the convergence speed toward an optimal weight configuration. With the multiplicative USP term $x^\epsilon$ in *Equation 10* being the only manifestation of presynaptic activity, there is no mechanism by which to take into account input variability. This can, in turn, also impede learning, by implicitly assigning equal importance to reliable and unreliable inputs. Furthermore, when compared to experimental evidence, this learning rule cannot explain heterosynaptic plasticity, as it is purely presynaptically gated.

In general, Euclidean-gradient descent is well-known to exhibit slow convergence in non-isotropic regions of the cost function (*Ruder, 2016*), with such non-isotropy frequently arising or being aggravated by an inadequate choice of parametrization (see *Ollivier, 2015* and *Figure 2*). In contrast, natural-gradient descent is, by construction, immune to these problems. The key idea of natural gradient as outlined by Amari is to follow the (locally) shortest path in terms of the neuron's firing distribution. Argued from a normative point of view, this is the only 'correct' path to consider, since plasticity aims to adapt a neuron's behavior, that is, its input-output relationship, rather than some internal parameter (*Figure 2*). In the following, we therefore drop the index from the synaptic weights $w$ to emphasize the parametrization-invariant nature of the natural gradient.

For the concept of a locally shortest path to make sense in terms of distributions, we require the choice of a distance measure for probability distributions. Since a parametric statistical model, such as the set of our neuron's realizable output distributions, forms a Riemannian manifold, a local distance measure can be obtained in form of a Riemannian metric. The Fisher metric (*Rao, 1945*), an infinitesimal version of the $D_{\mathrm{KL}}$, represents a canonical choice on manifolds of probability distributions (*Amari and Nagaoka, 2000*). On a given parameter space, the Fisher metric may be expressed in terms of a bilinear product with the Fisher information matrix

$$G(w) = \mathbb{E}\left[\frac{\partial \log p_w}{\partial w} \frac{\partial \log p_w}{\partial w}^T\right]_{p_w}. \tag{11}$$

The Fisher metric locally measures distances in the $p$-manifold as a function of the chosen parametrization. We can then obtain the natural gradient (which intuitively may be thought of as '$\partial C/\partial p$') by correcting the Euclidean gradient $\nabla_w^{\mathrm{E}} C := \partial C/\partial w$ with the distance measure above:

$$\nabla_w^{\mathrm{N}} C = G\left(w\right)^{-1} \nabla_w^{\mathrm{E}} C. \tag{12}$$

This correction guarantees invariance of the gradient under reparametrization (see also Sec. 'Reparametrization and the general natural-gradient rule'). The natural-gradient learning rule is then given as $\dot{\boldsymbol{w}} = -\eta \nabla^{\mathrm{N}}_{\boldsymbol{w}} C$. Calculating the right-hand expression for the case of Poisson-spiking neurons (for details, see Detailed derivation of the natural-gradient learning rule and Inverse of the Fisher Information Matrix), this takes the form

$$\dot{\boldsymbol{w}} = \eta \, \gamma_{\mathrm{s}} \left[ Y^* - \phi(V) \right] \frac{\phi'(V)}{\phi(V)} \frac{1}{f'(\boldsymbol{w})} \left[ c_\epsilon \frac{\boldsymbol{x}^\epsilon}{\boldsymbol{r}} - \gamma_{\mathrm{u}} \mathbf{1} + \gamma_{\mathrm{w}} \boldsymbol{f}(\boldsymbol{w}) \right] \;, \tag{13}$$

where $\boldsymbol{w}$ is an arbitrary weight parametrization that relates to the somatic amplitudes via a component-wise rescaling

$$\boldsymbol{w}^{\mathrm{s}} = \boldsymbol{f}(\boldsymbol{w}) = \left[ f_i(w_i) \right]^n_{1=1} . \tag{14}$$

Note that the attenuation function in *Equation 4* represents a special case of *Equation 14* for dendritic amplitudes.

We used the shorthand $\gamma_{\mathrm{s}}$, $\gamma_{\mathrm{u}}$, and $\gamma_{\mathrm{w}}$ for three scaling factors introduced by the natural gradient that we address in detail below. For easier reading, we use a shorthand notation in which multiplications, divisions and scalar functions of vectors apply component-wise.

*Equation 13* represents the complete expression of our natural-gradient rule, which we discuss throughout the remainder of the manuscript. Note that, while having used a standard sigmoidal transfer function throughout the paper, *Equation 13* holds for every sufficiently smooth $\phi$.

Natural-gradient learning conserves both the error term $\left[ Y^* - \phi(V) \right]$ and the USP contribution $\boldsymbol{x}^\epsilon$ from classical gradient-descent plasticity. However, by including the relationship between the parametrization of interest $\boldsymbol{w}$ and the somatic PSP amplitudes $\boldsymbol{f}(\boldsymbol{w})$, natural-gradient-based plasticity explicitly accounts for reparametrization distortions, such as those arising from PSP attenuation during propagation along the dendritic tree. Furthermore, natural-gradient learning introduces multiple scaling factors and new plasticity components, whose characteristics will be further explored in dedicated sections below (see also Sec. 'Global scaling factor' and 'Empirical Analysis of $\gamma_{\mathrm{u}}$ and $\gamma_{\mathrm{w}}$' for more details).

First of all, we note the appearance of two scaling factors (more details in 'Input and output-specific scaling'). On one hand, the size of the synaptic adjustment is modulated by a global scaling factor $\gamma_{\mathrm{s}}$, which adjusts synaptic weight updates to the characteristics of the output non-linearity, similarly to the synapse-specific scaling by the inverse of $\boldsymbol{f}'$. Furthermore $\gamma_{\mathrm{s}}$ also depends on the output statistics of the neuron, harmonizing plasticity across different states in the output distribution (see Sec. 'Global scaling factor'). On the other hand, a second, synapse-specific learning rate scaling accounts for the statistics of the input at the respective synapse, in the form of a normalization by the afferent input rate $c_\epsilon / \boldsymbol{r}$, where $c_\epsilon$ is a constant that depends on the PSP kernel (see Sec. 'Neuron model'). Unlike the global modulation introduced by $\gamma_{\mathrm{s}}$, this scaling only affects the USP-dependent plasticity component. Just as for Euclidean-gradient-based learning, the latter is directly evoked by the spike trains arriving at the synapse. Therefore, the resulting plasticity is homosynaptic, affecting only synapses which receive afferent input.

However, in the case of natural-gradient learning, this input-specific adaptation is complemented by two additional forms of heterosynaptic plasticity (Sec. 'Interplay of homosynaptic and heterosynaptic plasticity'). First, the learning rule has a bias term $\gamma_{\mathrm{u}}$ which uniformly adjusts all synapses and may be considered homeostatic, as it usually opposes the USP-dependent plasticity contribution. The amplitude of this bias does not exclusively depend on the afferent input at the respective synapse, but is rather determined by the overall input to the neuron. Thus, unlike the USP-dependent component, this heterosynaptic plasticity component equally affects both active and inactive synaptic connections. Furthermore, natural-gradient descent implies the presence of another plasticity component $\gamma_{\mathrm{w}} \boldsymbol{f}(\boldsymbol{w})$ which adapts the synapses depending on their current weight. More specifically, connections that are already strong are subject to larger changes compared to weaker ones. Since the proportionality factor $\gamma_{\mathrm{w}}$ only depends on global variables such as the membrane potential, this component also affects both active and inactive synapses.

The full expressions for $\gamma_{\mathrm{s}}$, $\gamma_{\mathrm{u}}$, and $\gamma_{\mathrm{w}}$ are functions of the membrane potential, its mean and its variance, which represent synapse-local quantities. In addition, $\gamma_{\mathrm{u}}$ and $\gamma_{\mathrm{w}}$ also depend on the total input $\sum_{i=1}^n x_i^\epsilon$ and the total instantaneous presynaptic rate $\sum_{i=1}^n r_i$. However, under reasonable

assumptions such as a high number of presynaptic partners and for a large, diverse set of empirically tested scenarios, we have shown that these factors can be reduced to simple functions of variables that are fully accessible at the locus of individual synapses:

$$\gamma_{\mathrm{u}} \approx c_{\epsilon} \quad \text{and} \quad \gamma_{\mathrm{w}} \approx c_{\mathrm{w}} V, \tag{15}$$

where $c_{\epsilon}$, $c_{\mathrm{w}}$ are constants (Sec. 'Global scaling factor' and 'Empirical Analysis of $\gamma_{\mathrm{u}}$ and $\gamma_{\mathrm{w}}$'). The above learning rule along with closed-form expressions for these factors represent the main analytical findings of this paper.

In the following, we demonstrate that the additional terms introduced in natural-gradient-based plasticity confer important advantages compared to Euclidean-gradient descent, both in terms of of convergence as well as with respect to biological plausibility. More precisely, we show that our plasticity rule improves convergence in a supervised learning task involving an anisotropic cost function, a situation which is notoriously hard to deal with for Euclidean-gradient-based learning rules (*Ruder, 2016*). We then proceed to investigate natural-gradient learning from a biological point of view, deriving a number of predictions that can be experimentally tested, with some of them related to in vivo observations that are otherwise difficult to explain with classical gradient-based learning rules.

## Natural-gradient plasticity speeds up learning

Non-isotropic cost landscapes can easily be provoked by non-homogeneous input conditions. In nature, these can arise under a wide range of circumstances, for elementary reasons that boil down to morphology (the position of a synapse along a dendrite can affect the attenuation of its afferent input) and function (different afferents perform different computations and thus behave differently). To evaluate the convergence behavior of our learning rule and compare it to Euclidean-gradient descent, we considered a very generic situation in which a neuron is required to map a diverse set of inputs onto a target output.

In order to induce a simple and intuitive anisotropy of the error landscape, we divided the afferent population into two equally sized groups of neurons with different firing rates (*Figure 3A*, firing rates were chosen as 10 Hz for group one and 50 Hz for group 2). The input spikes were low-pass-filtered with a difference of exponentials (see Sec. 'Neuron model for details'). This resulted in an asymmetric cost function (KL-divergence between the student and the target firing distribution, *Equation 9*), as visible from the elongated contour lines (*Figure 3D and E*). We further chose a realizable teacher by simulating a different neuron with the same input populations connected via a predefined set of target weights $w^*$. For the weight path plots in *Figure 3D and E* fixed target weight $w^* = (-\frac{0.3}{n}, \frac{0.5}{n})^T$ was chosen, whereas for the learning curve in *Figure 3F*, the target weight components were randomly sampled from a uniform distribution on $[-1/n, 1/n]$ (see Sec. 'Supervised Learning Task' for further details). *Figure 3B and C* shows that our natural-gradient rule enables the student neuron to adapt its weights to reproduce the teacher voltage $V^*$ and thereby its output distribution.

In the following, we compare learning in two student neurons, one endowed with Euclidean-gradient plasticity (*Equation 10*, *Figure 3D*) and one with our natural-gradient rule (*Equation 13*, *Figure 3E*). To better visualize the difference between the two rules, we used a two-dimensional input weight space, that is, one neuron per afferent population. While the negative Euclidean-gradient vectors stand, by definition, perpendicular to the contour lines of $C$, the negative natural-gradient vectors point directly towards the target weight configuration $w^*$. Due to the anisotropy of $C$ induced by the different input rates (see also *Figure 2B*), Euclidean-gradient learning starts out by mostly adapting the high-rate afferent weight and only gradually begins learning the low-rate afferent. In contrast, natural gradient adapts both synaptic weights homogeneously. This is clearly reflected by paths traced by the synaptic weights during learning.

Overall, this lead to faster convergence of the natural-gradient plasticity rule compared to Euclidean-gradient descent. In order to enable a meaningful comparison, learning rates were tuned separately for each plasticity rule in order to optimize their respective convergence speed. The faster convergence of natural-gradient plasticity is a robust effect, as evidenced in *Figure 3F* by the average learning curves over 1,000 trials.

In addition to the functional advantages described above, natural-gradient learning also makes some interesting predictions about biology, which we address below.

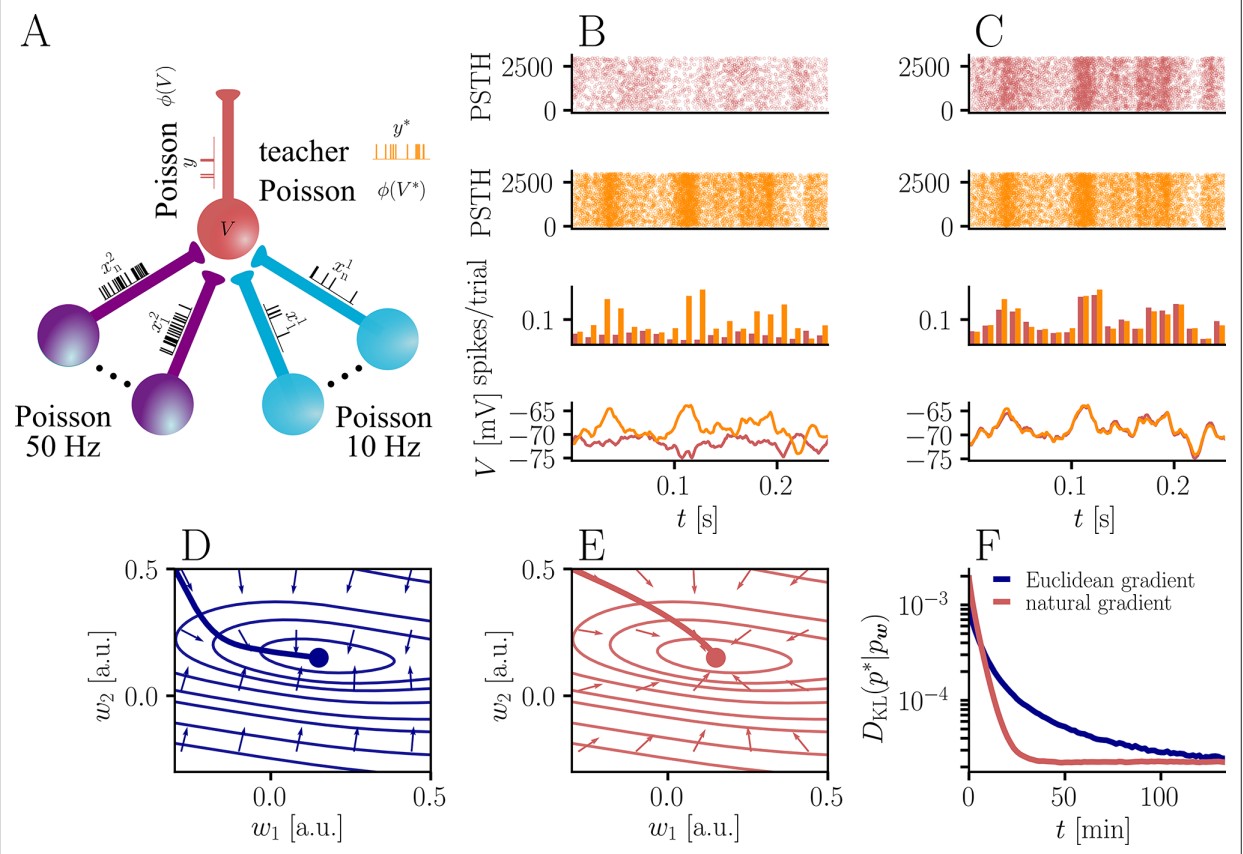

**Figure 3.** Natural-gradient plasticity speeds up learning in a simple regression task. (**A**) We tested the performance of the natural gradient rule in a supervised learning scenario, where a single output neuron had to adapt its firing distribution to a target distribution, delivered in form of spikes from a teacher neuron. The latter was modeled as a Poisson neuron firing with a time-dependent instantaneous rate,$\phi\left(\sum_{i=1}^{n} w_i^* x_i^\epsilon\right)$ where $\boldsymbol{w}^*$ represents a randomly chosen target weight vector. The input consisted of Poisson spikes from $n$ afferents, half of them firing at 10 Hz and 50 Hz, respectively. For our simulations, we used $n = 100$ afferents, except for the weight path plots in (**D**) and (**E**), where the number of afferents was reduced to $n = 2$ for illustration purposes. (**B–C**) Spike trains, PSTHs and voltage traces for teacher (orange) and student (red) neuron before (**B**) and after (**C**) learning with natural-gradient plasticity. During learning, the firing patterns of the student neuron align to those of the teacher neuron. The structure in these patterns comes from autocorrelations in this instantaneous rate. These, in turn, are due to mechanisms such as the membrane filter (as seen in the voltage traces) and the nonlinear activation function. (**D–E**) Exemplary weight evolution during Euclidean-gradient (**D**) and natural-gradient (**E**) learning given $n = 2$ afferents with the same two rates as before. Here, $w_1$ corresponds to $x^1$ in panel A (10 Hz input) and $w_2$ to $x^2$ (50 Hz input). Thick solid lines represent contour lines of the cost function $C$. The respective vector fields depict normalized negative Euclidean and natural gradients of the cost $C$, averaged over 2000 input samples. The thin solid lines represent the paths traced out by the input weights during learning averaged over 500 trials. (**F**) Learning curves for $n = 100$ afferents using natural-gradient and Euclidean-gradient plasticity. The plot shows averages over 1000 trials with initial and target weights randomly chosen from a uniform distribution $\mathcal{U}(-1/n, 1/n)$. Fixed learning rates were tuned for each algorithm separately to exhibit the fastest possible convergence to a root mean squared error of 0.8 Hz in the student neuron's output rate.

## Democratic plasticity

As discussed in the introduction, classical gradient-based learning rules do not usually account for neuron morphology. Since attenuation of PSPs is equivalent to weight reparametrization and our learning rule is, by construction, parametrization-invariant, it naturally compensates for the distance between synapse and soma. In *Equation 13*, this is reflected by a component-wise rescaling of the synaptic changes with the inverse of the attenuation function $f'$, which is induced by the Fisher information metric (see also Figure 8 and the corresponding section in the Materials and methods). Under the assumption of passive attenuation along the dendritic tree, we have

$$w_i^s = f_{d_i}(w_i^d) = \alpha_i(d_i) w_i^d, \qquad (16)$$

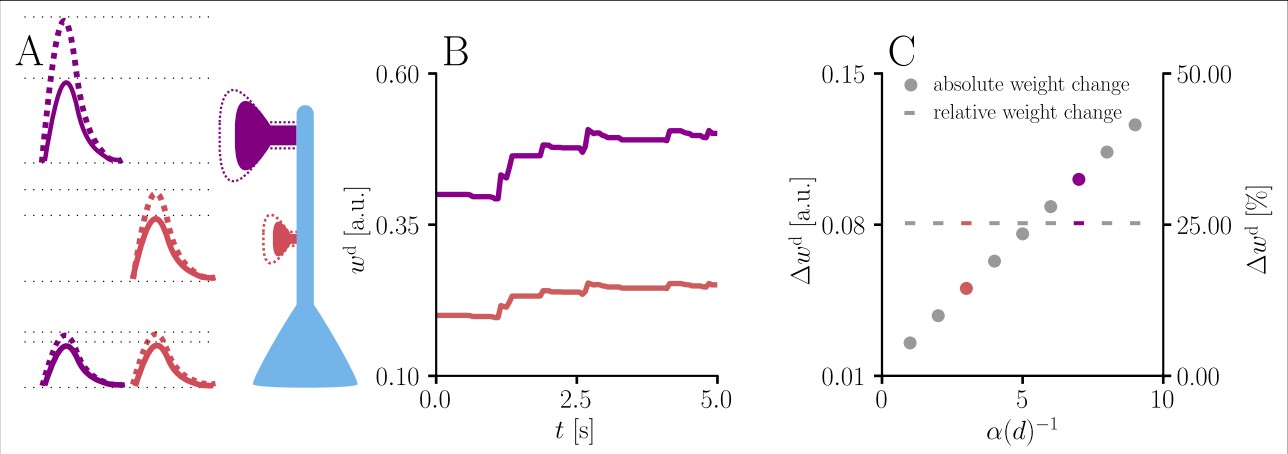

**Figure 4.** Natural-gradient learning scales synaptic weight updates depending on their distance from the soma. We stimulated a single excitatory synapse with Poisson input at 5 Hz, paired with a Poisson teacher spike train at 20 Hz. The distance d from soma was varied between 0 μm and 460 μm and attenuation was assumed to be linear and proportional to the inverse distance from soma. To make weight changes comparable, we scaled dendritic PSP amplitudes with $\alpha(d)^{-1}$ in order for all of them to produce the same PSP amplitude at the soma. (**A**) Example PSPs before (solid lines) and after (dashed lines) learning for two synapses at 3 μm and 7 μm. Application of our natural-gradient rule results in equal changes for the somatic PSPs. (**B**) Example traces of synaptic weights for the two synapses in (**A**). (**C**) Absolute and relative dendritic amplitude change after 5 s as a function of a synapse's distance from the soma.

where $d_i$ denotes the distance of the $i$th synapse from the soma. More specifically, $\boldsymbol{\alpha}(\boldsymbol{d}) = e^{-\boldsymbol{d}/\lambda}$, where $\lambda$ represents the electrotonic length scale. We can write the natural-gradient rule as

$$\dot{\boldsymbol{w}}^{\mathrm{d}} = \eta\,\gamma_{\mathrm{s}}\left[Y^* - \phi(V)\right]\frac{\phi'(V)}{\phi(V)}\left[\frac{c_{\epsilon}}{\boldsymbol{\alpha}(\boldsymbol{d})}\frac{\boldsymbol{x}^{\epsilon}}{\boldsymbol{r}} - \frac{\gamma_{\mathrm{u}}\mathbf{1}}{\boldsymbol{\alpha}(\boldsymbol{d})} + \gamma_{\mathrm{w}}\boldsymbol{w}^{\mathrm{d}}\right]\,. \qquad (17)$$

For functionally equivalent synapses (i.e. with identical input statistics), synaptic changes in distal dendrites are scaled up compared to proximal synapses. As a result, the effect of synaptic plasticity on the neuron's output is independent of the synapse location, since dendritic attenuation is precisely counterbalanced by weight update amplification.

We illustrate this effect with simulations of synaptic weight updates at different locations along a dendritic tree in *Figure 4*. Such 'democratic plasticity', which enables distal synapses to contribute just as effectively to changes in the output as proximal synapses, is reminiscent of the concept of 'dendritic democracy' (*Magee and Cook, 2000*). These experiments show increased synaptic amplitudes in the distal dendritic tree of multiple cell types, such as rat hippocampal CA1 neurons; dendritic democracy has therefore been presumed to serve the purpose of giving distal inputs a 'vote' on the neuronal output. Still, experiments show highly diverse PSP amplitudes in neuronal somata (*Williams and Stuart, 2002*). Our plasticity rule refines the notion of democracy by asserting that learning itself rather than its end result is rescaled in accordance with the neuronal morphology.

Whether such democratic plasticity ultimately leads to distal and proximal synapses having the same effective vote at the soma depends on their respective importance towards reaching the target output. In particular, if synapses from multiple afferents that encode the same information are randomly distributed along the dendritic tree, then democratic plasticity also predicts dendritic democracy, as the scaling of weight changes implies a similar scaling of the final learned weights. However, the absence of dendritic democracy does not contradict the presence of democratic plasticity, as afferents from different cortical regions might target specific positions on the dendritic tree (see, e.g., *Markram et al., 2004*). Furthermore, also with democratic plasticity in place, the functional efficacy of synapses at different locations on the dendritic tree will change differently based on different initial conditions. The experimental findings by *Froemke et al., 2005*, who report an inverse relationship between the amount of plasticity at a synapse and its distance from soma, are therefore completely consistent with the predictions of our learning rule, since they compared synapses that were not initially functionally equivalent.

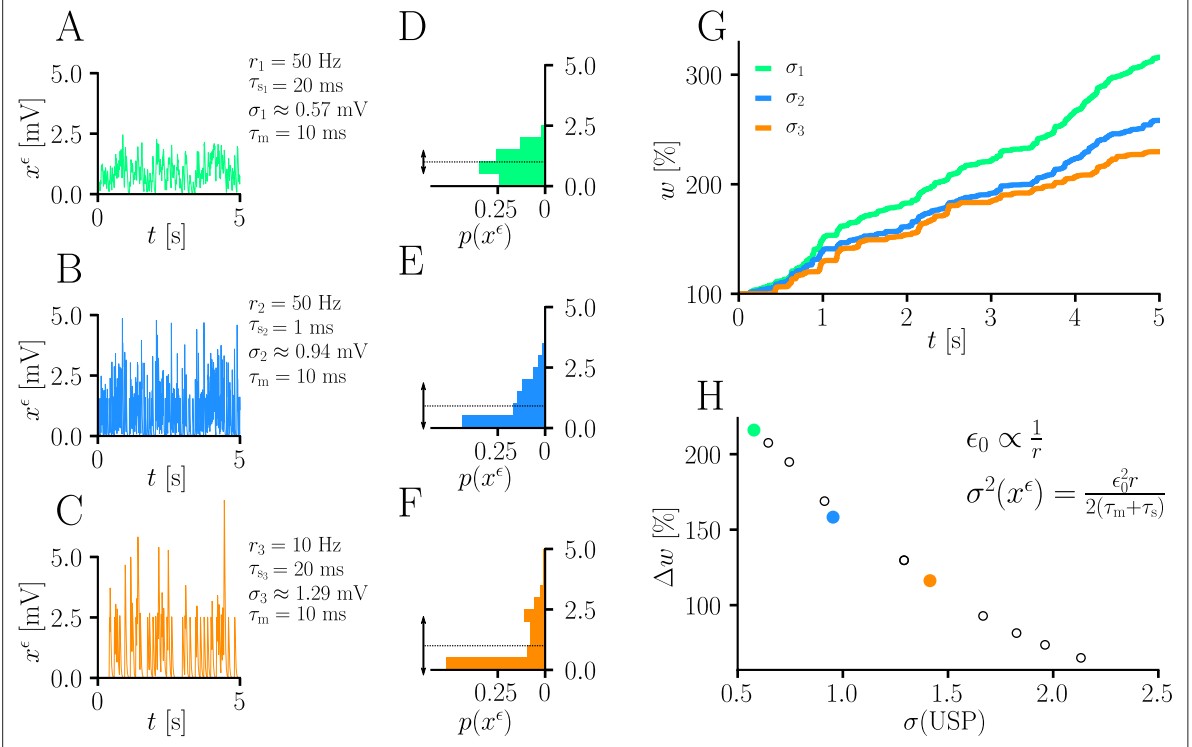

**Figure 5.** Natural-gradient learning scales approximately inversely with input variance. (**A–C**) Exemplary USPs $x_i^\epsilon$ and (**D–F**) their distributions for three different scenarios between which the USP variance $\sigma^2(x_i^\epsilon)$ is varied. In each scenario, a neuron received a single excitatory input with a given rate $r$ and synaptic time constant $\tau_s$. The soma always received teacher spikes at a rate of 80 Hz. To enable a meaningful comparison, the mean USP was conserved by appropriately rescaling the height $\epsilon_0$ of the USP kernel $\epsilon$ (see Sec. 'Neuron model'). (**A,D**) Reference simulation. (**B,E**) Reduced synaptic time constant, resulting in an increased USP variance $\sigma_2^2$. (**C,F**) Reduced input rate, resulting in an increased USP variance $\sigma_3^2$. (**G**) Synaptic weight changes over 5 s for the three scenarios above. (**H**) Total synaptic weight change after $t_0 = 5\,\mathrm{s}$ as a function of USP variance. Each data point represents a different pair of $r$ and $\tau_s$. The three scenarios above are marked with their respective colors.

Note also that dendritic democracy could, in principle, be achieved without democratic plasticity. However, this would be much slower than with natural-gradient learning, especially for distal synapses, as discussed in Sec. 'Natural-gradient plasticity speeds up learning'.

## Input and output-specific scaling

In addition to undoing distortions induced by, for example, attenuation, the natural-gradient rule predicts further modulations of the homosynaptic learning rate. The factor $\gamma_s$ in **Equation 13** represents an output-dependent global scaling factor (for both homo- and heterosynaptic plasticity):

$$\gamma_s = \mathbb{E}\left[\frac{\phi'(V)^2}{\phi(V)}\right]_{p_{\mathrm{usp}}}^{-1}. \tag{18}$$

Together with the $\phi'/\phi$ factor in **Equation 13**, the effective scaling of the learning rule is approximately $1/\phi'$. In other words, it increases the learning rate in regions where the sigmoidal transfer function is flat (see also Sec. 'Global scaling factor'). This represents an unmediated reflection of the philosophy of natural-gradient descent, which finds the steepest path for a small change in output, rather than in the numeric value of some parameter. The desired change in the output requires scaling the corresponding input change by the inverse slope of the transfer function.

Furthermore, synaptic learning rates are inversely correlated to the USP variance $\sigma^2(x^\epsilon)$ (**Figure 5**). In particular, for the homosynaptic component, the scaling is exactly equal to

$$\sigma^2(x_i^\epsilon)^{-1} = c_\epsilon/r_i \tag{19}$$

(see *Equation 13* and Sec. 'Neuron model'). In other words, natural-gradient learning explicitly scales synaptic updates with the (un)reliability – more specifically, with the inverse variance – of their input. To demonstrate this effect in isolation, we simulated the effects of changing the USP variance while conserving its mean. Moreover, to demonstrate its robustness, we independently varied two contributors to the input reliability, namely input rates (which enter $\sigma^2(x^\epsilon)$ directly) and synaptic time constants (which affect the PSP-kernel-dependent scaling constant $c_\epsilon$). *Figure 5* shows how unreliable input leads to slower learning, with an inverse dependence of synaptic weight changes on the USP variance. We note that this observation also makes intuitive sense from a Bayesian point of view, under which any information needs to be weighted by the reliability of its source. Furthermore, the approximately inverse scaling with the presynaptic firing rate is qualitatively in line with observations from *Aitchison and Latham, 2014* and *Aitchison et al., 2021*, although our interpretation and precise relationship is different.

## Interplay of homosynaptic and heterosynaptic plasticity

One elementary property of update rules based on Euclidean-gradient descent is their presynaptic gating, that is, all weight updates are scaled with their respective synaptic input $x^\epsilon$. Therefore, they are necessarily restricted to homosynaptic plasticity, as studied in classical LTP and LTD experiments (*Bliss and Lomo, 1973*; *Dudek and Bear, 1992*). As discussed above, natural-gradient learning retains a rescaled version of this homosynaptic contribution, but at the same time predicts the presence of two additional plasticity components. Contrary to homosynaptic plasticity, these components also adapt synapses to currently non-active afferents, given a sufficient level of global input. Due to their lack of input specificity, they give rise to heterosynaptic weight changes, a form of plasticity that has been observed in hippocampus (*Chen et al., 2013*; *Lynch et al., 1977*), cerebellum (*Ito and Kano, 1982*), and neocortex (*Chistiakova and Volgushev, 2009*), mostly in combination with homosynaptic plasticity. A functional interpretation of heterosynaptic plasticity, to which our learning rule also alludes, is as a prospective adaptation mechanism for temporarily inactive synapses such that, upon activation, they are already useful for the neuronal output.

Our natural-gradient learning rule *Equation 13* can be more summarily rewritten as

$$\Delta w = \Delta w^{\text{hom}} + \Delta w^{\text{het}_u} + \Delta w^{\text{het}_w} ,$$ (20)

where the three additive terms represent the variance-normalized homosynaptic plasticity, the uniform heterosynaptic plasticity and the weight-dependent heterosynaptic plasticity:

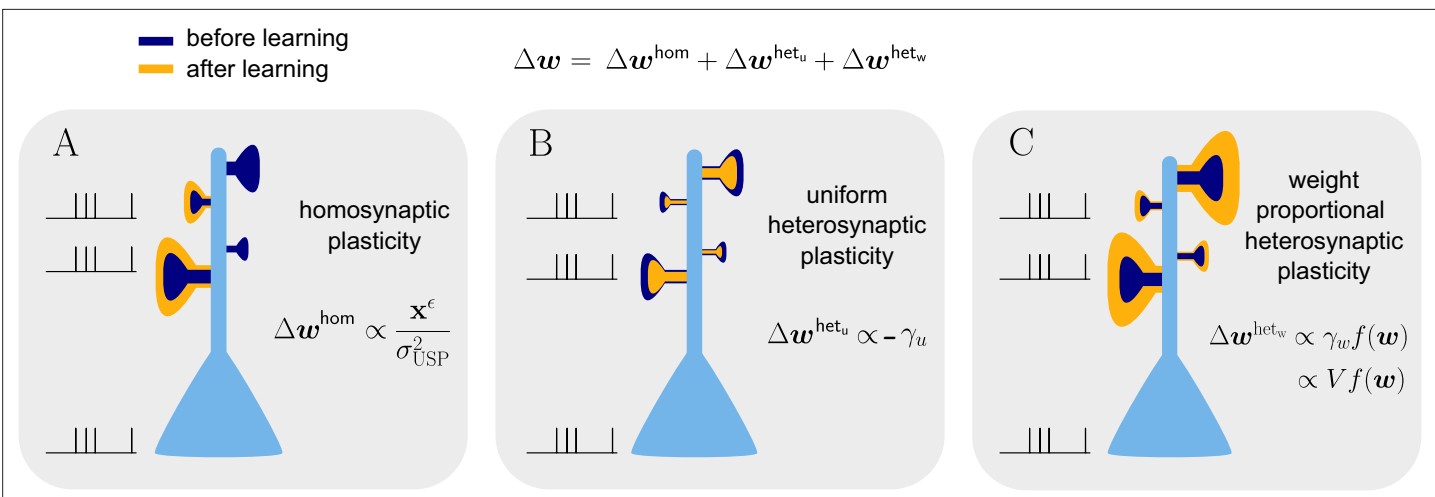

**Figure 6.** Natural-gradient learning combines multiple forms of plasticity. Spike trains to the left of the neuron represent afferent inputs to two of the synapses and teacher input to the soma. The two synapses on the right of the dendritic tree receive no stimulus. The teacher is assumed to induce a positive error. (**A**) The homosynaptic component adapts all stimulated synapses, leaving all unstimulated synapses untouched. (**B**) The uniform heterosynaptic component changes all synapses in the same manner, only depending on global activity levels. (**C**) The proportional heterosynaptic component contributes a weight change that is proportional to the current synaptic strength. The magnitude of this weight change is approximately proportional to a product of the current membrane potential above baseline and the weight vector.

$$\Delta \boldsymbol{w}^{\mathrm{hom}} \propto c_\epsilon \frac{\boldsymbol{x}^\epsilon}{\boldsymbol{r}} \,, \tag{21}$$

$$\Delta \boldsymbol{w}^{\mathrm{het}_{\mathrm{u}}} \propto -\gamma_{\mathrm{u}} \mathbf{1} \,, \tag{22}$$

$$\Delta \boldsymbol{w}^{\mathrm{het}_{\mathrm{w}}} \propto \gamma_{\mathrm{w}} \boldsymbol{f}\left(\boldsymbol{w}\right) \approx c_{\mathrm{w}} V \boldsymbol{f}(\boldsymbol{w}) \,, \tag{23}$$

with the common proportionality factor $\eta \gamma_{\mathrm{s}} \left[ Y^* - \phi(V) \right] \frac{\phi'(V)}{\phi(V)} \boldsymbol{f}'\left(\boldsymbol{w}\right)^{-1}$ composed of the learning rate, the output-dependent global scaling factor, the postsynaptic error, a sensitivity factor and the inverse attenuation function, in order of their appearance. The effect of these three components is visualized in *Figure 6B*. The homosynaptic term $\Delta \boldsymbol{w}^{\mathrm{hom}}$ is experienced only by stimulated synapses, while the two heterosynaptic terms act on all synapses. The first heterosynaptic term $\Delta \boldsymbol{w}^{\mathrm{het}_{\mathrm{u}}}$ introduces a uniform adjustment to all components by the same amount, depending on the global activity level. For a large number of presynaptic inputs, it can be approximated by a constant (see Sec. 'Empirical analysis of $\gamma_{\mathrm{u}}$ and $\gamma_{\mathrm{w}}$'). Furthermore, it usually opposes the homosynaptic change, which we address in more detail below.

In contrast, the contribution of the second heterosynaptic term $\Delta \boldsymbol{w}^{\mathrm{het}_{\mathrm{w}}}$ is weight-dependent, adapting all synapses in proportion to their current strength. This corresponds to experimental reports such as *Loewenstein et al., 2011*, which found in vivo weight changes in the neocortex to be proportional to the spine size, which itself is correlated with synaptic strength (*Asrican et al., 2007*). Our simulations in Sec. 'Empirical Analysis of $\gamma_{\mathrm{u}}$ and $\gamma_{\mathrm{w}}$' show that $\Delta \boldsymbol{w}^{\mathrm{het}_{\mathrm{w}}}$ is roughly a linear function of the membrane potential (more specifically, its deviation with respect to its baseline), which is reflected by the last approximation in *Equation 21*. Since the latter can be interpreted as a scalar product

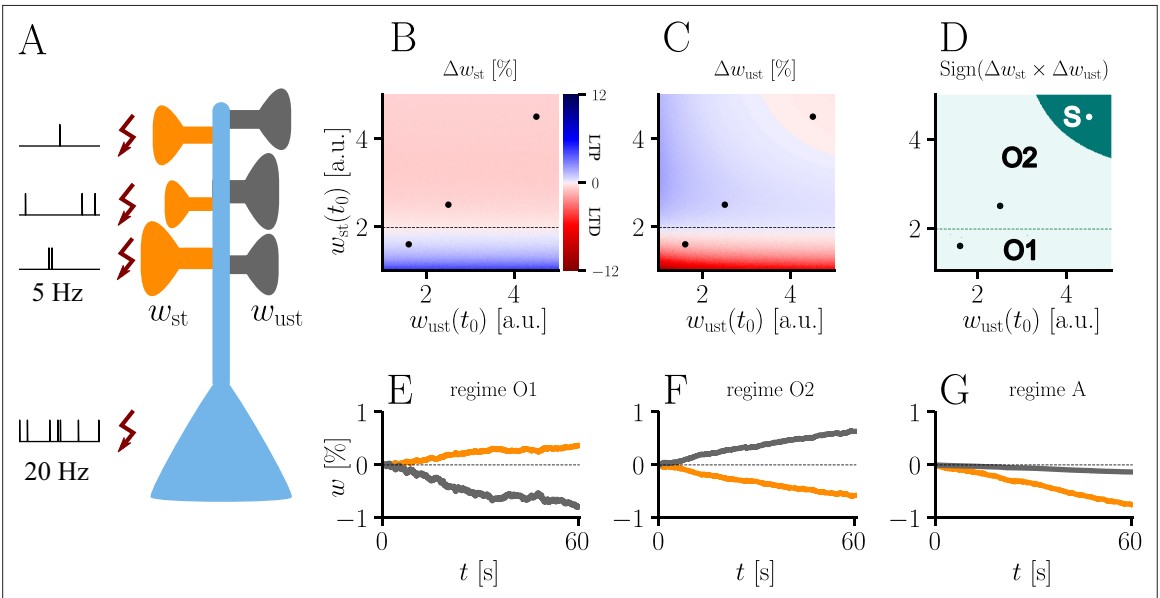

**Figure 7.** Interplay of homo- and heterosynaptic plasticity in natural-gradient learning. (**A**) Simulation setup. Five out of 10 inputs received excitatory Poisson input at 5 Hz. In addition, we assumed the presence of tonic inhibition as a balancing mechanism for keeping the neuron's output within a reasonable regime. Afferent stimulus was paired with teacher spike trains at 20 Hz and plasticity at both stimulated and unstimulated synapses was evaluated in comparison with their initial weights. For simplicity, initial weights within each group were assumed to be equal. (**B**) Weight change of stimulated weights (both homo- and heterosynaptic plasticity are present). These weight changes are independent of unstimulated weights. Equilibrium (dashed black line) is reached when the neuron's output matches its teacher and the error vanishes. For increasing stimulated weights, potentiation switches to depression at the equilibrium line. (**C**) Weight change of unstimulated weights (only heterosynaptic plasticity is present). For very high activity caused by very large synaptic weights, heterosynaptic plasticity always causes synaptic depression. Otherwise, plasticity at unstimulated synapses behaves exactly opposite to plasticity at stimulated synapses. Increasing the size of initial stimulated weights results in a change from depression to potentiation at the same point where potentiation turns into depression at stimulated synapses. (**D**) Direct comparison of plasticity at stimulated and unstimulated synapses. The light green area (O1, O2) represents opposing signs, dark green (**S**) represents the same sign (more specifically, depression). Their shared equilibrium is marked by the dashed green line and represents the switch from positive to negative error. (**E–G**) Relative weight changes of synaptic weights for stimulated and unstimulated synapses during learning, with initial weights picked from the different regimes indicated by the crosses in (**B, C, D**).

between the afferent input vector and the synaptic weight vector, it implies that spikes transmitted by strong synapses have a larger impact on this heterosynaptic plasticity component compared to spikes arriving via weak synapses. Thus, weaker synapses require more persistent and strong stimulation to induce significant changes and 'override the status quo' of the neuron. Since, following a period of learning, afferents connected via weak synapses can be considered uninformative for the neuron's target output, this mechanism ensures a form of heterosynaptic robustness towards noise.

The homo- and heterosynaptic terms exhibit an interesting relationship. To illustrate the nature of their interplay, we simulated a simple experiment (*Figure 7A*) with varying initial synaptic weights for both active and inactive presynaptic afferents. Stimulated synapses (*Figure 7B*) are seen to undergo strong potentiation (LTP) for very small initial weights; the magnitude of weight changes decreases for larger initial amplitudes until the neuron's output matches its teacher, at which point the sign of the postsynaptic error term flips. For even larger initial weights, potentiation at stimulated synapses therefore turns into depression (LTD), which becoms stronger for higher initial values of the stimulated synapses' weights. This is in line with the error learning paradigm, in which changes in synaptic weights seek to reduce the difference between a neuron's target and its output.

For unstimulated synapses (*Figure 7C*), we observe a reversed behavior. For small weights, the negative uniform term $\Delta w^{\mathrm{het_u}}$ dominates and plasticity is depressing. As for the homosynaptic case, the sign of plasticity switches when the weights become large enough for the error to switch sign. Therefore, in the regime where stimulated synapses experienced potentiation, unstimulated synapses are depressed and vice-versa. This reproduces various experimental observations: on one hand, potentiation of stimulated synapses has often been found to be accompanied by depression of unstimulated synapses (*Lynch et al., 1977*), such as in the amygdala (*Royer and Paré, 2003*) or the visual cortex (*Arami et al., 2013*); on the other hand, when the postsynaptic error term switches sign, depression at unstimulated synapses transforms into potentiation (*Wöhrl et al., 2007*; *Royer and Paré, 2003*).

While plasticity at stimulated synapses is unaffected by the initial state of the unstimulated synapses, plasticity at unstimulated synaptic connections depends on both the stimulated and unstimulated weights. In particular, when either of these grow large enough, the proportional term $\Delta w^{\mathrm{het_w}}$ overtakes the uniform term $\Delta w^{\mathrm{het_u}}$ and heterosynaptic plasticity switches sign again. Thus, for very large weights (top right corner of *Figure 7C*), heterosynaptic potentiation transforms back into depression, in order to more quickly quench excessive output activity. This behavior is useful for both supervised and unsupervised learning scenarios (*Zenke and Gerstner, 2017*), where it was shown that pairing Hebbian terms with heterosynaptic and homeostatic plasticity is crucial for stability.

In summary, we can distinguish three plasticity regimes for natural-gradient learning (*Figure 7D–G*). In two of these regimes, heterosynaptic and homosynaptic plasticity are opposed (O1, O2), whereas in the third, they are aligned and lead to depression (S). The two opposing regimes are separated by the zero-error equilibrium line, at which plasticity switches sign.

## Discussion

As a consequence of the fundamentally stochastic nature of evolution, it is no surprise that biology withstands confinement to strict laws. Still, physics-inspired arguments from symmetry and invariance can help uncover abstract principles that evolution may have gradually discovered and implemented into our brains. Here, we have considered parametrization invariance in the context of learning, which, in biological terms, translates to the fundamental ability of neurons to deal with diversity in their morphology and input-output characteristics. This requirement ultimately leads to various forms of scaling and heterosynaptic plasticity that are experimentally well-documented, but can not be accounted for by classical paradigms that regard plasticity as Euclidean-gradient descent. In turn, these biological phenomena can now be seen as a means to jointly improve and accelerate error-correcting learning.

Inspired by insights from information geometry, we applied the framework of natural gradient descent to biologically realistic neurons with extended morphology and spiking output. Compared to classical error-correcting learning rules, our plasticity paradigm requires the presence of several additional ingredients. First, a global factor adapts the learning rate to the particular shape of the voltage-to-spike transfer function and to the desired statistics of the output, thus addressing the diversity of neuronal response functions observed in vivo (*Markram et al., 2004*). Second, the homosynaptic component of plasticity is normalized by the variance of presynaptic inputs, which provides a direct

link to Bayesian frameworks of neuronal computation (*Aitchison and Latham, 2014*; *Jordan et al., 2020*). Third, our rule contains a uniform heterosynaptic term that opposes homosynaptic changes, downregulating plasticity and thus acting as a homeostatic mechanism (*Chen et al., 2013*; *Chistiakova et al., 2015*). Fourth, we find a weight-dependent heterosynaptic term that also accounts for the shape of the neuron's activation function, while increasing its robustness towards noise. Finally, our natural-gradient-based plasticity correctly accounts for the somato-dendritic reparametrization of synaptic strengths.

These features enable faster convergence on non-isotropic error landscapes, in line with results for multilayer perceptrons (*Yang and Amari, 1998*; *Rattray and Saad, 1999*) and rate-based deep neural networks (*Pascanu and Bengio, 2013*; *Ollivier, 2015*; *Bernacchia et al., 2018*). Importantly, our learning rule can be formulated as a simple, fully local expression, only requiring information that is available at the locus of plasticity.

We further note an interesting property of our learning rule, which it inherits directly from the Fisher information metric that underlies natural gradient descent, namely invariance under sufficient statistics (*Cencov, 1972*). This is especially relevant for biological neurons, whose stochastic firing effectively communicates information samples rather than explicit distributions. Thus, downstream computation is likely to require a reliable sample-based, that is, statistically sufficient, estimation of the afferent distribution's parameters, such as the sample mean and variance. This singles out our natural-gradient approach from other second-order-like methods as a particularly appealing framework for biological learning.

Many of the biological phenomena predicted by our invariant learning rule are reflected in existing experimental results. Our democratic plasticity can give rise to dendritic democracy, as observed by *Magee and Cook, 2000*. Moreover, it also relates to results by *Letzkus et al., 2006* and *Sjöström and Häusser, 2006* which describe a sign switch of synaptic plasticity between proximal and distal sites that is not easily reconcilable with naive gradient descent. As the authors themselves speculate, the most likely reason is the (partial) failure of action potential backpropagation through the dendritic tree. In some sense, this is a mirror problem to the issue we discuss here: while we consider the attenuation of input signals, these experimental findings could be explained by an attenuation of the output signal. A simple way of incorporating this aspect into our learning rule (but also into Euclidean rules) is to apply a (nonlinear) attenuation function to the teacher signal $Y^*$ (*Equations 10 and 13*). At a certain distance from the soma, this attenuation would become strong enough to switch the sign of the error term $Y^* - \phi(V)$ and thereby also, for example, LTP to LTD. However, addressing this at the normative level of natural gradient descent as opposed to a direct tweak of the learning rule is less straightforward.

Our rule also requires heterosynaptic plasticity, which has been observed in neocortex, as well as in deeper brain regions such as amygdala and hippocampus (*Lynch et al., 1977*; *Engert and Bonhoeffer, 1997*; *White et al., 1990*; *Royer and Paré, 2003*; *Wöhrl et al., 2007*; *Chistiakova and Volgushev, 2009*; *Arami et al., 2013*; *Chen et al., 2013*; *Chistiakova et al., 2015*), often in combination with homosynaptic weight changes. Moreover, we find that heterosynaptic plasticity generally opposes homosynaptic plasticity, which qualitatively matches many experimental findings (*Lynch et al., 1977*; *White et al., 1990*; *Royer and Paré, 2003*; *Wöhrl et al., 2007*) and can be functionally interpreted as an enhancement of competition. For very large weights, heterosynaptic plasticity aligns with homosynaptic changes, pushing the synaptic weights back to a sensible range (*Chistiakova et al., 2015*), as shown to be necessary for unsupervised learning (*Zenke and Gerstner, 2017*). In supervised learning it helps speed up convergence by keeping the weights in the operating range.

These qualitative matches of experimental data to the predictions of our plasticity rule provide a well-defined foundation for future, more targeted experiments that will allow a quantitative exploration of the relationship between heterosynaptic plasticity and natural gradient learning. In contrast to the mostly deterministic protocols used in the referenced literature, such experiments would require a Poisson-like, stochastic stimulation of the student neuron. To further facilitate a quantitative analysis, the experimental setup should, in particular, allow a clear separation between student and teacher, for example via a pairing protocol (see e.g. *Royer and Paré, 2003*, but note that their protocol was deterministic).

A further prediction that follows from our plasticity rule is the normalization of weight changes by the presynaptic variance. We would thus anticipate that increasing the variance in presynaptic

spike trains (through, e.g. temporal correlations) should reduce LTP in standard plasticity induction protocols. Also, we expect to observe a significant dependence of synaptic plasticity on neuronal response functions and output statistics. For example, flatter response functions should correlate with faster learning, in contrast to the inverse correlation predicted by classical learning rules derived from Euclidean-gradient descent. These propositions remain to be tested experimentally.

By following gradients with respect to the neuronal output rather than the synaptic weights themselves, we were able to derive a parametrization-invariant error-correcting plasticity rule on the single-neuron level. Error-correcting learning rules are an important ingredient in understanding biological forms of error backpropagation (*Sacramento et al., 2017*). In principle, our learning rule can be directly incorporated as a building block into spike-based frameworks of error backpropagation such as (*Sporea and Grüning, 2013*; *Schiess et al., 2016*). Based on these models, top-down feedback can provide a target for the somatic spiking of individual neurons, toward which our learning rule could be used to speed up convergence.

Explicitly and exactly applying natural gradient at the network level does not appear biologically feasible due to the existence of cross-unit terms in the Fisher information matrix $G$. These terms introduce non-localities in the natural-gradient learning rule, in the sense that synaptic changes at one neuron depend on the state of other neurons in the network. However, methods such as the unit-wise natural-gradient approach (*Ollivier, 2015*) could be employed to approximate the natural gradient using a block-diagonal form of $G$. The resulting learning rule would not exhibit biologically implausible co-dependencies between different neurons in the network, while still retaining many of the advantages of full natural-gradient learning. For spiking networks, this would reduce global natural-gradient descent to our local rule for single neurons.

## Materials and methods
### Neuron model
We chose a Poisson neuron model whose firing rate depends on the somatic membrane potential above the resting potential $V_{\text{rest}} = -70$ mV. They relate via a sigmoidal activation function

$$\phi(V) = \frac{\phi_{\text{max}}}{1+\exp[-\beta(V-\theta)]}, \tag{24}$$

with $\beta = 0.3$, $\theta = 10$mV, and a maximal firing rate $\phi_{\text{max}} = 100$ Hz. This means the activation function is centered at $-60$ mV, saturating around $-50$ mV. Note that the derivation of our learning rule does not depend on the explicit choice of activation function, but holds for arbitrary monotonically increasing, positive functions that are sufficiently smooth. For the sake of simplicity, refractoriness was neglected. For the same reason, we assumed synaptic input to be current-based, such that incoming spikes elicit a somatic membrane potential above baseline given by

$$V = \sum_i w_i x_i^\epsilon , \tag{25}$$

where

$$x_i^\epsilon(t) = \left[x_i * \epsilon\right](t) \tag{26}$$

denotes the unweighted synaptic potential (USP) evoked by a spike train

$$x_i = \sum_{t_i^{\text{f}}} \delta\left(t - t_i^{\text{f}}\right) \tag{27}$$

of afferent $i$, and $w_i$ is the corresponding synaptic weight. Here, $t_i^{\text{f}}$ denote the firing times of afferent $i$, and the synaptic response kernel $\epsilon$ is modeled as

$$\epsilon(t) = \epsilon_0 \frac{\Theta(t)}{(\tau_{\text{m}} - \tau_{\text{s}})} \left[\exp\left(-\frac{t}{\tau_{\text{m}}}\right) - \exp\left(-\frac{t}{\tau_{\text{s}}}\right)\right], \tag{28}$$

where $\epsilon_0$ is a scaling factor with units mV ms, and unless specified otherwise, we chose $\epsilon_0 = 1$ mV ms. For slowly changing input rates, mean and variance of the stationary unweighted synaptic potential are then given as (*Petrovici, 2016*)

$$\mathbb{E}\left[x_i^\epsilon\right] \approx \epsilon_0 r_i \tag{29}$$

and

$$\mathrm{Var}\left(x_i^\epsilon\right) \approx c_\epsilon^{-1} r_i\ , \tag{30}$$

with

$$c_\epsilon = \left(\int_0^\infty \epsilon^2 dt\right)^{-1} = 2\frac{\tau_\mathrm{m} + \tau_\mathrm{s}}{\epsilon_0^2}\ . \tag{31}$$

*Equations 29 and 30* are exact for constant input rates. Note that it is *Equation 30* that induces the inverse dependence of the homosynaptic term on the input rate discussed around *Equation 19*.

Unless indicated otherwise, simulations were performed with a membrane time constant $\tau_\mathrm{m} = 10\,\mathrm{ms}$ and a synaptic time constant $\tau_\mathrm{s} = 3\,\mathrm{ms}$. Hence, USPs had an amplitude of 60 mV and were normalized with respect to area under the curve and multiplied by the synaptic weights. Initial and target weights were chosen such that the resulting average membrane potential was within operating range of the activation function. As an example, in *Figure 3F*, the average initial excitatory weight was 0.005, corresponding to an EPSP amplitude of 300V.

In *Figure 5*, the scaling factor $\epsilon_0$ was additionally normalized proportionally to the input rate $r_i$ at the synapse in order to keep the mean USP constant and allow a comparison based solely on the variance.

## Sketch for the derivation of the somatic natural-gradient learning rule

The choice of a Poisson neuron model implies that spiking in a small interval $[t, t + \mathrm{d}t]$ is governed by a Poisson distribution. For sufficiently small interval lengths $\mathrm{d}t$, the probability of having a single spike in $[t, t + \mathrm{d}t]$ becomes Bernoulli with parameter $\phi(V_t)\,\mathrm{d}t$. The aim of supervised learning is to bring this distribution closer to a given target distribution with density $p^*$. The latter is delivered in form of a teacher spike train

$$Y^* = \sum_{t^{f*}} \delta\left(t - t^{f^*}\right)\ , \tag{32}$$

where the probability of having a teacher spike (denoted as $Y^* = 1$) in $[t, t + \mathrm{d}t]$ is Bernoulli with parameter

$$p^*\left(Y_t^* = 1 \mid x_t^\epsilon\right) = \phi_t^*(V)\,\mathrm{d}t. \tag{33}$$

Note that to facilitate the convergence analysis, in *Figure 3* we chose a realizable teacher $\phi_t^* = \phi\left(\sum_{i=1}^n w_i^* x_i^\epsilon\right)$ generated by a given set of teacher weights $w^*$.

We measure the error between the desired and the current input-output spike distribution in terms of the Kullback-Leibler divergence given in *Equation 9*, which attains its single minimum when the two distributions are equal. Note that while the $D_\mathrm{KL}$ is a standard measure to characterize 'how far' two distributions are apart, its behavior can sometimes be slightly unintuitive since it is not a metric. In particular, it is not symmetric and does not satisfy the triangle inequality.

Classical error learning follows the Euclidean gradient of this cost function, given as the vector of partial derivatives with respect to the synaptic weights. A short calculation (Sec. 'Detailed derivation of the natural-gradient learning rule') shows that the resulting Euclidean-gradient-descent learning rule is given by *Equation 10*. By correcting the vector of partial derivatives for the distance distortion between the manifold of input-output distributions and the synaptic weight space, given in terms of the Fisher information matrix $G(w)$, we obtain the natural gradient (*Equation 12*). We then followed an approach by *Amari, 1998* to derive an explicit formula for the product on the right hand side of *Equation 12*.

In Sec. 'Detailed derivation of the natural-gradient learning rule', we show that given independent input spike trains, the Fisher information matrix defined in *Equation 11* can be decomposed with respect to the vector of input rates $r$ and the current weight vector $w$ as

$$G^\mathrm{s}\left(w^\mathrm{s}\right) = c_1\left(\epsilon_0^2 r r^T + \Sigma_\mathrm{usp}\right) + c_2 \epsilon_0 \left[\Sigma_\mathrm{usp} w^\mathrm{s} r^T + r\left(\Sigma_\mathrm{usp} w^\mathrm{s}\right)^T\right] + c_3 \left(\Sigma_\mathrm{usp} w^\mathrm{s}\right)\left(\Sigma_\mathrm{usp} w^\mathrm{s}\right)^T. \tag{34}$$

Here,

$$\Sigma_{\text{usp}} = \text{diag}\{c_\epsilon^{-1} r_i\}_{i=1}^n \tag{35}$$

is the covariance matrix of the unweighted synaptic potentials, and $c_1, c_2, c_3$ are coefficients (see *Equation 82* for their definition) depending on the mean $\mu_V$ and variance $\sigma_V^2$ of the membrane potential, and on the total rate

$$q = c_\epsilon \epsilon_0^2 \sum_i r_i . \tag{36}$$

Through repeated application of the Sherman-Morrison-Formula, the inverse of $G(w)$ can be obtained as

$$G^s \left( w^s \right)^{-1} = \gamma_s \left[ \Sigma_{\text{usp}}^{-1} + \left( c_\epsilon \epsilon_0 g_1 \mathbf{1} + g_2 w^s \right) c_\epsilon \epsilon_0 \mathbf{1}^T + \left( c_\epsilon \epsilon_0 g_3 \mathbf{1} + g_4 w^s \right) w^{sT} \right] . \tag{37}$$

Here the coefficients $\gamma_s, g_1, g_2, g_3, g_4$, which are defined in *Equation 102*, are again functions of mean and variance of the membrane potential, and of the total rate. Consequently, the natural-gradient rule in terms of somatic amplitudes is given by

$$\dot{w}^s = \gamma_s \left[ Y^* - \phi(V) \right] \frac{\phi'(V)}{\phi(V)} \left( \frac{c_\epsilon x^\epsilon}{r} - \gamma_u \mathbf{1} + \gamma_w w^s \right) . \tag{38}$$

Note that the formulas for,

$$\gamma_u = -c_\epsilon \epsilon_0 \left( g_1 c_\epsilon \epsilon_0 \sum_{i=1}^n x_i^\epsilon + g_3 V \right) \quad \text{and} \quad \gamma_w = \left( g_2 c_\epsilon \epsilon_0 \sum_{i=1}^n x_i^\epsilon + g_4 V \right) \tag{39}$$

arise from the product of the inverse Fisher information matrix and the Euclidean gradient, using $\mathbf{1}^T x^\epsilon = \sum_{i=1}^n x_i^\epsilon$ and $w^{sT} x^\epsilon = V$. Due to the complicated expressions for $g_1, \ldots, g_4$ (*Equation 82*), *Equation 39* only provides limited information about the behavior of $\gamma_u$ and $\gamma_w$. Therefore, we performed an empirical analysis based on simulation data (Sec. 'Empirical analysis of $\gamma_u$ and $\gamma_w$'; Figure 11). In a stepwise manner we first evaluated $g_1, \ldots, g_4$ under various conditions, which revealed that the products with $g_2$ and $g_3$ in *Equation 39* are neglible in most cases compared to the other terms, hence

$$\gamma_u \approx c_\epsilon^2 \epsilon_0^2 \sum_{i=1}^n x_i^\epsilon \text{ and } \gamma_w \approx g_4 V. \tag{40}$$

Furthermore, for a sufficient number of input afferents, we can approximate $g_1 \approx -q^{-1}$. Since $q = c_\epsilon \epsilon_0 \mathbb{E} \left[ \sum_{i=1}^n x_i^\epsilon \right]$, by the central limit theorem we have

$$\gamma_u \approx c_u c_\epsilon \tag{41}$$

for large $n$ and $c_u = \epsilon_0 = 1$. Moreover, while the variance of $g_4$ across weight samples increases with the number and firing rate of input afferents, its mean stays approximately constant across conditions. This lead to the approximation

$$\gamma_w \approx c_w V, \tag{42}$$

where $c_w$ is constants across weights, input rates and the number of input afferents.

To evaluate the quality of our approximations, we tested the performance of learning in the setting of *Figure 3* when $\gamma_u$ and $\gamma_w$ were replaced by their approximations (*Equations 112; 113*). The test was performed for several input patterns (Sec. 'Evaluation of the approximated natural-gradient rule', Sec. 'Performance of the approximated learning rule'). It turned out that a convergence behavior very similar to natural-gradient descent could be achieved with $c_u = 0.95$, which worked much better in practice than $c_u = 1$. For $c_w$ a choice of 0.05 which was close to the mean of $c_w$ worked well.

For these input rate configurations and choices of constants, we additionally sampled the negative gradient vectors for random initial weights and USPs (Sec. 'Evaluation of the approximated natural-gradient rule', Sec. 'Performance of the approximated learning rule') and compared the angles and length difference between natural-gradient vectors and the approximation to the ones between natural and Euclidean gradient.

## Reparametrization and the general natural-gradient rule

To arrive at a more general form of the natural-gradient learning rule, we consider a parametrization $w$ of the synaptic weights which is connected to the somatic amplitudes via a smooth component-wise coordinate change $f = (f_1, \ldots, f_n)$, such that $w_i^s = f_i(w_i)$. A Taylor expansion shows that small weight changes then relate via the derivative of $f$

$$\Delta w_i^s = f_i'(w_i)\, \Delta w_i . \tag{43}$$

On the other hand, we can also express the cost function in terms of $w$, with $C[w] = C^s[f(w^s)]$, and directly calculate the Euclidean gradient of $C$ in terms of $w$. By the chain rule, we then have

$$\Delta w = -\nabla_w^E C = -\frac{\partial C}{\partial w} = -\frac{\partial C^s}{\partial w^s}\frac{\partial w^s}{\partial w} = -\mathrm{diag}\{f_i'(w_i)\}\nabla_w^E C^s = \mathrm{diag}\{f_i'(w_i)\}\Delta w^s. \tag{44}$$

Plugging this into **Equation 43**, we obtain an inconsistency: $\Delta w_i^s = f_i'(w_i)^2 \Delta w_i^s$. Hence the predictions of Euclidean-gradient learning depend on our choice of synaptic weight parametrization (**Figure 1**).

In order to obtain the natural-gradient learning rule in terms of $w$, we first express the Fisher Information Matrix in the new parametrization, starting with **Equation 60**

$$G(w) = \mathbb{E}\left[\frac{\partial \log p_w}{\partial w}\frac{\partial \log p_w}{\partial w}^T\right]_{p_w} \tag{45}$$

$$= \mathbb{E}\left[\left(\frac{\partial \log p_w}{\partial w^s}\frac{\partial w^s}{\partial w}\right)\left(\frac{\partial \log p_w}{\partial w^s}\frac{\partial w^s}{\partial w}\right)^T\right]_{p_w} \tag{46}$$

$$= \mathrm{diag}\{f_i'(w_i)\}\mathbb{E}\left[\frac{\partial \log p_w}{\partial w^s}\frac{\partial \log p_w}{\partial w^s}^T\right]_{p_w}\mathrm{diag}\{f_i'(w_i)\} \tag{47}$$

$$= \mathrm{diag}\{f_i'(w_i)\}G^s(w^s)\,\mathrm{diag}\{f_i'(w_i)\}. \tag{48}$$

Inserting both **Equation 44** and **Equation 47** into **Equation 12**, we obtain the natural-gradient rule in terms of $w$ (**Equation 13**). As illustrated in **Figure 8**, unlike for Euclidean-gradient descent, the result is consistent with **Equation 43**.

## Simulation details

All simulations were performed in python and used the numpy and scipy packages. Differential equations were integrated using a forward Euler method with a time step of 0.5 ms.

### Supervised learning task

A single output neuron was trained to spike according to a given target distribution in response to incoming spike trains from n independently firing afferents. To create an asymmetry in the input, we chose one half of the afferents' firing rates as 10 Hz, while the remaining afferents fired at 50 Hz. The supervision signal consisted of spike trains from a teacher that received the same input spikes. To allow an easy interpretation of the results, we chose a realizable teacher, firing with rate $\phi(V^*)$, where $V^* = w^{*T}x^\epsilon$ for some optimal set of weights $w^*$. However, our theory itself does not include assumptions about the origin and exact form of the teacher spike train.

For the learning curves in **Figure 3F**, initial and target weight components were chosen randomly (but identically for the two curves) from a uniform distribution on $\mathcal{U}(-1/n, 1/n)$, corresponding to maximal PSP amplitudes between $-600\ \mu V$ and $600\ \mu V$ for the simulation with $n = 100$ input neurons. Learning curves were averaged over 1000 initial and target weight configurations. In addition, the minimum and maximum values are shown in **Figure 9**, as well as the mean Euclidean distance between student and teacher weights and between student and teacher firing rates. We did not enforce Dales' law, thus about half of the synaptic input was inhibitory at the beginning but sign changes were permitted. This means that the mean membrane potential above rest covered a maximal range $[-30\ \text{mV}, 30\ \text{mV}]$. Learning rates were optimized as $\eta_n = 6 * 10^{-4}$ for the natural-gradient-descent algorithm and $\eta_e = 4.5 * 10^{-7}$ for Euclidean-gradient descent, providing the fastest possible convergence to a residual root mean squared error in output rates of 0.8 Hz. To confirm that the convergence

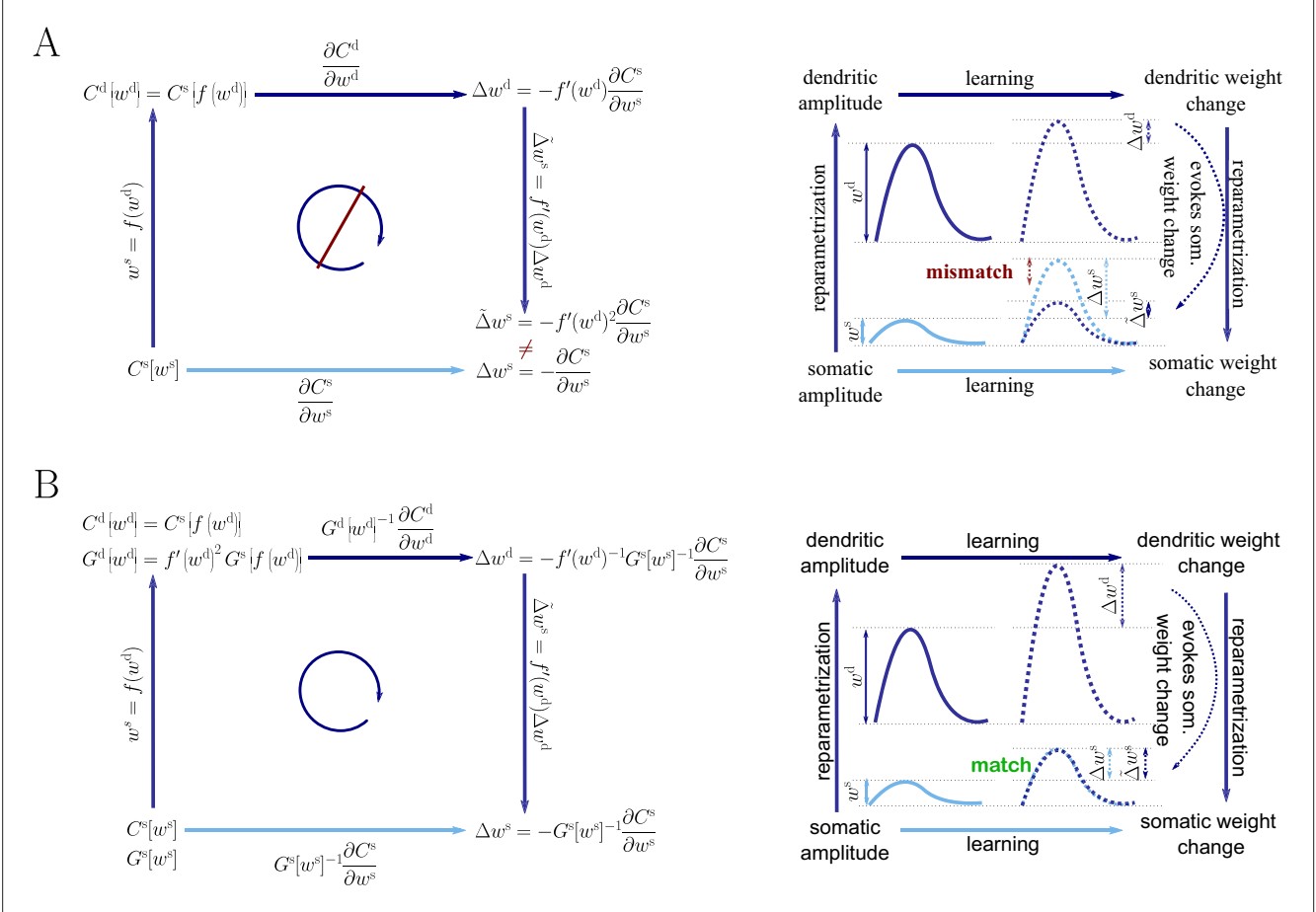

**Figure 8.** Natural-gradient descent does not depend on chosen parametrization. Mathematical derivation and phenomenological correlates. EPSPs before learning are represented as continuous, after learning as dashed curves. The light blue arrow represents gradient descent on the error as a function of the somatic EPSP $C^s[w^s]$ (also shown in light blue). The resulting weight change leads to an increase $\Delta w^s$ in the somatic EPSP after learning. The dark blue arrows track the calculation of the same gradient, but with respect to the dendritic EPSP (also shown in dark blue): (1) taking the attenuation into account in order to compute the error as a function of $w^d$, (2) calculating the gradient, followed by (3) deriving the associated change in $\tilde{\Delta} w^s$, again considering attenuation. (**A**) For Euclidean-gradient descent. (**B**) For natural-gradient descent. Unlike for Euclidean-gradient descent, the factor $f'(w)^2$ is compensated, since its inverse enters via the Fisher information. This leads to the synaptic weights updates, as well as the associated evolution of a neuron's output statistics over time, being equal under the two parametrizations.

behavior of both natural-gradient and Euclidean-gradient learning is robust, we varied the learning rate between $0.5\eta_0$ and $2\eta_0$ and measured the time until the $D_{KL}$ first reached a value of $5 * 10^{-5}$ (**Figure 9**). Here $\eta_0 = \eta_e$ for Euclidean-gradient descent and $\eta_0 = \eta_n$ for natural-gradient descent.

Per trial, the expectation over USPs in the cost function was evaluated on a randomly sampled test set of 50 USPs that resulted from input spike trains of 250 ms. The expectation over output spikes was calculated analytically.

For the weight path simulation with two neurons (**Figure 3D–E**), we chose a fixed initial weight $w_0 = \left(-\frac{0.3}{n}, \frac{0.5}{n}\right)^T$, a fixed target weight $w^* = \left(\frac{0.15}{n}, \frac{0.15}{n}\right)^T$, and learning rates $\eta_n = 2.5 * 10^{-4}$ and $\eta_e = 7. * 10^{-7}$. Weight paths were averaged over 500 trials of 6000s duration each.

The vector plots in **Figure 3D–E** display the average negative normalized natural and Euclidean-gradient vectors across 2000 USP samples per synapse ($n = 2$) on a grid of weight positions on $\left[\frac{-0.4}{n}, \frac{0.6}{n}\right]^2$, with the first coordinate of the gridpoints in $\left\{\frac{-0.28}{n}, \frac{-0.1}{n}, \frac{0.08}{n}, \frac{0.26}{n}, \frac{0.44}{n}\right\}$ and the second in $\left\{\frac{-0.22}{n}, \frac{-0.02}{n}, \frac{0.26}{n}, \frac{0.5}{n}\right\}$. Each USP sample was the result of a 1 s spike train at rate $r_1 = 10$ Hz and $r_2 = 50$ Hz respectively. The contour lines were obtained from 2000 samples of the $D_{KL}$ along a grid

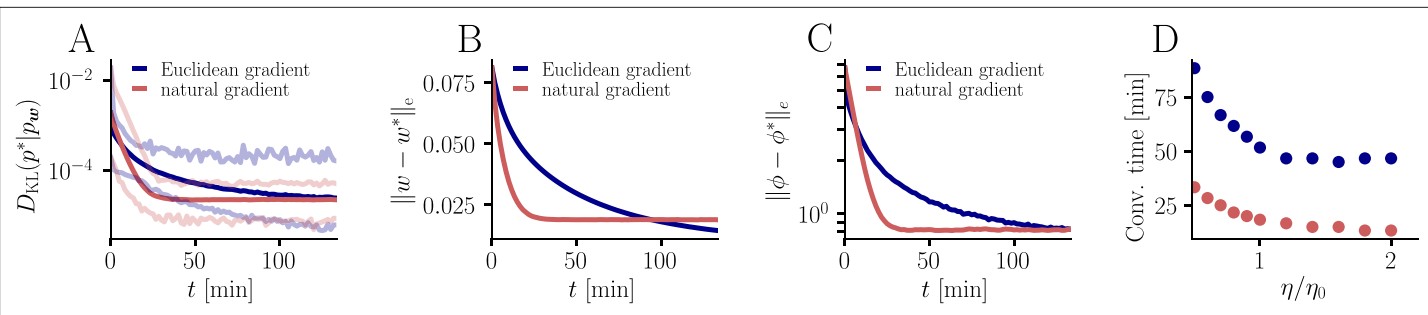

**Figure 9.** Further convergence analysis of natural-gradient-descent learning. Unless stated otherwise, all simulation parameters are the same as in *Figure 3*. (**A**) In addition to the average learning curves from *Figure 3F* (solid lines), we show the minimum and maximum values (semi-transparent lines) during learning. (**B**) Plot of the mean Euclidean distance between student and teacher weight. Note that a smaller distance in weights does not imply a smaller $D_{KL}$, nor a smaller distance in firing rates. This is due to the non-linear relationship between weights and firing rates. (**C**) Development of mean Euclidean distance between student and teacher firing rate during learning. (**D**) Robustness of learning against perturbations of the firing rate. We varied the learning rate for natural-gradient and Euclidean-gradient descent relative to the learning rate $\eta_0$ used in the simulations for *Figure 3F* (EGD: $\eta_0 = \eta_e = 4.5 * 10^{-7}$, NGD: $\eta_0 = \eta_n = 6 * 10^{-4}$), and measured the time until the $D_{KL}$ first reached a value of $5 * 10^{-5}$.

on $\left[\frac{-0.4}{n}, \frac{0.6}{n}\right]^2$ (distance between two grid points in one dimension: $\frac{0.006}{n}$) and displayed at the levels 0.001, 0.003, 0.005, 0.009, 0.015, 0.02, 0.03, 0.04.

For the plots in *Figure 3B–C*, we used initial, final, and target weights from a sample of the learning curve simulation. We then randomly sampled input spike trains of 250 ms length and calculated the resulting USPs and voltages according to *Equation 26* and *Equation 25*. The output spikes shown in the raster plot were then sampled from a discretized Poisson process with $dt = 5. * 10^{-4}$. We then calculated the PSTH with a bin size of 12.5 ms.

## Distance dependence of amplitude changes

A single excitatory synapse received Poisson spikes at 5 Hz, paired with Poisson teacher spikes at 20 Hz. The distance from the soma was varied between 0 μm and 460 μm. Learning was switched on for 5 s with an initial weight corresponding to 0.05 at the soma, corresponding to a PSP amplitude of 3 mV. Initial dendritic weights were scaled up with the proportionality factor $\alpha(d)^{-1}$ depending on the distance from the soma, in order for input spikes to result in the same somatic amplitude independent of the synaptic position. Example traces are shown for $\alpha(d)^{-1} = 3$ and $\alpha(d)^{-1} = 7$.

## Variance dependence of amplitude changes

We stimulated a single excitatory synapse with Poisson spikes, while at the same time providing Poisson teacher spike trains at 80 Hz. To change USP variance independently from mean, unlike in the other exercises, the input kernel in *Equation 28* was additionally normalized by the input rate. USP variance was varied by either keeping the input rate at 10 Hz while varying the synaptic time constant $\tau_s$ between 1 ms and 20 ms, or fixing $\tau_s$ at 20 ms and varying the input rate between 10 Hz and 50 Hz.

## Comparison of homo- and heterosynaptic plasticity

Out of $n$ =10 excitatory synapses of a neuron, we stimulated 5 by Poisson spike trains at 5 Hz, together with teacher spikes at 20 Hz, and measured weight changes after 60 s of learning. To avoid singularities in the homosynaptic term of learning rule, we assumed in the learning rule that unstimulated synapses received input at some infinitesimal rate, thus effectively setting $\Delta w^{hom} = 0$. Initial weights for both unstimulated and stimulated synapses were varied between $\frac{1}{n}$ and $\frac{5}{n}$. For reasons of simplicity, all stimulated weights were assumed to be equal, and tonic inhibition was assumed by a constant shift in baseline membrane potential of -5 mV. Example weight traces are shown for initial weights of $\frac{1.6}{n}, \frac{2.5}{n}$, and $\frac{4.5}{n}$ for both stimulated and unstimulated weights. The learning rate was chosen as $\eta = 0.01$.

For the plots in *Figure 7B C*, we chose the diverging colormap matplotlib.cm.seismic in matplotlib. The colormap was inverted such that red indicates negative values representing synaptic depression and blue indicates positive values representing synaptic potentiation. The colormap was linearly

discretized into 500 steps. To avoid too bright regions where colors cannot be clearly distinguished, the indices 243–257 were excluded. The colorbar range was manually set to a symmetric range that includes the max/min values of the data. In *Figure 7D*, we only distinguish between regions where plasticity at stimulated and at unstimulated synapses have opposite signs (light green), and regions where they have the same sign (dark green).

## Approximation of learning-rule coefficents

We sampled the values for $g_1, \ldots, g_4$ from *Equation 82* for different afferent input rates. The input rate $r$ was varied between 5 Hz and 55 Hz for $n = 100$ neurons. The coefficients were evaluated for randomly sampled input weights (20 weight samples of dimension $n$, each component sampled from a uniform distribution $\mathcal{U}\left(-5/n, 5/n\right)$).

In a second simulation, we varied the number $n$ of afferents between 10 and 200 for a fixed input rate of 20 Hz, again for randomly sampled input weights (20 weight samples of dimension $n$, each component sampled from a uniform distribution $\mathcal{U}\left(-5/n, 5/n\right)$).

In a next step, we compared the sampled values of $g_1$ as a function of the total input rate $n * r$ to the values of the approximation given by $g_1 \approx -q^{-1}$ ($r$ between 5 Hz and 55 Hz, $n$ between 10 and 200 neurons, 20 weight samples of dimension $n$, each component sampled from a uniform distribution $\mathcal{U}\left(-5/n, 5/n\right)$).

Afterwards, we plotted the sampled values of $\gamma_u$ as a function of the approximation $s$ (*Equation 111*, $r$ between 5 Hz and 55 Hz, $n = 100$, 20 weight samples of dimension $n$, each component sampled from a uniform distribution $\mathcal{U}\left(-5/n, 5/n\right)$, 20 USP-samples of dimension $n$ for each rate/weight-combination).

Next, we investigated the behavior of $\gamma_w$ as a function of $g_4 V$ ($r$ between 5 Hz and 55 Hz, $n = 100$, 20 weight samples of dimension $n$, each component sampled from a uniform distribution $\mathcal{U}\left(-5/n, 5/n\right)$, 20 USP-samples of dimension $n$ for each rate/weight-combination), and in last step, as a function of $c_w V$ with a constant $c_w = 0.05$.

## Evaluation of the approximated natural-gradient rule

We evaluated the performance of the approximated natural-gradient rule in *Equation 114* (with $c_u = 0.95$ and $c_w = 0.05$) compared to Euclidean-gradient descent and the full rule in *Equation 13* in the learning task of *Figure 3* under different input conditions (n=100, Group 1: 10 Hz/ Group 2: 30 Hz, Group 1: 10 Hz/ Group 2: 50 Hz, Group 1: 20 Hz/ Group 2: 20 Hz, Group 1: 20 Hz/ Group 2: 40 Hz). The learning curves were averaged over 1000 trials with input and target weight components randomly chosen from a uniform distribution on $\mathcal{U}\left(-1/n, 1/n\right)$. Learning rate parameters were tuned individually for each learning rule and scenario according to *Table 1*. All other parameters were the same as for *Figure 3F*.

For the angle histograms in Figure 12A-B, we simulated the natural, Euclidean and approximated natural weight updates for several input and initial weight conditions. Similar to the setup in *Figure 3* we separated the $n = 100$ input afferents in two groups firing at different rates (Group1/Group2: 10 Hz/10 Hz, 10 Hz/30 Hz, 10 Hz/50 Hz, 20 Hz/20 Hz, 20 Hz/40 Hz). For each input pattern, 100 Initial weight components were sampled randomly from a uniform distribution $\mathcal{U}\left(-5/n, 5/n\right)$, while the target weight was fixed at $\boldsymbol{w}^* = \left(\frac{0.15}{n}, \frac{0.15}{n}\right)^T$. For each initial weight, 100-long input spike trains were sampled and the average angle between the natural-gradient weight update and the approximated natural-gradient weight update at $t = 1$ s was calculated. The same was done for the average angle between the natural and the Euclidean weight update.

**Table 1.** Learning rates.

| $r_1$ | $r_2$ | $\eta_n$ | $\eta_a$ | $\eta_e$ |
|---|---|---|---|---|
| 10 Hz | 30 Hz | 0.000655 | 0.00055 | 0.00000110 |
| 10 Hz | 50 Hz | 0.000600 | 0.00045 | 0.00000045 |
| 20 Hz | 20 Hz | 0.000650 | 0.00053 | 0.00000118 |
| 20 Hz | 40 Hz | 0.000580 | 0.00045 | 0.00000055 |

## Detailed derivation of the natural-gradient learning rule

Here, we summarize the mathematical derivations underlying our natural-gradient learning rule (*Equation 13*). While all derivations in Sec. 'Detailed derivation of the natural-gradient learning rule' and Sec. 'Inverse of the Fisher Information Matrix' are made for the somatic parametrization and can then be extended to other weight coordinates as described in Sec. 'Reparametrization and the general natural-gradient rule', we drop the index s in $w^s$ for the sake of readability.

Supervised learning requires the neuron to adapt its synapses in such a way that its input-output distribution approaches a given target distribution with density $p^*$. For a given input spike pattern $x$, at each point in time, the probability for a Poisson neuron to fire a spike during the interval $[t, t + dt]$ (denoted as $y_t = 1$) follows a Bernoulli distribution with a parameter $\phi_t dt = \phi(V_t) dt$, depending on the current membrane potential. The probability density of the binary variable $y_t$ on $\{0, 1\}$, describing whether or not a spike occurred in the interval $[t, t + dt]$, is therefore given by

$$p_w\left(y_t \mid x_t^\epsilon\right) = \left(\phi_t dt\right)^{y_t} \left(1 - \phi_t dt\right)^{(1-y_t)} , \tag{49}$$

and we have

$$p_w\left(y_t, x_t^\epsilon\right) = \left(\phi_t dt\right)^{y_t} \left(1 - \phi_t dt\right)^{(1-y_t)} p_{\mathrm{usp}}\left(x_t^\epsilon\right) , \tag{50}$$

where $p_{\mathrm{usp}}$ denotes the probability density of the unweighted synaptic potentials $x_t^\epsilon$. Measuring the distance to the target distribution in terms of the Kullback-Leibler divergence, we arrive at

$$C(w) = D_{\mathrm{KL}}\left(p^* \| p_w\right) = \mathbb{E}\left[\log\left[\frac{p^*\left(y_t, x_t^\epsilon\right)}{p_w\left(y_t, x_t^\epsilon\right)}\right]\right]_{p^*} = \mathbb{E}\left[\log\left[\frac{p^*\left(y_t \mid x_t^\epsilon\right)}{p_w\left(y_t \mid x_t^\epsilon\right)}\right]\right]_{p^*} . \tag{51}$$

Since the target distribution does not depend on the synaptic weights, the negative Euclidean gradient of the $D_{\mathrm{KL}}$ equals

$$-\nabla_w^{\mathrm{E}} C = -\frac{\partial C}{\partial w} = \mathbb{E}\left[\frac{\partial}{\partial w}\log\left[p_w\left(y_t \mid x_t^\epsilon\right)\right]\right]_{p^*} . \tag{52}$$

We may then calculate

$$\frac{\partial}{\partial w}\log p_w\left(y_t \mid x_t^\epsilon\right) = \frac{\partial}{\partial w}\left[y_t \log\left(\phi_t dt\right) + \left(1 - y_t\right)\log\left(1 - \phi_t dt\right)\right] \tag{53}$$

$$= \left(\frac{y_t}{\phi_t dt} - \frac{1 - y_t}{1 - \phi_t dt}\right)\phi_t' dt x_t^\epsilon \tag{54}$$

$$= \left(y_t - \phi_t dt\right)\frac{\phi_t' dt}{\phi_t dt\left(1 - \phi_t dt\right)}x_t^\epsilon \tag{55}$$

$$\approx \left(y_t - \phi_t dt\right)\frac{\phi_t'}{\phi_t}x_t^\epsilon , \tag{56}$$

where *Equation 55* follows from the fact that $y_t \in \{0, 1\}$ and for *Equation 56* we neglected the term of order $dt^2$ which is small compared to the remainder. Plugging *Equation 56* into *Equation 52* leads to the Euclidean-gradient descent online learning rule, given by

$$\dot{w}_e = \left[Y_t^* - \phi(V_t)\right]\frac{\phi_t'}{\phi_t}x_t^\epsilon . \tag{57}$$

Here,

$$Y_t^* = \sum_f \delta\left(t - t^f\right) \tag{58}$$

is the teacher spike train. We obtain the negative natural gradient by multiplying *Equation 57* with the inverse Fisher information matrix, since

$$\nabla_w^{\mathrm{N}} C = G\left(w\right)^{-1} \nabla_w^{\mathrm{E}} C , \tag{59}$$

with the Fisher information matrix $G(w)$ at $w$ being defined as

$$G(w) = \mathbb{E}\left[\frac{\partial \log p_w}{\partial w} \frac{\partial \log p_w}{\partial w}^T\right]_{p_w}. \tag{60}$$

Exploiting that, just like for the target density, the density of the USPs does not depend on $w$, so

$$\frac{\partial}{\partial w} \log p_w(y_t, x_t^\epsilon) = \frac{\partial}{\partial w}\left[\log p_w(y_t | x_t^\epsilon) + \log p_{\text{usp}}(x_t^\epsilon)\right] \tag{61}$$

$$= \frac{\partial}{\partial w} \log p_w(y_t | x_t^\epsilon), \tag{62}$$

we can insert the previously derived formula *Equation 56* for the partial derivative of the log-likelihood. Hence, using the tower property for expectation values and the definition of $p_w$ (*Equation 49*, *Equation 50*), *Equation 60* transforms to

$$G(w) = \mathbb{E}\left[(y_t - \phi_t \mathrm{d}t)^2 \frac{\phi_t'^2}{\phi_t^2} x_t^\epsilon x_t^{\epsilon T}\right]_{p_w} \tag{63}$$

$$= \mathbb{E}\left[\mathbb{E}\left[(y_t - \phi_t \mathrm{d}t)^2 \frac{\phi_t'^2}{\phi_t^2}\right]_{p_w(y|x_t^\epsilon)} x_t^\epsilon x_t^{\epsilon T}\right]_{p_{\text{usp}}} \tag{64}$$

$$= \mathbb{E}\left[\left(\phi_t \mathrm{d}t \left(1 - \phi_t \mathrm{d}t\right)^2 + \left(1 - \phi_t \mathrm{d}t\right)\left(\phi_t \mathrm{d}t\right)^2\right) \frac{\phi_t'^2}{\phi_t^2} x_t^\epsilon x_t^{\epsilon T}\right]_{p_{\text{usp}}} \tag{65}$$

$$= \mathbb{E}\left[\left(\phi_t \mathrm{d}t - \left(\phi_t \mathrm{d}t\right)^2\right) \frac{\phi_t'^2}{\phi_t^2} x_t^\epsilon x_t^{\epsilon T}\right]_{p_{\text{usp}}} \tag{66}$$

$$\approx \mathbb{E}\left[\mathrm{d}t \frac{\phi_t'^2}{\phi_t} x_t^\epsilon x_t^{\epsilon T}\right]_{p_{\text{usp}}}. \tag{67}$$

In order to arrive at an explicit expression for the natural-gradient learning rule, we further decompose the Fisher information matrix, which will then enable us to find a closed expression for its inverse.

Inspired by the approach in *Amari, 1998*, we exploit the fact that a positive semi-definite matrix is uniquely defined by its values as a bivariate form on any basis of $\mathbb{R}^n$. Choosing a basis for which the bilinear products with $G(w)$ are of a particularly simple form, we are able to decompose the Fisher Information Matrix by constructing a sum of matrices whose values as a bivariate form on the basis equal are equal to those of $G(w)$. Due to the structure of this particular decomposition, we may then apply well-known formulas for matrix inversion to obtain $G(w)^{-1}$.

Consider the basis $\mathcal{B} = \{w, b_1, \ldots, b_{n-1}\}$ such that the vectors $\sqrt{\Sigma_{\text{usp}}} w, \sqrt{\Sigma_{\text{usp}}} b_1, \ldots, \sqrt{\Sigma_{\text{usp}}} b_{n-1}$ are orthogonal to each other. Here, $\Sigma_{\text{usp}}$ denotes the covariance matrix of the USPs which in the case of independent Poisson input spike trains is given as

$$\Sigma_{\text{usp}} = \text{diag}(c_\epsilon^{-1} r_i). \tag{68}$$

In this case, the matrix square root reduces to the component-wise square root. Note that for any $b, b' \in \mathcal{B}$ with $b \neq b'$, the random variables $b^T x_t^\epsilon$ and $b'^T x_t^\epsilon$ are uncorrelated, since

$$\text{Cov}\left(b^T x_t^\epsilon, b'^T x_t^\epsilon\right) = \text{Cov}\left(\sum_{j=1}^n b_j x_{t,j}^\epsilon, \sum_{k=1}^n b'_k x_{t,k}^\epsilon\right) \tag{69}$$

$$= \sum_{j=1}^n \sum_{k=1}^n b_j b'_k \text{Cov}\left(x_{t,j}^\epsilon, x_{t,k}^\epsilon\right) \tag{70}$$

$$= b^T \Sigma_{\text{usp}} b' = 0. \tag{71}$$

We make the mild assumptions of having small afferent populations firing at the same input rate, and that the basis $\mathcal{B}$ is constructed in such way that the basis vectors are not too close to the coordinate axes, such that the products $\boldsymbol{b}^T \boldsymbol{x}_t^\epsilon$ are not dominated by a single component. Then, for sufficiently large $n$, every linear combination of the random variables $\boldsymbol{b}^T \boldsymbol{x}_t^\epsilon$ and $\boldsymbol{b}'^T \boldsymbol{x}_t^\epsilon$ is approximately normally distributed, thus, the two random variables follow a joint bivariate normal distribution. Furthermore, uncorrelated random variables that are jointly normally distributed are independent. Since functions of independent random variables are also independent, this allows us to calculate all products of the form $\boldsymbol{b}^T G(\boldsymbol{w}) \boldsymbol{b}'$ for $\boldsymbol{b}, \boldsymbol{b}' \in \mathcal{B} \setminus \{\boldsymbol{w}\}$, as we can transform the expectation of products

$$\boldsymbol{b}^T G(\boldsymbol{w}) \boldsymbol{b}' = \mathbb{E}\left[ \mathrm{d}t \frac{\phi'^2[V(t)]}{\phi[V(t)]} \left( \boldsymbol{b}^T \boldsymbol{x}_t^\epsilon \right) \left( \boldsymbol{x}_t^{\epsilon T} \boldsymbol{b} \right) \right]_{p_{\text{usp}}} \tag{72}$$

into products of expectations. Taking into account that $V(t) = \boldsymbol{w}^T \boldsymbol{x}_t^\epsilon$ and $\mathbb{E}[\boldsymbol{x}^\epsilon] = \boldsymbol{r}$, we arrive at

$$\boldsymbol{b}^T G(\boldsymbol{w}) \boldsymbol{b}' = I_1(\boldsymbol{w}) \left( \epsilon_0 \boldsymbol{r}^T \boldsymbol{b} \right) \left( \epsilon_0 \boldsymbol{r}^T \boldsymbol{b}' \right), \text{ for } \boldsymbol{b} \neq \boldsymbol{b}' \text{ and } \boldsymbol{b}, \boldsymbol{b}' \neq \boldsymbol{w}, \tag{73}$$

$$\boldsymbol{b}^T G(\boldsymbol{w}) \boldsymbol{b} = I_1(\boldsymbol{w}) \left( \boldsymbol{b}^T \Sigma_{\text{usp}} \boldsymbol{b} + \left( \epsilon_0 \boldsymbol{r}^T \boldsymbol{b} \right)^2 \right), \text{ for } \boldsymbol{b} \neq \boldsymbol{w}, \tag{74}$$

$$\boldsymbol{w}^T G(\boldsymbol{w}) \boldsymbol{b} = \boldsymbol{b}^T G(\boldsymbol{w}) \boldsymbol{w} = I_2(\boldsymbol{w}) \left( \epsilon_0 \boldsymbol{r}^T \boldsymbol{b} \right), \text{ for } \boldsymbol{b} \neq \boldsymbol{w}, \tag{75}$$

$$\boldsymbol{w}^T G(\boldsymbol{w}) \boldsymbol{w} = I_3(\boldsymbol{w}). \tag{76}$$

Here, $I_1, I_2, I_3$ denote the generalized voltage moments given as

$$I_1(\boldsymbol{w}) = \mathbb{E}\left[ \frac{\phi'^2_t}{\phi_t} \right]_{p_{\text{usp}}} = \frac{1}{\sqrt{2\pi\sigma_v^2}} \int_{-\infty}^{\infty} \frac{\phi'(u)^2}{\phi(u)} \exp\frac{-(u-\mu_v)^2}{2\sigma_v^2} du, \tag{77}$$

$$I_2(\boldsymbol{w}) = \mathbb{E}\left[ \frac{\phi'^2_t}{\phi_t} V_t \right]_{p_{\text{usp}}} = \frac{1}{\sqrt{2\pi\sigma_v^2}} \int_{-\infty}^{\infty} \frac{\phi'(u)^2}{\phi(u)} u \exp\frac{-(u-\mu_v)^2}{2\sigma_v^2} du, \tag{78}$$

$$I_3(\boldsymbol{w}) = \mathbb{E}\left[ \frac{\phi'^2_t}{\phi_t} V_t^2 \right]_{p_{\text{usp}}} = \frac{1}{\sqrt{2\pi\sigma_v^2}} \int_{-\infty}^{\infty} \frac{\phi'(u)^2}{\phi(u)} u^2 \exp\frac{-(u-\mu_v)^2}{2\sigma_v^2} du. \tag{79}$$

The integral formulas follow from the fact that for a large number of input afferents, and under the mild assumption of all synaptic weights roughly being of the same order of magnitude, the membrane potential approximately follows a normal distribution with mean $\mu_v = \epsilon_0 \sum_{i=1}^n w_i r_i$ and variance $\sigma_v^2 = \frac{1}{c_\epsilon} \sum_{i=1}^n w_i^2 r_i$. Here, $\epsilon_0 = 1$ mV ms and $c_\epsilon$ is given by **Equation 31**, see Sec. 'Neuron model'.

Based on the above calculations, we construct a candidate $\tilde{G}(\boldsymbol{w})$ for a decomposition of $G(\boldsymbol{w})$. We start with the matrix $c_1 \left( \epsilon_0^2 \boldsymbol{r}\boldsymbol{r}^T + \Sigma_{\text{usp}} \right)$, since its easy to see that

$$\boldsymbol{b}^T c_1 \left( \epsilon_0^2 \boldsymbol{r}\boldsymbol{r}^T + \Sigma_{\text{usp}} \right) \boldsymbol{b}' = \boldsymbol{b}^T G(\boldsymbol{w}) \boldsymbol{b}' \text{ for all } b, b' \in \mathcal{B}. \tag{80}$$

Exploiting the orthogonal properties according to which we constructed $\mathcal{B}$ to carefully add more terms such that also the other identities in **Equation 73** hold, we arrive at

$$\tilde{G}(\boldsymbol{w}) = c_1 \left( \epsilon_0^2 \boldsymbol{r}\boldsymbol{r}^T + \Sigma_{\text{usp}} \right) + c_2 \epsilon_0 \left[ \Sigma_{\text{usp}}\boldsymbol{w}\boldsymbol{r}^T + \boldsymbol{r} \left( \Sigma_{\text{usp}}\boldsymbol{w} \right)^T \right] + c_3 \left( \Sigma_{\text{usp}}\boldsymbol{w} \right) \left( \Sigma_{\text{usp}}\boldsymbol{w} \right)^T. \tag{81}$$

Here,

$$c_1 = I_1, \quad c_2 = \frac{I_2 - I_1 \mu_v}{\sigma_v^2}, \quad c_3 = \frac{I_3 - I_1 \left( \mu_v^2 + \sigma_v^2 \right) - 2c_2 \mu_v \sigma_v^2}{\sigma_v^4}. \tag{82}$$

To check that indeed $\tilde{G}(\boldsymbol{w}) = G(\boldsymbol{w})$, it suffices to check the values of $\tilde{G}$ as a bilinear form on the basis $\mathcal{B}$.

## Inverse of the Fisher information matrix

As the expectation of the outer product of a vector with itself, $G(w)$ is per construction symmetric and positive semidefinite. From the previous calculations, it follows that for elements $b$ of a basis $\mathcal{B}$ of $\mathbb{R}^n$ the products $b^T G(w) b$ are strictly positive. Hence $G(w)$ is positive definite and thus invertible. We showed that,

$$G(w) = c_1 \left( \epsilon_0^2 r r^T + \Sigma_{\text{usp}} \right) + c_2 \epsilon_0 \left[ \Sigma_{\text{usp}} w r^T + r \left( \Sigma_{\text{usp}} w \right)^T \right] + c_3 \left( \Sigma_{\text{usp}} w \right) \left( \Sigma_{\text{usp}} w \right)^T .$$

We introduce the notation.

$$\tilde{w} = \Sigma_{\text{usp}} w \text{ and } \tilde{r} = \epsilon_0 \Sigma_{\text{usp}}^{-1} r . \tag{83}$$

Then,

$$G(w) = \underbrace{c_1 \Sigma_{\text{usp}}}_{M_1} + \underbrace{(c_1 \epsilon_0 r + c_2 \tilde{w}) \epsilon_0 r^T}_{M_2} + \underbrace{(c_2 \epsilon_0 r + c_3 \tilde{w}) \tilde{w}^T}_{M_3} . \tag{84}$$

For the following calculations, we will repeatedly use the identities

$$w^T \epsilon_0 r = \mu_{\text{v}} , \ w^T \Sigma_{\text{usp}} w = \sigma_{\text{v}}^2, \text{ and } \tilde{r}^T \epsilon_0 r = q . \tag{85}$$

By the Sherman-Morrison-Woodbury formula, the inverse of an invertible rank one correction $(A + uv^T)$ of an invertible matrix A is given by

$$\left( A + uv^T \right)^{-1} = A^{-1} - \frac{A^{-1} uv^T A^{-1}}{1 + v^T A^{-1} u} . \tag{86}$$

Applying this to invert $G(w)$, as a first step, we consider the term $M_1 + M_2$ and identify $M_1 = A$ and $M_2 = uv^T$. Its inverse is given by

$$\left( M_1 + M_2 \right)^{-1} = M_1^{-1} - k_1 \Sigma_{\text{usp}}^{-1} \left( c_1 \epsilon_0 r + c_2 \tilde{w} \right) \epsilon_0 r^T \Sigma_{\text{usp}}^{-1} = M_1^{-1} - k_1 \left( c_1 \tilde{r} + c_2 w \right) \tilde{r}^T , \tag{87}$$

with

$$k_1 = \left\{ c_1^2 \left[ 1 + \epsilon_0 r^T M_1^{-1} \left( c_1 \tilde{r} + c_2 w \right) \right] \right\}^{-1} = c_1^{-1} \left[ c_1 (q + 1) + c_2 \mu_{\text{v}} \right]^{-1} . \tag{88}$$

Applying the Sherman-Morrison-Woodbury formula a second time, this time with $M_1 + M_2 = A$ and $M_3 = uv^T$, we obtain

$$\begin{aligned} G(w)^{-1} &= \left[ (M_1 + M_2) + M_3 \right]^{-1} \\ &= (M_1 + M_2)^{-1} - k_2 (M_1 + M_2)^{-1} M_3 (M_1 + M_2)^{-1} \\ &= M_1^{-1} - k_1 \left( c_1 \tilde{r} + c_2 w \right) \tilde{r}^T \\ &\quad - k_2 \left[ M_1^{-1} - k_1 \left( c_1 \tilde{r} + c_2 w \right) \tilde{r}^T \right] M_3 \left[ M_1^{-1} - k_1 \left( c_1 \tilde{r} + c_2 w \right) \tilde{r}^T \right] \\ &= M_1^{-1} - k_1 \left( c_1 \tilde{r} + c_2 w \right) \tilde{r}^T - k_2 M_1^{-1} M_3 M_1^{-1} \\ &\quad + k_1 k_2 M_1^{-1} M_3 \left( c_1 \tilde{r} + c_2 w \right) \tilde{r}^T + k_1 k_2 \left( c_1 \tilde{r} + c_2 w \right) \tilde{r}^T M_3 M_1^{-1} \\ &\quad + k_1^2 k_2 \left( c_1 \tilde{r} + c_2 w \right) \tilde{r}^T M_3 \left( c_1 \tilde{r} + c_2 w \right) \tilde{r}^T . \end{aligned} \tag{89}$$

Here,

$$k_2 = \left\{ 1 + \tilde{w}^T \left[ M_1^{-1} - k_1 \left( c_1 \tilde{r} + c_2 w \right) \tilde{r}^T \right] \left( c_2 \epsilon_0 r + c_3 \tilde{w} \right) \right\}^{-1} \tag{90}$$

$$= \left[ 1 + c_1^{-1} \left( c_2 \mu_{\text{v}} + c_3 \sigma_{\text{v}}^2 \right) - k_1 \left( c_1 \mu_{\text{v}} + c_2 \sigma_{\text{v}}^2 \right) \left( c_2 q + c_3 \mu_{\text{v}} \right) \right]^{-1} . \tag{91}$$

Plugging in the definitions of $M_3$ and $M_1$, and using that

$$M_1^{-1} M_3 = c_1^{-1} \left( c_2 \tilde{r} + c_3 w \right) \tilde{w}^T , \tag{92}$$

we arrive at

$$G\left(\boldsymbol{w}\right)^{-1} = M_1^{-1} - k_1\left(c_1\tilde{\boldsymbol{r}} + c_2\boldsymbol{w}\right)\tilde{\boldsymbol{r}}^T - k_2 c_1^{-1}\left(c_2\tilde{\boldsymbol{r}} + c_3\boldsymbol{w}\right)\tilde{\boldsymbol{w}}^T M_1^{-1} \tag{93}$$

$$+ k_1 k_2 c_1^{-1}\left(c_2\tilde{\boldsymbol{r}} + c_3\boldsymbol{w}\right)\tilde{\boldsymbol{w}}^T\left(c_1\tilde{\boldsymbol{r}} + c_2\boldsymbol{w}\right)\tilde{\boldsymbol{r}}^T + k_1 k_2\left(c_1\tilde{\boldsymbol{r}} + c_2\boldsymbol{w}\right)\tilde{\boldsymbol{r}}^T\left(c_2\epsilon_0\boldsymbol{r} + c_3\tilde{\boldsymbol{w}}\right)\tilde{\boldsymbol{w}}^T M_1^{-1} \tag{94}$$

$$- k_1^2 k_2\left(c_1\tilde{\boldsymbol{r}} + c_2\boldsymbol{w}\right)\tilde{\boldsymbol{r}}^T\left(c_2\epsilon_0\boldsymbol{r} + c_3\tilde{\boldsymbol{w}}\right)\tilde{\boldsymbol{w}}^T\left(c_1\tilde{\boldsymbol{r}} + c_2\boldsymbol{w}\right)\tilde{\boldsymbol{r}}^T \tag{95}$$

$$= c_1^{-1}\Sigma_{\text{usp}}^{-1} - k_1\left(c_1\tilde{\boldsymbol{r}} + c_2\boldsymbol{w}\right)\tilde{\boldsymbol{r}}^T \tag{96}$$

$$- k_2 c_1^{-2}\left(c_2\tilde{\boldsymbol{r}} + c_3\boldsymbol{w}\right)\boldsymbol{w}^T \tag{97}$$

$$+ k_1 k_2 c_1^{-1}\left(c_2\tilde{\boldsymbol{r}} + c_3\boldsymbol{w}\right)\left(c_1\mu_{\text{v}} + c_2\sigma_{\text{v}}^2\right)\tilde{\boldsymbol{r}}^T + k_1 k_2 c_1^{-1}\left(c_1\tilde{\boldsymbol{r}} + c_2\boldsymbol{w}\right)\left(c_2 q + c_3\mu_{\text{v}}\right)\boldsymbol{w}^T \tag{98}$$

$$- k_1^2 k_2\left(c_1\tilde{\boldsymbol{r}} + c_2\boldsymbol{w}\right)\left(c_1\mu_{\text{v}} + c_3\sigma_{\text{v}}^2\right)\left(c_2 q + c_3\mu_{\text{v}}\right)\tilde{\boldsymbol{r}}^T . \tag{99}$$

After resorting the terms and grouping, we obtain the inverse of the Fisher Information Matrix as

$$G\left(\boldsymbol{w}\right)^{-1} = \frac{1}{c_1}\left[\Sigma_{\text{usp}}^{-1} + \left(g_1\tilde{\boldsymbol{r}} + g_2\boldsymbol{w}\right)\tilde{\boldsymbol{r}}^T + \left(g_3\tilde{\boldsymbol{r}} + g_4\boldsymbol{w}\right)\boldsymbol{w}^T\right] \tag{100}$$

$$= \gamma_{\text{s}}\left[\Sigma_{\text{usp}}^{-1} + \left(c_\epsilon\epsilon_0 g_1 \mathbf{1} + g_2\boldsymbol{w}\right)c_\epsilon\epsilon_0 \mathbf{1}^T + \left(c_\epsilon\epsilon_0 g_3 \mathbf{1} + g_4\boldsymbol{w}\right)\boldsymbol{w}^T\right] , \tag{101}$$

with

$$g_1 = c_1\left\{-k_1 c_1 + k_1 k_2 c_1^{-1} c_2\left(c_1\mu_{\text{v}} + c_2\sigma_{\text{v}}^2\right) - k_1^2 k_2 c_1\left[\left(c_1\mu_{\text{v}} + c_2\sigma_{\text{v}}^2\right)\left(c_2 q + c_3\mu_{\text{v}}\right)\right]\right\} \tag{102}$$

$$g_2 = c_1\left\{-k_1 c_2 + c_1^{-1} k_1 k_2 c_3\left(c_1\mu_{\text{v}} + c_2\sigma_{\text{v}}^2\right) - k_1^2 k_2 c_2\left[\left(c_1\mu_{\text{v}} + c_2\sigma_{\text{v}}^2\right)\left(c_2 q + c_3\mu_{\text{v}}\right)\right]\right\} \tag{103}$$

$$g_3 = c_1\left[k_1 k_2\left(c_2 q + c_3\mu_{\text{v}}\right) - k_2 c_1^{-2} c_2\right] \tag{104}$$

$$g_4 = c_1\left[k_1 k_2 c_1^{-1} c_2\left(c_2 q + c_3\mu_{\text{v}}\right) - k_2 c_1^{-2} c_3\right] . \tag{105}$$

Note that instead of applying the Sherman-Morrison-Woodbury formula twice one could also directly use the version for rank-2 corrections, known as the Woodbury identity (**Max, 1950**). This can be seen by rewriting **Equation 84** as

$$G\left(\boldsymbol{w}\right) = c_1\Sigma_{\text{usp}} + UCU^T, \tag{106}$$

with

$$U = \begin{pmatrix} r_1 & \tilde{w}_1 \\ \vdots & \vdots \\ r_n & \tilde{w}_n \end{pmatrix}$$

and

$$C = \begin{pmatrix} c_1\epsilon_0^2 & c_2\epsilon_0 \\ c_2\epsilon_0 & c_3 \end{pmatrix} .$$

By the Woodbury identity for rank-2 corrections, we then have

$$G\left(\boldsymbol{w}\right)^{-1} = c_1^{-1}\Sigma_{\text{usp}}^{-1} - \tilde{U}\left(C^{-1} + U^T\tilde{U}\right)^{-1}\tilde{U}^T,$$

with $\tilde{U} = c_1^{-1}\Sigma_{\text{usp}}^{-1} U$.

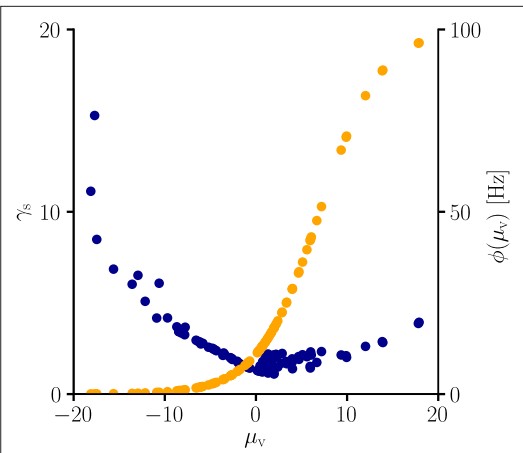

**Figure 10.** Global learning rate scaling $\gamma_s$ as a function of the mean membrane potential. We sampled the global learning rate factor $\gamma_s$ (blue) for various conditions. In line with **Equation 107**, $\gamma_s$ is boosted in regions where the transfer function is flat, that is, $\phi'(V)$ is small. The global scaling factor is additionally increased in regions where the transfer function reaches high absolute values.

## Analysis of the learning rule coefficients

In order to gain an intuitive understanding of **Equation 13** and to judge its suitability as an in-vivo plasticity rule, we require insights into the behavior of the coefficients $\gamma_s, \gamma_u,$ and $\gamma_w$ under various circumstances.

## Global scaling factor

The global scaling factor $\gamma_s$ is given as

$$\gamma_s = c_1^{-1} = \left\{ \mathbb{E}\left[ \frac{\phi'^2_t}{\phi_t} \right]_{p_{\mathrm{usp}}} \right\}^{-1}.$$

(107)

The above formula reveals that $\gamma_s$ is closely tied to the firing nonlinearity of the neuron, as well as the statistics of output. The scaling by the inverse slope of the output nonlinearity amplifies the synaptic update in regions where a small change in weight and thus in the membrane potential would not lead to a noticeable change in output distribution. A further scaling with $\phi$ additionally amplifies the synaptic change in high-output regimes (**Figure 10**). This is in line with the spirit of natural gradient that ensures that the size of the synaptic weight update is homogeneous in terms of $D_{\mathrm{KL}}$ change rather than in absolute weight terms. Furthermore, the rescaling is based on the average output statistics, which the synapse might have access to via backpropagating action potentials (**Stuart et al., 1997**), rather than an instantaneous value.

## Empirical analysis of $\gamma_u$ and $\gamma_w$

While the formula for $\gamma_s$ provides a rather good intuition of this coefficients' behavior, from the derivations in the previous sections, it becomes clear that such a straightforward interpretation is not readily available from the formulas for $\gamma_u$ and $\gamma_w$. Defined as

$$\gamma_u = -c_\epsilon \epsilon_0 \left( c_\epsilon \epsilon_0 g_1 \sum_{i=1}^n x_i^\epsilon + g_3 V \right)$$

(108)

and

$$\gamma_w = c_\epsilon \epsilon_0 g_2 \sum_{i=1}^n x_i^\epsilon + g_4 V,$$

(109)

where $g_1, \ldots g_4$ are given in **Equation 102**, these coefficients depend, apart from the membrane potential and its first and second moments, on the total input and its mean the total rate. This raises the question whether the synapse, which in general only has limited access to global quantities, can implement $\gamma_u$ and $\gamma_w$. We therefore used an empirical analysis through simulations to obtain a more detailed insight.

As a starting point, we sampled $g_1, \ldots, g_4$ for various input conditions (**Figure 11A–D**, refer to Sec. 'Approximation of learning-rule coefficents' for simulation details). Here, we varied the afferent input rate $r$ between 5 Hz and 55 Hz for $n = 100$ neurons and evaluated the value of the respective coefficient for a randomly sampled input weight. In a second simulation (**Figure 11E–H**, Sec. 'Approximation of learning-rule coefficents'), we varied the number $n$ of afferent inputs between 10 and 200 neurons for fixed input rate 20 Hz, again with randomly chosen input weights. This revealed an approximately inverse proportional relationship between the average input rate $r$ and $g_1, g_2$ and $g_3$ respectively. Furthermore, these coefficients seemed to be approximately inversely proportional to the number $n$ of afferent inputs. However, this was not true for $g_4$, whose mean seemed to stay approximately

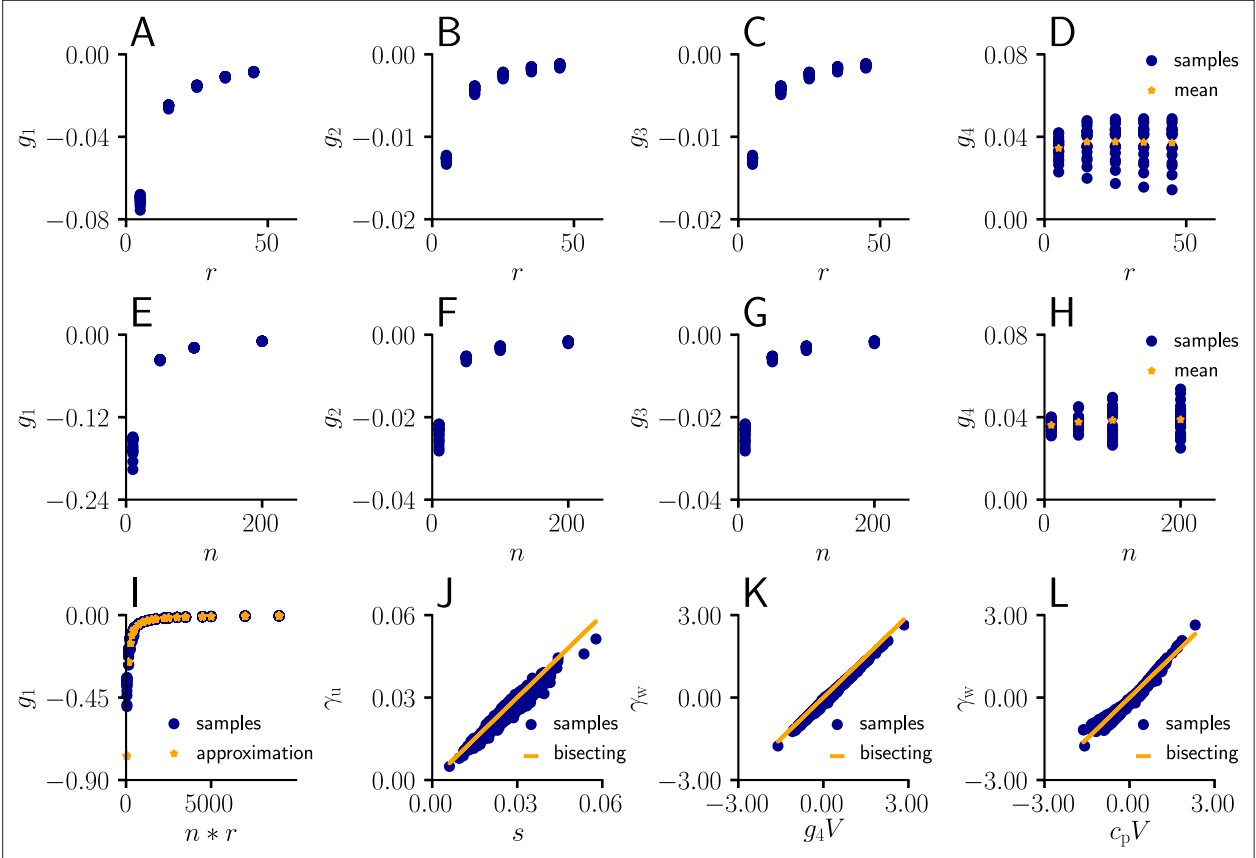

**Figure 11.** Learning rule coefficients can be approximated by simpler quantities. (**A**)-(**D**) Samples values for $g_1, \ldots, g_4$ for different afferent input rates. (**E**)-(**H**) In a second simulation, we varied the number $n$ of afferent inputs. (**I**) Comparison of the sampled values of $g_1$ (blue) as a function of the total input rate $n * r$ to the values of the approximation given by $g_1 \approx -q^{-1}$. (**J**) Sampled values of $\gamma_{\mathrm{u}}$ (blue) as a function of the approximation s (**Equation 111**). The proximity of the sampled values to the diagonal indicates that $s$ may indeed serve as an approximation for $\gamma_{\mathrm{u}}$. (**K**) Sampled values of $\gamma_{\mathrm{w}}$ (blue) as a function of $g_4 V$. The proximity of the sampled values to the diagonal indicates that $g_4 V$ serves as an approximation for $\gamma_{\mathrm{w}}$. (**L**) Same as (**K**), but with $g_4$ replaced by a constant $c_w = 0.05$.

constants across input rates, although the scattering across the mean value (for different weight samples) increased.

While the value of the membrane potential stays bounded also for a large number of inputs (due to the normalization of the synaptic weight range with $\frac{1}{n}$), the total sum of USPs increases with an increasing number of inputs. Therefore, for large enough $n$, the term $g_3 V$ can be neglected, so

$$\gamma_{\mathrm{u}} \approx c_\epsilon^2 \epsilon_0^2 g_1 \sum_{i=1}^n x_i^\epsilon. \tag{110}$$

A closer look at the behavior of $g_1$ shows that, for a sufficiently large number of neurons or high input rates, $g_1 \approx -q^{-1}$, which we verified by simulations (**Figure 11I**, Sec. 'Approximation of learning-rule coefficents'). In consequence,

$$\gamma_{\mathrm{u}} \approx s = \frac{c_\epsilon \sum_{i=1}^n x_i^\epsilon}{\sum_{i=1}^n r_i} . \tag{111}$$

To verify this approximation, we sampled the values of $\gamma_{\mathrm{u}}$ for various conditions (**Figure 11J**, Sec. 'Approximation of learning-rule coefficents') and compared them against the approximation in **Equation 111**, which confirmed our approximation. Since the input rate is the mean value of the USP, assuming large enough populations with the same input rate and a sufficient number of input afferents, by the central limit theorem we have

$$\gamma_{\mathrm{u}} \longrightarrow c_{\mathrm{u}} c_\epsilon \text{ for } n \to \infty , \tag{112}$$

with $c_u = \epsilon_0$. However, in practice, for $\epsilon_0 = 1\,\mathrm{mV\,ms}$, a learning behavior much closer to natural gradient was obtained when $c_u$ was slightly smaller than 1 such as $c_u = 0.95$ (Sec. 'Quadratic transfer function').

As a starting point to approximate $\gamma_w$, we noticed that the mean of $g_4$ stayed approximately constant when varying the input rate or the number of input afferents. On the other hand, $g_2$ rapidly tends to zero in those cases, so we assumed that $g_2 \sum_{i=1}^{n} x_i^\epsilon$ stays either constant or goes to zero in the limit of large n. Since $c_\epsilon \epsilon_0 g_2$ seemed to be rather small compared to $g_4$, we hypothesized $\gamma_w \approx g_4 V$, which was confirmed by simulations (*Figure 11K*, Sec. 'Approximation of learning-rule coefficents'). As a second step (*Figure 11L*, Sec. 'Approximation of learning-rule coefficents'), since $g_4$ seemed to be constant in the mean, we approximated

$$\gamma_w \approx c_w V,\qquad(113)$$

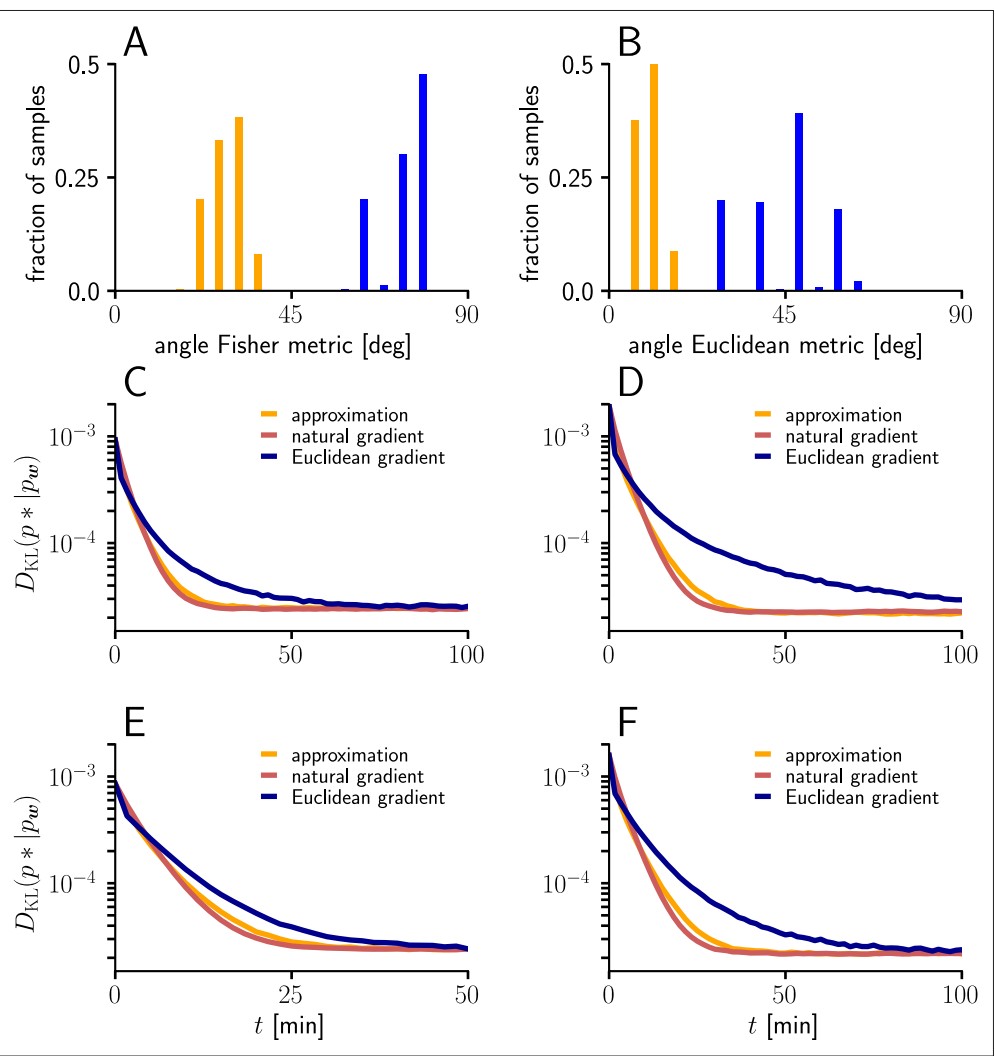

**Figure 12.** Natural-gradient learning can be approximated by a simpler rule in many scenarios. (**A**) Mean Fisher angles between true and approximated weight updates (orange) and between natural and Euclidean weight updates (blue), for $n = 100$. Results for several input patterns were pooled (group1/group2: 10 Hz/10 Hz 10 Hz/30 Hz, 10 Hz/50 Hz, 20 Hz/20 Hz, 20 Hz/40 Hz). Initial weights and input spikes were sampled randomly (100 randomly sampled initial weight vectors per input pattern; for each, angles were averaged over 100 input spike train samples per afferent). (**B**) Same as (**A**), but angles measured in the Euclidean metric. (**C–F**) Comparison of learning curves for natural gradient (red), Euclidean gradient (blue) and approximation (orange) for $n = 100$ afferents. Simulations were performed in the setting of *Figure 3*, under multiple input conditions. (**C**) Group one firing with 10 Hz, group two firing at 30 Hz. (**D**) Group one firing with 10 Hz, group two firing at 50 Hz. (**E**) Group one firing with 20 Hz, group two firing at 20 Hz. (**F**) Group one firing with 20 Hz, group two firing at 40 Hz.

where simulations with $c_w = 0.05$, close to the mean of $g_4$ showed a learning behavior close to natural-gradient learning (*Figure 12C–F*). Replacing $\gamma_u$ and $\gamma_w$ in *Equation 13* by the expressions in *Equation 112* and *Equation 113*, we obtain the approximated natural-gradient rule

$$\dot{\boldsymbol{w}}_a = \eta \, \gamma_s \left[ Y^* - \phi(V) \right] \frac{\phi'(V)}{\phi(V)} \boldsymbol{f}' \left( \boldsymbol{w} \right)^{-1} \left[ \frac{c_\epsilon \boldsymbol{x}^\epsilon}{\boldsymbol{r}} - c_\epsilon c_u + c_w V \boldsymbol{f} \left( \boldsymbol{w} \right) \right] \; . \tag{114}$$

## Performance of the approximated learning rule

Simulations of natural, Euclidean and approximated natural-gradient weight updates for several input patterns and randomly sampled initial conditions (Sec. 'Evaluation of the approximated natural-gradient rule') showed that the average angles (both in the Euclidean metric and in the Fisher metric) between the true and approximated natural-gradient weight update were small compared to the average angle between Euclidean and natural-gradient weight update (*Figure 12A–B*). This was confirmed by the learning curves for several tested input conditions in the setting of *Figure 3*, since the performance of the approximation lay in between the natural and the Euclidean gradient's performance (*Figure 12C–F*, simulation details: Sec. 'Evaluation of the approximated natural-gradient rule'). It can hence be regarded as a trade-off between optimal learning speed, parameter invariance and biological implementability. Note that plugging the above calculations into *Equation 13* still keeps invariance under coordinate-wise smooth parameter changes.

## **Quadratic transfer function**

While for all our simulations, we used the sigmoidal transfer function given in *Equation 24*, the derivations that lead to *Equation 100* also hold for other choices of transfer function. A particularly simple and thereby instructive result can be obtained for a transfer function $\phi$ that satisfies

$$\frac{(\phi')^2}{\phi} = 1 \; . \tag{115}$$

This is the case for a rectified quadratic transfer function

$$\phi \left( V \right) = \tfrac{1}{4} \left( V - \theta \right)^2 \Theta(V - \theta) \; , \tag{116}$$

where $\Theta$ is the Heaviside step function. To prevent issues with division by zero and to ensure that *Equation 115* holds for all relevant membrane voltages $V$, we assume $\theta \ll \mu_v - \sigma_v$. Then, *Equation 67* reduces to

$$G \left( \boldsymbol{w} \right) \approx \mathbb{E} \left( \mathrm{d} t \boldsymbol{x}_t^\epsilon \boldsymbol{x}_t^{\epsilon T} \right)_{p_{\mathrm{usp}}} \; , \tag{117}$$

where the right side no longer depends on $\boldsymbol{w}$, thus yielding $\gamma_w = 0$. Furthermore, we have

$$I_1 = 1 \; , \tag{118}$$
$$I_2 = \mu_v \; , \tag{119}$$
$$I_3 = \mu_v^2 + \sigma_v^2 \; , \tag{120}$$

and therefore $c_2 = c_3 = 0$ (see *Equation 82*). Plugging this into *Equation 81*, we obtain

$$G \left( \boldsymbol{w} \right) = \epsilon_0^2 \, \boldsymbol{r} \boldsymbol{r}^T + \Sigma_{\mathrm{usp}} \; , \tag{121}$$

with $\Sigma_{\mathrm{usp}}$ defined in *Equation 35*. Inverting this using the Sherman-Morrison-Woodbury Formula, we arrive at

$$G \left( \boldsymbol{w} \right)^{-1} = \Sigma_{\mathrm{usp}}^{-1} - \frac{c_\epsilon^2 \epsilon_0^2}{q+1} \boldsymbol{1} \boldsymbol{1}^T \; . \tag{122}$$

Inserting this version of the Fisher information matrix into *Equation 12*, we get $\gamma_s = 1$ and $\gamma_w = 0$ (see *Equation 107 and 109*), thus obtaining a simplified version of the natural-gradient learning rule as

$$\dot{\boldsymbol{w}} = \eta \left[ Y^* - \phi(V) \right] \frac{1}{\phi'(V)} \boldsymbol{f}' \left( \boldsymbol{w} \right)^{-1} \left( \frac{c_\epsilon \boldsymbol{x}^\epsilon}{\boldsymbol{r}} - \gamma_u \mathbf{1} \right) , \tag{123}$$

$$\gamma_u = \frac{c_\epsilon^2 \epsilon_0^2 \sum_i x_i^\epsilon}{(q+1)} = c_\epsilon \frac{c_\epsilon \epsilon_0^2 \sum_i x_i^\epsilon}{(c_\epsilon \epsilon_0^2 \sum_i r_i + 1)} \tag{124}$$

with $q$ as in *Equation 36*. This is in line with our empirical approximation for $\gamma_u$ for the case of the sigmoidal transfer function (*Equation 111*). It is interesting to note that, for a large number of inputs, we have $\gamma_u \to c_\epsilon$ and the average of the parenthesis in *Equation 123*, taken in isolation, reduces to zero. However, this does not mean that, on average, there is no plasticity. Instead, we can say that, for a quadratic activation regime, the natural gradient is purely driven by the correlation between the postsynaptic error and the (normalized and centered) presynaptic input.

## Continuous-time limit

Under the Poisson-process assumption, firing a spike in one of the time bins is independent from the spikes in other bins, therefore the probability for firing in two disjunct time intervals $[t_1 + dt]$ and $[t_2 + dt]$ is given as

$$p_{\boldsymbol{w}} \left( y_{t_1}, y_{t_2} \big| \boldsymbol{x}_{t_1}^\epsilon, \boldsymbol{x}_{t_1}^\epsilon \right) = p_{\boldsymbol{w}} \left( y_{t_1} \big| \boldsymbol{x}_{t_1}^\epsilon \right) p_{\boldsymbol{w}} \left( y_{t_2} \big| \boldsymbol{x}_{t_1}^\epsilon \right) . \tag{125}$$

Since $\mathbb{E} \left[ \frac{\partial \log p_{\boldsymbol{w}}}{\partial \boldsymbol{w}} \right]_{p_{\boldsymbol{w}}} = 0$, we have

$$\mathbb{E} \left[ \frac{\partial \log p_{\boldsymbol{w}} \left( y_{t_1}, y_{t_2} \big| \boldsymbol{x}_{t_1}^\epsilon, \boldsymbol{x}_{t_2}^\epsilon \right)}{\partial \boldsymbol{w}} \frac{\partial \log p_{\boldsymbol{w}} \left( y_{t_1}, y_{t_2} \big| \boldsymbol{x}_{t_1}^\epsilon, \boldsymbol{x}_{t_2}^\epsilon \right)^T}{\partial \boldsymbol{w}} \right]_{p_{\boldsymbol{w}}} \tag{126}$$

$$= \mathbb{E} \left[ \frac{\partial \log p_{\boldsymbol{w}} \left( y_{t_1} \big| \boldsymbol{x}_{t_1}^\epsilon \right)}{\partial \boldsymbol{w}} \frac{\partial \log p_{\boldsymbol{w}} \left( y_{t_1} \big| \boldsymbol{x}_{t_1}^\epsilon \right)^T}{\partial \boldsymbol{w}} \right]_{p_{\boldsymbol{w}}} + \mathbb{E} \left[ \frac{\partial \log p_{\boldsymbol{w}} \left( y_{t_2} \big| \boldsymbol{x}_{t_2}^\epsilon \right)}{\partial \boldsymbol{w}} \frac{\partial \log p_{\boldsymbol{w}} \left( y_{t_2} \big| \boldsymbol{x}_{t_2}^\epsilon \right)^T}{\partial \boldsymbol{w}} \right]_{p_{\boldsymbol{w}}} , \tag{127}$$

so the Fisher information matrix is additive.
In the continuous-time limit, where the interval $[0, T]$ is decomposed as $[0, T] = \cup_{i=0}^{k-1} [t_i, t_{i+1}]$, with $t_0 = 0, t_k = T$ and $k \longrightarrow \infty$, we have $dt = \frac{T}{k} \longrightarrow 0$. Therefore,

$$p_{\boldsymbol{w}} \left( y_{t_0}, \ldots, y_{t_{k-1}}, \boldsymbol{x}_{t_0}^\epsilon, \ldots, \boldsymbol{x}_{t_k}^\epsilon \right) = \Pi_{i=0}^{k-1} p_{\boldsymbol{w}} \left( y_i, \boldsymbol{x}_{t_i}^\epsilon \right) . \tag{128}$$

Then, under the assumption that $T$ is small and firing rates are approximately constant on $[0, T]$, for the Fisher information matrix, we have

$$G_{[0,T]} \left( \boldsymbol{w} \right) = \sum_{i=0}^{k-1} G_{t_i} \left( \boldsymbol{w} \right) \approx k \frac{T}{k} \mathbb{E} \left[ \frac{\phi_{t_0}'^2}{\phi_{t_0}} \boldsymbol{x}_{t_0}^\epsilon \boldsymbol{x}_{t_0}^{\epsilon \, T} \right]_{p_{usp}} \tag{129}$$

$$= T \, \mathbb{E} \left[ \frac{\phi_{t_0}'^2}{\phi_{t_0}} \boldsymbol{x}_{t_0}^\epsilon \boldsymbol{x}_{t_0}^{\epsilon \, T} \right]_{p_{usp}} . \tag{130}$$

## Acknowledgements

This work has received funding from the European Union 7th Framework Programme under grant agreement 604102 (HBP), the Horizon 2020 Framework Programme under grant agreements 720270, 785907 and 945539 (HBP) and the SNF under the personal grant 310030L_156863 (WS) and the

Sinergia grant CRSII5_180316. Calculations were performed on UBELIX (http://www.id.unibe.ch/hpc), the HPC cluster at the University of Bern. We thank Oliver Breitwieser for the figure template, and Simone Surace and João Sacramento, as well as the other members of the Senn and Petrovici groups for many insightful discussions. We are also deeply grateful to the late Robert Urbanczik for many enlightening conversations on gradient descent and its mathematical foundations. Furthermore, we wish to thank the Manfred Stärk Foundation for their ongoing support.

## Additional information

### Funding

| Funder | Grant reference number | Author |
|---|---|---|
| European Union 7th Framework Programme | 604102 | Walter Senn |
| Horizon 2020 Framework Programme | 720270 | Walter Senn<br>Mihai A Petrovici |
| Horizon 2020 Framework Programme | 785907 and 945539 | Walter Senn<br>Mihai A Petrovici |
| Swiss National Fonds | personal grant 310030L_156863 (WS) | Walter Senn |
| Swiss National Fonds | Sinergia grant CRSII5_180316 | Walter Senn |
| Manfred Stärk Foundation | personal grant | Mihai A Petrovici |

The funders had no role in study design, data collection and interpretation, or the decision to submit the work for publication.

### Author contributions

Elena Kreutzer, Conceptualization, Data curation, Formal analysis, Investigation, Methodology, Software, Validation, Visualization, Writing – original draft, Writing – review and editing; Walter Senn, Conceptualization, Formal analysis, Funding acquisition, Investigation, Resources, Supervision, Visualization, Writing – original draft, Writing – review and editing; Mihai A Petrovici, Conceptualization, Formal analysis, Funding acquisition, Investigation, Methodology, Project administration, Resources, Supervision, Validation, Visualization, Writing – original draft, Writing – review and editing

### Author ORCIDs

Elena Kreutzer (iD) http://orcid.org/0000-0001-7975-7206
Walter Senn (iD) http://orcid.org/0000-0003-3622-0497
Mihai A Petrovici (iD) http://orcid.org/0000-0003-2632-0427

### Decision letter and Author response

Decision letter https://doi.org/10.7554/eLife.66526.sa1
Author response https://doi.org/10.7554/eLife.66526.sa2

## Additional files

### Supplementary files
• Transparent reporting form
• Source data 1. Source data for figures.

### Data availability
The data used to generate the figures and the simulation code are available on GitHub, (copy archived at swh:1:rev:4da5611348ea969aff461886f85842fe36a8571a).

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
