## [Editor Report]

The natural gradient has a long and rich history in machine learning. Here, the authors derive a biologically plausible implementation of natural-gradient-based plasticity for spiking neurons, which renders learning invariant under dendritic transformations. This new synaptic learning rule makes several very interesting experimental predictions with respect to the interplay of homo- and heterosynaptic plasticity, and with regard to the scaling of plasticity by the presynaptic variance.

---

## [Decision Letter]

**Decision letter after peer review:**

Thank you for submitting your article "Natural-gradient learning for spiking neurons" for consideration by *eLife*. Your article has been reviewed by 3 peer reviewers, one of whom is a member of our Board of Reviewing Editors, and the evaluation has been overseen by a Reviewing Editor and John Huguenard as the Senior Editor. The reviewers have opted to remain anonymous.

Essential revisions:

This paper derives a natural gradient learning rule for a spiking neuron in a supervised setting. Unlike conventional gradients in Euclidean space, the natural gradient is invariant under reparameterization, thus achieving fast convergence regardless of the position of synaptic contacts on the dendritic tree. The authors relate their rule to experimentally observed properties of synaptic plasticity, such as heterosynaptic regularization. We're not aware of any other work that applied natural gradient to a spiking neuron in a formal way, and we believe it is an important contribution.

That said, we do have several comments. All are relatively minor (many have to do with presentation), and, we hope, easy to address. Because this is a combination of three reviews, they're not exactly in order of appearance in the manuscript. But hopefully not too far off.

1. We're slightly concerned with the fact that the weight update scales inversely with the firing rate. First, presumably for small enough firing rates something in the derivation breaks down. Second, this makes a very strong prediction. The only data we know of that relates to this is in Aitchison and Latham, 2014 (now published in Nature Neuroscience vol 24, pgs 565-571, 2021), where they showed, in one experiment, that learning rate scales as 1/sqrt{r} for moderate firing rates (greater than about 1 Hz), and saturates at small firing rates. This is one experiment, so doesn't rule out the theory here, but these points should be discussed.

2. The link to biology is often very cursory. References to experiments are often in the form 'so and so found something a bit similar'. Wherever possible, you should try to make precision comparisons to experiments. And on page 10 you say that the weight dependent heterosynaptic terms explains the central weight distributions found in cortex. However, the weight dependent heterosynaptic term is just one term out of three, and the other terms are not weight-dependent. If you want to argue this, you should be more rigorous (and in particular take into account all three terms).

3. The conclusion of the Zenke and Gerstner paper is, we believe, incorrect: A learning rule with an intrinsic weight dependence, such as Oja's rule, does not need homeostasis. This clearly doesn't affect any of your analysis, but it does mean that the statement "…… (Zenke and Gerstner, 2017), where it was shown that pairing Hebbian terms with heterosynaptic and homeostatic plasticity is crucial for stability" is, we believe, not true.

4. Figure 1B is pretty incomprehensible, and it doesn't help that the equations are small and gray, making them hard to read. We would suggest dropping it, especially since the point is made very well in the text.

Along the same lines (but much later), in Sec. ‘The naive Euclidean gradient is not parametrization-invariant’ we don't see any contradiction between Equations 26 and Equation 27; they are just two different definitions of Δ w^s^. It might be an inconsistency, but not a contradiction.

5. Figure 3: Considering that several approximations were made in the derivation of the learning rule (even for Equation 7), learning performance should be evaluated a bit more carefully. In particular:

a. Please plot the distance between student weights and teacher weights, on the same timescale as in Figure 3F (either in Euclidean space or in Fisher metric space).

b. Please plot performance versus learning rate, so we can get a sense of robustness. "Performance" could either be asymptotic loss or time to achieve a particular loss. Also, can you speculate how the learning rate of a neuron can be optimized?

c. What's the target output firing rate? It might have been stated somewhere, but we missed it. It would be nice if it were in the figure caption, like it is in Figure 4.

d. In Figure 3, n=2 and n=100 are both used. It should be clear which panel is which.

e. Panels B and C: what's the timescale? There's only one labeled time point (0.1), so it's impossible to tell. And how many trials did it take to learn? We would have expected considerable learning during 2500 trials.

f. The direction of the gradient depends on whether or not there's a spike on the teacher neuron. Which means the path in weight space is noisy, and the weights should not converge -- instead, they should exhibit fluctuations forever. But the trajectories in D and F were very smoothed. Is that because they're averaged over a large number of trials? Or was some extra processing done? Please explain.

6. Figure 4:

a. Can you reproduce experimental results on dendritic position dependent plasticity (eg. Letzkus JJ, Kampa BM, Stuart GJ, 2006; Sjöström PJ, Häusser M, 2006). These experimental results are inconsistent with vanilla Euclidean gradient; hence they would provide support for biological relevance of the proposed learning rule.

b. "the distance from soma was varied between 1 microns to 10 microns". This seems small; the characteristic length of dendritic decay is in order of 100 microns (Williams and Stuart, 2002).

7. Figure 5: Here, you compared three neurons each receiving single excitatory input with different input characteristics. Do you expect the same result to be true for a single neuron receiving three excitatory inputs? We're curious about it because that's a more relevant scenario. Some casual remarks would be helpful.

8. Figure 7:

a. We're somewhat confused that you you can distinguish between the three forms of plasticity in Equations 11-13, at least for the stimulated synapses. Don't all three forms contribute at once? But in panel B you say, "Weight change of stimulated weights (homosynaptic)". And in panel C, you should have seen homosynaptic plasticity (see next comment). Could you explain how you can single out one form of plasticity? Or else drop the comment. Or maybe we misunderstood?

b. In the legend of Figure 7, it is mentioned that the rest of synapses received 0.01Hz input, but in Sec. ‘Comparison of homo- and heterosynaptic plasticity’, that information is omitted. Did you actually stimulate these synapses? If so, we would have expected a big jump in synaptic weight when there was a spike, and in 60 s there should have been 3 spikes (0.01x60x5).

c. 7G: A should be S?

d. Equation 13 is out of the blue. Could you tell us where this comes from? And from the bottom of page 10, we have

"…… is roughly a linear function of the membrane potential (more specifically, its deviation with respect to its baseline)."

What does baseline refer to? We didn't see anything about baseline in Equation 25, which is where Equation 13 comes from.

e. Could you tell us how the colors in panels B-D were computed? Presumably some assumptions were made. If so, what were they?

9. Equation 4: we believe that the right hand side should not have a minus sign. More importantly, we should be told that Y* consists of a sum of δ-functions; the statement "Here, Y* denotes a teacher spike train sampled from p*" was unhelpful. It's also important to tell us exactly how Y* is sampled, which is, presumably from

P(spike from Y* \in [t, t+dt]) = phi(sum_j_ w*_j_ x_j_^ε^) dt.

If that's not correct, then we're thoroughly confused.

10. The second paragraph on page 4 makes some strong statements, and we're not sure all of them are true. For instance,

"Convergence of learning is therefore harmed by the slow adaptation of distal synapses compared to equally important proximal counterparts."

Presumably, "adaptation" means "learning". If so, why is adaptation necessarily slow at distal synapses? That would seem to depend on the biological learning rule. If not, what does "adaptation" mean in this context?

And,

"With the multiplicative USP term x^ε^ in Equation 4 being the only manifestation of presynaptic activity, there is no mechanism by which to take into account input variability, which can, in turn, also impede learning."

This is a bit out of the blue. Why should input variability affect learning? So far the authors have not said anything about this. It comes up later, but right now it's pretty obscure.

11. Last paragraph on page 4: the motivation for using the Fisher information matrix is a bit obscure. (For instance, what does "it is generally the unique metric that remains invariant under sufficient statistics" mean?) We strongly suspect that we won't be only ones who are lost. So it would be good to motivate it in plain language. And in particular, it would be nice to show, at the very least, that using the Fisher information matrix automatically makes the learning rule invariant with respect to a change of variables. (Which is pretty easy to do.)

Along the same lines, after Equation 9 you say

"This represents an unmediated reflection of the philosophy of natural gradient descent, which finds the steepest path for a small change in output, rather than in the numeric value of some parameter."

This has been the theme all along, but was it ever shown? As far as we can tell, you just used the inverse of the Fisher information matrix. Is it obvious that it has the above effect?

12. It would be extremely useful to write down log p_w_ in the main text.

13. The quantities in Equation 7 should be defined immediately -- right now we have to read down half a page to figure out what they are. And you should tell us immediately how γ_u_ and γ_w_ could be computed locally.

14. In the section "Natural gradient speeds up learning", please tell us exactly what the cost function is (which, presumably, means telling us w* and the filters you use to convert incoming spike trains to x). And it would be good if the firing rates were given in the main text, so one doesn't have to look at the figure.

15. Because of the factor of phi'(V)/phi(V) in Equations 7 and 8, the effective scaling in Equation 9 is, approximately, 1/phi'(V). This doesn't change the point that a shallow slope implies a high learning rate, but thing's aren't quite as bad as Equation 9 implies. This seems worth mentioning.

16. You mention, on the top of page 8, "that the absence of dendritic democracy does not contradict the presence of democratic plasticity". It seems worth mentioning, just for completeness, that dendritic democracy can be achieved without democratic plasticity. In particular, as far as we can tell, the Euclidean gradient is also consistent with dendritic democracy, although convergence to the democratic state might be slower that for the natural gradient.

17. p. 10 "In comparison, input from weak synapses only has……" We couldn't get the logic of this and the following sentences. This non-trivial claim is not explicitly tested in the subsequent paragraph, but reappears in the discussion. You should expand on it, or remove it.

18. p22: "The integral formulas follow from……". It would help if you referred to Equations 20 and 21 (in Sec. ‘Neuron model’) here. We were confused by the sudden appearance of ε_o_ and c_ε_. Also, you should mention how they are related to dt.

19. It is somewhat confusing that you set up the problem about dendritic distance, but then first address, in Figure 3, the role of firing rate, before showing the solution to dendritic plasticity. Perhaps starting with Figure 4, contrasting standard gradients to natural gradients would help.

20. Page 7/8, you say "neurons are not symmetrical geometric objects". You should make it clear what this means (presumably it means that they have dendrites, and, therefore, some weights are attenuated). In addition, it's not clear that not being a symmetrical geometric objects implies a non-isotropic cost landscapes, since how isotropic the cost landscape is depends on the input and cost function as well as the attenuation in the dendrites. In principle they former could cancel the latter, producing an isotropic cost landscapes even with strongly attenuating neurons. It seems best to drop this point.

21. A number of studies have recently emphasized that in large networks most machine learning problems have highly degenerate minima (e.g. Belkin et al. et al. PNAS). Does the algorithm generalize to such situations? Your answer may be "we don't know". But whatever the answer, it would be worth mentioning in the paper.

22. Page 7, last paragraph: should the reference to Figure 5 be to Figure 4?

23. The abundance of superscripts, subscripts and decorations in the variables throughout the manuscript make it very hard to read. We would suggest simplifying notation as much as possible. In particular:

a. The superscript on the gradient operator doesn't help, as it requires extra memorization -- which is hard because it's not standard. You would be much better off putting the superscript on the cost function, and not redefining the gradient. It's especially confusing in Equation 6, since at first glance the superscript n means take n derivatives.

b. There are two w's: w^d^ and w_s_. Except sometimes there's a w without a superscript. It should always be clear which w you're referring to.

c. The notation in Equation 7 may make for easier reading, but it makes it very difficult to figure out what's actually going on, especially since there's no easy way to tell vectors from scalars. We would strongly recommend using components (dw_i_/dt = ……). It would make it much more clear, with very little cost.

24. Along the same line, the phrase "unweighted synaptic potential (USP) train" is pretty distracting. We would strongly suggest dropping it, since everybody agrees what a spike train is. Plus, we searched, and "weighted dendritic potentials" was used only once, in the sentence before unweighted synaptic potentials were defined. So "unweighted" seems a bit redundant.

25. Important equations should, in our opinion, be displayed. That's because readers (OK, some of these readers) always go back and search for important quantities, and they're easier to see if displayed. (This was triggered by the expression for V, a few lines up from Equation 2.)

26. The setup should be made clear up front: the neuron is receiving a teacher spike train, y_t_, and using those spike trains to update its synapses. The statement before Equation 3, "Assuming that the neuron strives to reproduce a target firing distribution p*(y|x)" was not helpful (I couldn't really make sense of it, so I kind of ignored it). And so it took me forever to figure out what was going on.

27. Along the same lines, wouldn't it make sense to just say the neuron is trying to maximize the log probability of the teacher spikes? It gives the same cost function, and it's a lot less obscure than saying the cost function is the KL distance. Clearly a matter of taste, but log probability seems more sensible.

28. On the top of page 9, it says that learning rates are inversely correlated to the variance of σ^2^(x^ε^). It needs to be clear why the equations say that; right now it's pretty much out of the blue.

29. End of discussion (page 12): "error-correcting plasticity rule". Why is it error correcting?

30. Very end of discussion:

"Explicitly and exactly applying natural gradient at the network level does not appear biologically feasible due to the existence of cross-unit terms in the Fisher information matrix G. However, methods such as the unit-wise natural-gradient approach (Ollivier, 2015) could be employed to approximate the natural gradient using a block-diagonal form of G. For spiking networks, this would reduce global natural-gradient descent to our local rule for single neurons."

We didn't understand that -- it should be unpacked.

31. Given that this is *eLife*, and there's no page limit, we would strongly urge you to get rid of the Appendix and put all the analysis in Methods. We suspect anybody willing to read Methods will want to see the algebra in the Appendix. In any case, all relevant quantities (e.g., g_1_, ……) should be in Methods; the reader shouldn't have to hunt them down in what will potentially be another document. And certainly, the gradient, which is central to the whole endeavor, should be in Methods. As should Equation S18, so that the reader knows where G(w) comes from.

32. There's a much easier derivation of G(w), Equation 31:

int dx exp(h dot t) N(x; mu, Σ) f(a dot x)

is easy to evaluate (here x is a vector and N(x; mu, Σ) is a Gaussian with mean mu and covariance Σ). Take two derivatives, and, voila, you have.. Same answer, but easier on the reader.

In addition, the derivation of its inverse can be more concise. G(w) is a sum of a diagonal matrix and a rank-2 modulation. Denoting

V = (r w') and C = [[c1 e^2^, c2 e], [c2 e, c3]],

we may write

G(w) = c1 Σ + V C V^T^.

Applying the Woodbury identity,

G^-1^(w) = Σ^-1^/c1 – V' (C^-1^ + V Σ^-1^ V^T^)^-1^ V'^T^,

where

V' = (r'/e w).

Thus,

g = [[g1 g3], [g2 g4]]

is given as

g = – (C^-1^ + V Σ^-1^ V^T^) ^-1^.

33. Aren't Equations S12-13 a repeat of arguments made above, on the same page? Or is this something new? Either way, it should be clear (and if it's a repeat, maybe it can be dropped).

[Editors' note: further revisions were suggested prior to acceptance, as described below.]

Thank you for resubmitting your work entitled "Natural-gradient learning for spiking neurons" for further consideration by *eLife*. Your revised article has been evaluated by John Huguenard (Senior Editor) and a Reviewing Editor.

The manuscript has been improved but there are some remaining issues that need to be addressed, as outlined below:

The good news: The reviewers are quite positive, and there are no major concerns.

The bad news: Reviewer 1 has many comments requiring clarification. But almost all should be relatively straightforward to address in a timely way.

*Reviewer #1:*

The manuscript is much improved, although it's still a bit hard to read. Some of which we think could be easily fixed. Specific comments follow.

1. p 1: "It certainly could be the case that evolution has favored one particular parametrization over all others during its gradual tuning of synaptic plasticity, but this would necessarily imply sub-optimal convergence for all but a narrow set of neuron morphologies and connectome configurations."

We found this confusing, since parametrization and learning rules are not directly coupled. Did you mean "evolution has favored one particular parametrization over all others, and updates weights using Euclidean gradient descent ……"?

2. One more attempt to kill USP: why not call x^ε^ filtered spike trains, since that's what they are? "Unweighted synaptic potential" is completely unfamiliar, and with a paper this complicated, that's the last thing one needs -- every single time we read that phrase, we had to mentally translate it to filtered spike trains.

3. Technically, x_iε_ should depend on distance to the soma: because of dendritic filtering, the farther a psp travels along a dendrite the more it spreads out (in time). You should mention this, and speculate on whether it will make much difference to your analysis. It seems like it might, since c_ε_ scales with the timescale of the psp.

4. In both places where you mention spikes (x_i_, above Equation 4 and Y^*^, above Equation 9) you should point out that they are a sum of δ functions. "Spikes" isn't so well defined in this field.

5. If you want anybody besides hard core theorists do understand what you're doing, you'll need to be explicit about what p* is: it's Equation 7 with phi replaced by phi*, the target firing rate as a function of voltage. We strongly suggest you do that. In particular, the description "a target firing distribution p*(y∣x^ε^)" doesn't help much, since a firing rate distribution is usually a distribution over firing rates. What you mean, though, is that p* gives you the true probability of a spike in a time interval.

5. It would be very helpful, in Equation 9, to add

\approx dt[ (phi – phi*) – phi* log(phi/phi*) ]

(and point out that it's exact in the limit dt  0). Then Equation 10 is easy -- just say that phi* is replaced by Y*.

6. Starting in Equation 10, you switch to a weight without a superscript. We found this very confusing, since the learning rule you wrote down in Equation 10 is exactly the one for w^s^. We strongly suggest you never use w. But if there is a reason to do so, you should be crystal clear about what it is. As far as we could tell, this switch to w is completely unsignalled. So maybe it's a typo?

7. p 4: "which is likely to harm the convergence speed towards an optimal weight configuration."

The word "likely" is not all that convincing. Can you be a lot stronger? Are there relevant refs? Or you could point to the simulations in Figure 3.

8. p 4: "In the following, we therefore drop the index from the synaptic weights w to emphasize the parametrization-invariant nature of the natural gradient."

In fact, what's really going on is that you're deriving the natural gradient learning rule (Equation 13) with V given by

V = sum_i_ f(w_i_) x_iε_

which is actually very confusing, since f was used previously to relate w^s^ to w^d^ (Equation 4). Is the f in Equation 14 the same as the one in Equation 4? If so, it should be w^d^ on the right hand side of Equation 14. If not, you should use a different symbol for the function.

As you can see, your notation is confusing us. Our suggestion would be to always use superscripts s or d, and never use w by itself.

9. Figure 2:

"(A) During supervised learning, the error between the current and the target state is measured in terms of a cost function defined on the neuron's output space; in our case, this is the manifold formed by the neuronal output distributions p(y, x)."

What does "defined on the neuron's output space" mean? Doesn't the cost function depend only on the weights, since it's an average over the input-output function? Which is more or less what you say in the next sentence,

"As the output of a neuron is determined by the strength of incoming synapses, the cost C is an implicit function of the afferent weight vector w.!

Although we're not sure why you say "implicit" function; isn't it a very explicit function (see point 5 above).

"If, instead, we follow the gradient on the output manifold itself, it becomes independent of the underlying parametrization."

Since we don't know what the output manifold is (presumably the statistical manifold in panel A? not that that helps), this sentence doesn't make sense to us.

"In contrast, natural-gradient learning will locally correct for distortions arising from non-optimal parametrizations."

This is a highly nontrivial statement, and there's nothing in the manuscript so far that makes this obvious. I think a reference is needed here. Or point the reader to Figure 3 for an example?

10. Equation 13: You may not like components, but it would be extremely helpful to put them in Equation 13, or maybe add an equation with components to make it clear. There is a precedent; you use components in Equation 4. And let's face it, anybody who isn't mathematically competent will have been lost long ago, and anybody who is mathematically competent will want to be sure what's going on. And things are actually ambiguous, since it's really hard to tell what's a vector and what's a scalar. In particular, γ_u_ should be treated as a vector, even though it's not bold, so it's not immediately clear whether γ_s_ should also be treated as a vector. If nothing else, you should use γ_u_ {\bf 1}; that would help a little. Also, the convention these days when writing f(w) is for f not to be bold, but instead define it as a pointwise nonlinearity. You don't have to follow conventions, but it does make it easier on the reader.

The same comment applies to Equation 17.

11. In Equation 14, presumably w^s^ on the left hand side should be bold? And why not use indices, as in Equation 4? Much more clear, and fewer symbols.

12. p 6, typo: "inactive inactive".

13. Two questions and one comment about Figure 3F. First, why don't the orange and blue lines start in the same place? Second, the blue line doesn't appear to be saturating. Can the Euclidean gradient do better than the natural gradient? If so, that's important to point out. But maybe it's not true, and is a by-product of your optimization method?

And the comment: presumably the D_KL_ term on the y-axis has a factor of dt. That makes it pretty much impossible to interpret, since we don't know what dt is (probably it's somewhere, but it's not in an obvious place). We suggest removing the factor of dt, and making it clear that when phi is close to phi*, what's being plotted is <(phi-phi*)^2/2 phi*>. That will make the plot much easier to interpret.

14. p 8: "As a result, the effect of synaptic plasticity on the neuron's output is independent of the synapse location, since dendritic attenuation is precisely counterbalanced by weight update amplification."

Because of the term γ_w_ w^d^, this is true only if w^d^ \propto 1/α(d). But, as you point out later, this doesn't have to be the case. So this statement needs to be modified. Maybe it's approximately true?

15. Figure 5, several comments:

In the equation in panel H, r should be in the numerator, not the denominator, and a factor of ε_02_ is missing. In addition, the term on the right hand side can be simplified:

σ^2^(x^ε^) = r ε_02_/2(tau_1_ + tau_2_).

Also it would be a good idea to switch to tau_s_ and tau_m_ here rather than tau_1_ and tau_2_, to be consistent with Methods, and to not clash with the tau's in panels A-B.

And in panels A-B, presumably it should be tau_si_ rather than tau_i_?. Also, tau_m_ should be reported here.

It would make it a lot easier on the reader if you wrote down an equation for cε, which isn't so complicated: c_ε_/r_i_ = 2 (tau_m_ + tau_s_)/ ε_02_ r_i_. Equation 19 is a natural place to do that.

Because of the other two terms in Equation 17, it's not true that "Natural-gradient learning scales inversely with input variance." It should be clear that this is approximate.

16. In Equation 19, it should be x_iε_ on the left hand side, not x^ε^.

17. p 10: "Furthermore, it is also consistent with data from Aitchison and Latham (2014) and Aitchison et al.et al. (2021), as well as with their observation of an inverse dependence on presynaptic firing rates, although our interpretation is different from theirs."

There is no justification for this statement: in Aitchison et al.et al. a plot of Δ w/w versus r showed 1/sqrt(r) scaling. That would be consistent with the theory in this paper only if the weight scales as 1/sqrt(r). It might, but without checking, you can't make that claim.

That data is weak, so the fact that you don't fit it is hardly the end of the world. However, what's more important is to point out that the theory here makes very different predictions that the theory in Aitchison et al.et al. That way, experimentalists will have something to do.

Finally, the inverse scaling with firing rate must eventually saturate, since there's a limit to how big weight changes can be. You should comment on this, if briefly.

18. Figure 7, we're pretty lost, for several reasons:

a. Because homosynaptic scales as 1/r, it's not possible to have unstimulated synapses (for which r=0); at least not with the current derivation. If you want to have unstimulated synapses, you need to show that it is indeed possible, by computing G when r_i=0_ for some of the i's.

b. From the bottom of page 27, γ_u_ \approx ε_0_ c_ε_. Thus, the first two plasticity terms are

c_ε_ (x^ε^/r – ε_0_).

Because = r ε_0_, these two terms approximately cancel. Which means everything should be driven by the last term, c_w_ V f(w). This doesn't seem consistent with the explanation in Figure 7.

c. Because of the term c_w_ V f(w), shouldn't there be a strong dependence on the unstimulated weights in panels B-D?

All this should be clarified.

19. p 13: "We further note an interesting property of our learning rule, which it inherits directly from the Fisher information metric that underlies natural gradient descent, namely invariance under sufficient statistics (Cencov, 1972)." What does "invariance under sufficient statistics" refer to?

20. p 13: "A further prediction that follows from our plasticity rule is the normalization of weight changes by the presynaptic variance. We would thus anticipate that increasing the jitter in presynaptic spike trains should reduce LTP in standard plasticity induction protocols."

Presumably this statement comes from the fact that the learning rate scales inversely with the variance of the filtered spike train. However, it's not clear that jittering spike trains increases the variance. What would definitely increase the variance, though, is temporal correlations. Maybe this could be swapped in for jittering? Either that, or explain why jittering increases variance. And also, it would be nice to refer the reader to the dependence of the learning rate on variance.

21. You should comment on where you think the teacher spike train comes from. Presumably it's delivered by PSPs at the soma, but those don't propagate back to synapses. How do you envision the teacher spike trains communicating with the synapses? Even if the answer is "we don't know", it's important to inform the reader -- this may simply be an avenue for future research.

22. In Equations 29 and 30, you should explain why you use \approx. Presumably because that's because you assumed a constant firing rate?

23. Equation 31: it would be extremely useful to point out that c_ε_ = 2(tau_m_ + tau_s_)/ε_02_, along with a reference (or calculate it yourself somewhere).

24. After Equation 31, "Unless indicated otherwise, simulations were performed with a membrane time constant tau_m_ = 10 ms and a synaptic time constant τs = 3 ms. Hence, USPs had an amplitude of 60 mV".

Doesn't the amplitude of the USPs depend on ε_0_? And 60 mV seems pretty big. Is that a typo?

25. Equation 34: Missing parentheses around Σ_USP_ w^s^ in the second term in parentheses.

26. Equation 24 and the line above it: should w_i_ be w_id_? If not, you shouldn't use f, since that was used to translate somatic to dendritic amplitude.

27. Equation 115: Should it be Theta(V-theta)?

28. Equation 117 is identical to Equation 67. Did you mean to drop the term (phi')^2^/phi?

Also, (phi')^2^/phi = 1 when V > theta. But presumably it's equal to 0 when V < theta. If so, Equation 117 should be

G(w) = E(dt Theta(V-theta) x xT).

if that's correct, then the integrals I_1_-I_3_ do not reduce to the values given in Equations 118-120.

*Reviewer #2:*

The authors have done a good job in the revision.

A number of small suggestions remaining:

– Equation 13. Wouldn't it be easier to write this for a single component wi)˙=……

– Figure 3 shows temporal structure in the teacher signal but this is nowhere explained in the main text.

– Fig3D+E perhaps the axis or caption can indicate whether w1,2 is 10 or 50Hz.

– Figure 4A: Shouldn't the purple top solid curve (dendritic voltage) be taller than the solid orange curve?

– Fig5D+E+F might look better with the position of the y-axis left instead of right.

---

## [Author Response]

Essential revisions:This paper derives a natural gradient learning rule for a spiking neuron in a supervised setting. Unlike conventional gradients in Euclidean space, the natural gradient is invariant under reparameterization, thus achieving fast convergence regardless of the position of synaptic contacts on the dendritic tree. The authors relate their rule to experimentally observed properties of synaptic plasticity, such as heterosynaptic regularization. We're not aware of any other work that applied natural gradient to a spiking neuron in a formal way, and we believe it is an important contribution.That said, we do have several comments many have to do with presentation, and, we hope, easy to address. Because this is a combination of three reviews, they're not exactly in order of appearance in the manuscript. But hopefully not too far off.1. We're slightly concerned with the fact that the weight update scales inversely with the firing rate. First, presumably for small enough firing rates something in the derivation breaks down. Second, this makes a very strong prediction. The only data we know of that relates to this is in Aitchison and Latham, 2014 (now published in Nature Neuroscience vol 24, pgs 565-571, 2021), where they showed, in one experiment, that learning rate scales as 1/sqrt{r} for moderate firing rates (greater than about 1 Hz), and saturates at small firing rates. This is one experiment, so doesn't rule out the theory here, but these points should be discussed.

The inverse scaling comes from the first term in the Fisher information matrix, which is proportional to the input’s covariance matrix. This fits nicely into the philosophy of natural gradient descent and the resulting synaptic democracy, as isotropization of the cost with respect to the weights also implies a renormalization of the inputs with respect to their variance. For the particular case of Poisson firing discussed in our manuscript, the variance of an input is proportional to its rate, hence the inverse dependence of our learning rule w.r.t. the input rate. The rule is well-behaved as long as the input rates are not exactly zero, which is arguably the case in a non-pathological brain. Of course, depending on the exact experimental scenario and stimulation protocol, the other heterosynaptic components could override this scaling effect and even invert it. We also referenced Aitchison et al. as our prediction is, indeed, similar to theirs, and also compatible with their data, but made clear that our interpretation is different (lines 270-272 in the revised manuscript with tracked changes).

2. The link to biology is often very cursory. References to experiments are often in the form 'so and so found something a bit similar'. Wherever possible, you should try to make precision comparisons to experiments. And on page 10 you say that the weight dependent heterosynaptic terms explains the central weight distributions found in cortex. However, the weight dependent heterosynaptic term is just one term out of three, and the other terms are not weight-dependent. If you want to argue this, you should be more rigorous (and in particular take into account all three terms).

We tried to clarify the relationship to experimental data, while remaining cautious in order to not overstate our claims, since most experiments have not been performed in the exact same settings as those in which we tested our model. For example, many experiments from the referenced literature on heterosynaptic plasticity rely on deterministic stimulation protocols rather than on Poisson input (e.g., Royer and Paré, 2003, White et al. 1990). However, the qualitative matches of our results to this data represent encouraging evidence, while also providing an inspiration for future, more targeted experiments that also allow a quantitative comparison. To emphasize this point, we included an additional paragraph in the discussion, outlining potential directions of future experiments.

Furthermore, as highlighted by Häusser, data on the dependence of plasticity on synaptic location is rare, “as most such studies to date have implicitly assumed that synaptic function at distal and proximal synapses is identical.” (Häusser, 2001, p. R12). This puts the findings of Froemke, Poo, and Dan, 2005 in relation to our predictions on democratic plasticity, as we discuss at the end of Section 2.4 (lines 245-249).

As to the last point, we argue that the existence of the het_w_ term corresponds to the observation made by Loewenstein et al. (2011) about changes in spine size being proportional to the spine size itself (in light of Asrican et al. (2007), who show that synaptic strength is positively correlated with spine size). However, we agree that the remark about heterosynaptic plasticity “explaining” the central weight distribution in cortex might have been phrased too strongly – “correspondence” is, in this case, certainly a better descriptor than “explanation”.

3. The conclusion of the Zenke and Gerstner paper is, we believe, incorrect: A learning rule with an intrinsic weight dependence, such as Oja's rule, does not need homeostasis. This clearly doesn't affect any of your analysis, but it does mean that the statement "… (Zenke and Gerstner, 2017), where it was shown that pairing Hebbian terms with heterosynaptic and homeostatic plasticity is crucial for stability" is, we believe, not true.

You are absolutely right about the role of the weight-dependent term in Oja’s rule, but we don’t think that this necessarily contradicts Zenke and Gernstner. It might only be a matter of wording, but we would argue that in Oja’s rule, the negative, weight-dependent term is, in addition to being heterosynaptic, also homeostatic, because it stabilizes (normalizes) the weight vector by punishing too large weights.

4. Figure 1B is pretty incomprehensible, and it doesn't help that the equations are small and gray, making them hard to read. We would suggest dropping it, especially since the point is made very well in the text.

Agreed, we moved Figure 1B to the corresponding Figure 8 (Section 6.3 of the revised manuscript) in the Methods.

Along the same lines (but much later), in Sec. ‘The naive Euclidean gradient is not parametrization-invariant’ we don't see any contradiction between Equations 26 and Equation 27; they are just two different definitions of \Δ w^s^. It might be an inconsistency, but not a contradiction.

Agreed, done (line 621).

5. Figure 3: Considering that several approximations were made in the derivation of the learning rule (even for Equation 7), learning performance should be evaluated a bit more carefully. In particular:a. Please plot the distance between student weights and teacher weights, on the same timescale as in Figure 3F (either in Euclidean space or in Fisher metric space).

Good point, we added another Figure (#9) in the methods. We also plotted the distance between student and teacher firing rates, as well as the “minimum” and “maximum” learning curves.

b. Please plot performance versus learning rate, so we can get a sense of robustness. "Performance" could either be asymptotic loss or time to achieve a particular loss. Also, can you speculate how the learning rate of a neuron can be optimized?

Our simulations show (panel D in the new Figure 9) that the convergence behavior is robust against perturbations of the learning rate.

Meta-learning is, in general, a complex subject of active research, and we are aware of several related experimental (e.g., https://elifesciences.org/articles/51439) and algorithmic (e.g., https://elifesciences.org/articles/66273) studies. For our purposes, the optimization of eta_0_ is a relatively simple case of univariate optimization, so our simple grid search was sufficient.

c. What's the target output firing rate? It might have been stated somewhere, but we missed it. It would be nice if it were in the figure caption, like it is in Figure 4.

The teacher was modeled as a Poisson neuron that fires with a time-dependent rate, given by \phi (\sum_i=1_^n^ w*_i_ \xepsi), with a randomly sampled target weight w^*^. We added a sentence in the caption of Figure 3 to clarify this.

d. In Figure 3, n=2 and n=100 are both used. It should be clear which panel is which.

Absolutely, we adapted the caption accordingly.

e. Panels B and C: what's the timescale? There's only one labeled time point (0.1), so it's impossible to tell. And how many trials did it take to learn? We would have expected considerable learning during 2500 trials.

We added a second labeled time point so that the time scale in the panel becomes clear. Furthermore, we renamed the ordinate of the scatter plots into PSTH, since the label “trial” might have been misleading. The scatter plots in B and C show repeated sampling of the teacher and student firing distribution before and after learning, for one of the sample traces in 3F. This means they do not depict the learning process itself, but illustrate the firing distribution of the student and teacher at fixed points in the learning process.

f. The direction of the gradient depends on whether or not there's a spike on the teacher neuron. Which means the path in weight space is noisy, and the weights should not converge -- instead, they should exhibit fluctuations forever. But the trajectories in D and F were very smoothed. Is that because they're averaged over a large number of trials? Or was some extra processing done? Please explain.

You are correct, the weight paths were averaged over 500 trials. We included this information in the figure caption so it is easier to find.

6. Figure 4:a. Can you reproduce experimental results on dendritic position dependent plasticity (eg. Letzkus JJ, Kampa BM, Stuart GJ, 2006; Sjöström PJ, Häusser M, 2006). These experimental results are inconsistent with vanilla Euclidean gradient; hence they would provide support for biological relevance of the proposed learning rule.

Thank you for raising this interesting point. It is certainly true that this is conceptually related to the issue we discuss here – in some sense, it’s a mirror problem, namely the attenuation of output signals versus the attenuation of input signals. We have added a paragraph in the discussion where we describe the relationship between these findings and the predictions of our learning rule (lines 366-376).

b. "the distance from soma was varied between 1 microns to 10 microns". This seems small; the characteristic length of dendritic decay is in order of 100 microns (Williams and Stuart, 2002).

Thank you for pointing this out, this is a mistake in the caption of Figure 4 that appeared during the final editing of the paper. It is not the distance from soma that is varied between 1 and 10 microns, but rather the inverse attenuation factor α(d) which is varied between 1 and 10. Since α(d)=exp(-d/λ), this means that d/λ is varied between 0 and ln(10) ≅ 2,3. With an electrotonic length scale of approximately 200 μm, as described in Williams and Stuart (2002), this corresponds to a distance d from soma between 0 and 460 μm. We corrected this in the revised manuscript (caption of Figure 4).

7. Figure 5: Here, you compared three neurons each receiving single excitatory input with different input characteristics. Do you expect the same result to be true for a single neuron receiving three excitatory inputs? We're curious about it because that's a more relevant scenario. Some casual remarks would be helpful.

When the activity of inputs does not overlap in time (at any point in time, only one is active), the results will hold exactly as in Figure 5. We chose this scenario because it is the only one where a clear-cut comparison can be made, without additional complications induced by their interaction.

In the case of simultaneously active inputs, we still expect the variance to strongly impact the learning rate in an inverse-proportional manner, since Equation 34 of the revised manuscript (line 585) shows that the input variance is an important contributor to the Fisher information matrix, whose inverse plays an essential role in the natural gradient update (see Equation 12 of the revised manuscript, line 134). From Equation 37 (revised manuscript, line 591), it becomes clear that the homosynaptic component of our learning rule exactly scales with the inverse variance of the unweighted synaptic potential. We also ran computer simulations for this scenario, but the resulting data is much more difficult to study, since due to the interactions of the different inputs, the behavior of the heterosynaptic components becomes much more complicated.

8. Figure 7:a. We're somewhat confused that you you can distinguish between the three forms of plasticity in Equations 11-13, at least for the stimulated synapses. Don't all three forms contribute at once? But in panel B you say, "Weight change of stimulated weights (homosynaptic)". And in panel C, you should have seen homosynaptic plasticity (see next comment). Could you explain how you can single out one form of plasticity? Or else drop the comment. Or maybe we misunderstood?

You are right, stimulated synapses are affected by all three terms simultaneously, while unstimulated synapses are only affected by the heterosynaptic terms. The confusion resulted from our overloading of the terms “homosynaptic” and “heterosynaptic”. In literature, they are often used synonymously with “stimulated” and “unstimulated”. Here, we use the terms to describe the different components of the learning rule. We have revised the caption and text to avoid the resulting ambiguity.

b. In the legend of Figure 7, it is mentioned that the rest of synapses received 0.01Hz input, but in Sec. ‘Comparison of homo- and heterosynaptic plasticity’, that information is omitted. Did you actually stimulate these synapses? If so, we would have expected a big jump in synaptic weight when there was a spike, and in 60 s there should have been 3 spikes (0.01x60x5).

You are right, and we addressed this misunderstanding in the revised caption of Figure 7 and Sec. ‘Comparison of homo- and heterosynaptic plasticity’ (lines 677-679). In the simulation, the synapses received no input spikes at all. The assumption of infinitesimally small input rates was simply an argument that allowed us to avoid division by zero in the homosynaptic term. Biologically, this would simply correspond to zero homosynaptic plasticity for zero input.

c. 7G: A should be S?

Yes, you are right. We corrected it in the figure.

d. Equation 13 is out of the blue. Could you tell us where this comes from?

Indeed, this was not sufficiently clear in the previous version. Furthermore, Equation 13 ( = Equation 23 of the revised manuscript, line 285) contained a typo as the last identity only holds approximately. We corrected it and added a reference to the section in the Methods (former SI) where the empirical analysis of γ_w_ is explained.

And from the bottom of page 10, we have"… is roughly a linear function of the membrane potential (more specifically, its deviation with respect to its baseline)."What does baseline refer to? We didn't see anything about baseline in Equation 25, which is where Equation 13 comes from.

By “baseline” we refer to the somatic membrane potential V_rest_ of the neuron in the absence of afferent input. Throughout the paper, we assumed a value of -70 mV for V_rest_ (see Sec. ‘Neuron model’). In contrast, the variable V captures the part of the somatic membrane potential which is evoked by afferent input (Equation 5, line 84). Hence, the total somatic membrane potential is given as V_rest_ + V. The definitions of both V_rest_ and V can also be found in the first paragraph of the Methods section (Sec. ‘Neuron model’ of the revised manuscript, lines 542 and 547 / Equation 25). We now explicitly define both V_rest_ and V at the very beginning of Section 2.2 (line 79).

e. Could you tell us how the colors in panels B-D were computed? Presumably some assumptions were made. If so, what were they?

Yes, we made the following assumptions, which we have now also included in the Methods, Sec. ‘Comparison of homo- and heterosynaptic plasticity’ (lines 683-689).

For the plots in 7B and 7C, we chose the diverging colormap matplotlib.cm.seismic in matplotlib. The colormap was inverted such that red indicates negative values representing synaptic depression and blue indicates positive values representing synaptic potentiation. The colormap was linearly discretized into 500 steps. To avoid too bright regions where colors cannot be clearly distinguished, the indices 243-257 were excluded. The colorbar range was manually set to a symmetric range that includes the max/min values of the data.

In Figure 7D, we only distinguish between regions where plasticity at stimulated and at unstimulated synapses have opposite signs (light green), and regions where they have the same sign (dark green).

9. Equation 4: we believe that the right hand side should not have a minus sign. More importantly, we should be told that Y* consists of a sum of δ-functions; the statement "Here, Y* denotes a teacher spike train sampled from p*" was unhelpful. It's also important to tell us exactly how Y* is sampled, which is, presumably fromP(spike from Y* \in [t, t+dt]) = phi(sum_j_ w*_j_ x_jε_) dt.If that's not correct, then we're thoroughly confused.

You are right about the minus sign, we corrected this typo (line 101 / Equation 10 of the revised manuscript).

We also added a reference to the Methods where we provide more information regarding the teacher spike train. For brevity and readability, and since the main text is already quite heavy on equations, we opted for having these equations in the Methods.

You are also right that for the simulations in Figure 3 we choose a realizable teacher, i.e., one that we know can be learned; as you say, its spikes are sampled from

P(spike from Y* \in [t, t+dt]) = phi(sum_j_ w*_j_ x_jε_) dt.

We made this assumption to facilitate the analysis of our learning rule, but please note that Equation 13 (line 137) of the revised manuscript can be derived in a more general setting where the teacher spikes are generated from an arbitrary (inhomogeneous) Poisson process.

10. The second paragraph on page 4 makes some strong statements, and we're not sure all of them are true. For instance,"Convergence of learning is therefore harmed by the slow adaptation of distal synapses compared to equally important proximal counterparts."Presumably, "adaptation" means "learning". If so, why is adaptation necessarily slow at distal synapses? That would seem to depend on the biological learning rule. If not, what does "adaptation" mean in this context?

Yes, “adaptation” means “learning” in this context. And you are right, the statement was not precise enough. We were referring specifically to a dendritic parametrization, which might appear natural for local computation. We have corrected the text accordingly (lines 109-113).

And,"With the multiplicative USP term x^ε^ in Equation 4 being the only manifestation of presynaptic activity, there is no mechanism by which to take into account input variability, which can, in turn, also impede learning."This is a bit out of the blue. Why should input variability affect learning? So far the authors have not said anything about this. It comes up later, but right now it's pretty obscure.

You are right, we should have at least provided some intuition for the reader until they reach the corresponding section in the manuscript. We have amended the text accordingly (lines 114-115).

11. Last paragraph on page 4: the motivation for using the Fisher information matrix is a bit obscure. (For instance, what does "it is generally the unique metric that remains invariant under sufficient statistics" mean?) We strongly suspect that we won't be only ones who are lost. So it would be good to motivate it in plain language.

Indeed, invariance under sufficient statistics is not a commonplace property and needs explanation. To not disrupt the flow in this already dense section, we now simply refer to some classical literature on information geometry to motivate the canonical nature of the Fisher metric (line 129). We then specifically explain invariance under sufficient statistics and address its relevance for neuronal computation in the Discussion section (lines 358-364).

And in particular, it would be nice to show, at the very least, that using the Fisher information matrix automatically makes the learning rule invariant with respect to a change of variables. (Which is pretty easy to do.)

We agree that at this point in the manuscript it would be useful to state this invariance explicitly. To not disrupt the flow of reading, we added a corresponding sentence in the main manuscript (line 135), along with a reference to the detailed derivation and figure in the Methods.

Along the same lines, after Equation 9 you say"This represents an unmediated reflection of the philosophy of natural gradient descent, which finds the steepest path for a small change in output, rather than in the numeric value of some parameter."This has been the theme all along, but was it ever shown? As far as we can tell, you just used the inverse of the Fisher information matrix. Is it obvious that it has the above effect?

We used this argument when introducing the Fisher metric in the first place (line 125 onwards) and tried to visualize it in Figure 2:

“The key idea of natural gradient as outlined by Amari is to follow the (locally) shortest path in terms of the neuron's firing distribution. Argued from a normative point of view, this is the only "correct" path to consider, since plasticity aims to adapt a neuron's behavior, i.e., its input-output relationship, rather than some internal parameter (Figure 2).”

Building on your other suggestions, we made a number of modifications to Section 2.2. to better explain different aspects of the theory. We hope that now readers are provided with sufficient intuition before reaching the “unmediated reflection” remark in Section 2.5.

12. It would be extremely useful to write down log p_w_ in the main text.

Done, but without the log, for brevity (line 90 / Equation 8 of the revised manuscript).

13. The quantities in Equation 7 should be defined immediately -- right now we have to read down half a page to figure out what they are. And you should tell us immediately how γ_u_ and γ_w_ could be computed locally.

Agreed. We referenced the factors briefly directly after the equations (lines 139-140) and added expressions for γ_u_ and γ_w_ to the main manuscript (line 174 / Equation 15 of the revised manuscript).

14. In the section "Natural gradient speeds up learning", please tell us exactly what the cost function is (which, presumably, means telling us w* and the filters you use to convert incoming spike trains to x). And it would be good if the firing rates were given in the main text, so one doesn't have to look at the figure.

Done. We have included more information in the text and also added references to the corresponding sections in the methods (lines 195-204).

15. Because of the factor of phi'(V)/phi(V) in Equations 7 and 8, the effective scaling in Equation 9 is, approximately, 1/phi'(V). This doesn't change the point that a shallow slope implies a high learning rate, but thing's aren't quite as bad as Equation 9 implies. This seems worth mentioning.

Good idea, done (lines 256-257).

16. You mention, on the top of page 8, "that the absence of dendritic democracy does not contradict the presence of democratic plasticity". It seems worth mentioning, just for completeness, that dendritic democracy can be achieved without democratic plasticity. In particular, as far as we can tell, the Euclidean gradient is also consistent with dendritic democracy, although convergence to the democratic state might be slower that for the natural gradient.

Done (lines 250-251).

17. p. 10 "In comparison, input from weak synapses only has…" We couldn't get the logic of this and the following sentences. This non-trivial claim is not explicitly tested in the subsequent paragraph, but reappears in the discussion. You should expand on it, or remove it.

We understand how our previous formulation may have caused some confusion here, so we modified the text in the hope of avoiding this misunderstanding (lines 296-304). In short: Assuming a single spike arrives via a strong synapse, it will cause a larger PSP (and therefore a larger V compared to the baseline) than the same spike arriving via a weak synapse. Thus, on average, weak synapses require more stimulus than strong synapses for the same weight change to occur.

18. p22: "The integral formulas follow from…". It would help if you referred to Equations 20 and 21 (in Sec. ‘Neuron model’) here. We were confused by the sudden appearance of ε_o_ and c_ε_. Also, you should mention how they are related to dt.

Done (lines 762-763).

19. It is somewhat confusing that you set up the problem about dendritic distance, but then first address, in Figure 3, the role of firing rate, before showing the solution to dendritic plasticity. Perhaps starting with Figure 4, contrasting standard gradients to natural gradients would help.

Indeed, we asked ourselves the same question when writing the manuscript. In the end, we decided to present our results in this order for several reasons. First, Figure 3 is intended to be the classical “it works” figure, before delving into the details and implications. Second, it does portray some fundamental properties of the natural gradient, namely the isotropization of the cost function close to the optimum. Third, it addresses the purely functional perspective (not only does it work, but it also can surpass the standard Euclidean approach) early on, which we believe to be important. Fourth, if it came after Figure 4, we feel it would disrupt the flow by switching from physiology to function then to physiology again. So while we certainly agree that swapping the figures would carry some benefit, we believe that it would also come with certain downsides and would therefore prefer the current order.

20. Page 7/8, you say "neurons are not symmetrical geometric objects". You should make it clear what this means (presumably it means that they have dendrites, and, therefore, some weights are attenuated). In addition, it's not clear that not being a symmetrical geometric objects implies a non-isotropic cost landscapes, since how isotropic the cost landscape is depends on the input and cost function as well as the attenuation in the dendrites. In principle they former could cancel the latter, producing an isotropic cost landscapes even with strongly attenuating neurons. It seems best to drop this point.

You are absolutely right that the different factors that affect the shape of a cost function could, in principle, cancel out. We thought that at this point it would be instructive to convey the intuition that not only are cost landscapes often anisotropic, but that, given the diversity of morphologies and input signals, it would be an incredible coincidence if cost landscapes were isotropic in the first place. We modified the text (lines 188-192) to make it more clear, following your suggestions.

21. A number of studies have recently emphasized that in large networks most machine learning problems have highly degenerate minima (e.g. Belkin et al. PNAS). Does the algorithm generalize to such situations? Your answer may be "we don't know". But whatever the answer, it would be worth mentioning in the paper.

The generalization of our learning rule to larger networks is certainly a research direction that we are very interested to explore. Variants of natural-gradient learning have already been successfully applied to rate-based deep networks, and their convergence behavior has often been found to compare favorably to methods based on Euclidean gradient descent (see e.g. Bernacchia et al. 2018, Ollivier 2015, Pascanu and Bengio 2013).

We are therefore optimistic that also for the case of Poisson-spiking neurons, natural-gradient learning will generalize well to the large network case. In particular, since our learning rule relies on local information, we do not expect it to be affected by the presence of degenerate minima in a different way than standard backpropagation.

A detailed analysis of this scenario is however beyond the scope of the current paper, since there are still many open questions on how to best train stochastically spiking deep networks with gradient methods, even for Euclidean gradient descent.

22. Page 7, last paragraph: should the reference to Figure 5 be to Figure 4?

You are right, we corrected this typo (line 231).

23. The abundance of superscripts, subscripts and decorations in the variables throughout the manuscript make it very hard to read. We would suggest simplifying notation as much as possible. In particular:a. The superscript on the gradient operator doesn't help, as it requires extra memorization -- which is hard because it's not standard. You would be much better off putting the superscript on the cost function, and not redefining the gradient. It's especially confusing in Equation 6, since at first glance the superscript n means take n derivatives.

We very much agree with the aim of simplifying notation, and we tried our best to do so. Moreover, the point about the nth derivative is absolutely correct. We tried to address it by capitalizing the superscript and making it non-italic. However, we think that the superscript must remain on the gradient, because it would otherwise suggest that it is the cost function that changes. While it is true that it is useful to view the natural gradient as “isotropizing the cost function”, strictly speaking, the cost function does not change. For the exact same cost function under the exact same parametrization, the two gradients predict different trajectories in weight space.

b. There are two w's: w^d^ and w^s^. Except sometimes there's a w without a superscript. It should always be clear which w you're referring to.

Agreed, and we have amended the manuscript accordingly. We use all three of these notations to emphasize the difference between somatic, dendritic and parametrization-invariant learning. In Equation 10 (revised manuscript, line 101), the somatic amplitude is learned, and the text below now makes it explicit. As we move to the natural gradient, the learning becomes parametrization-invariant, and the superscript-free version of the weights is used to make this point. We added a sentence above Equation 11 in which we say this explicitly (lines 123-124).

c. The notation in Equation 7 may make for easier reading, but it makes it very difficult to figure out what's actually going on, especially since there's no easy way to tell vectors from scalars. We would strongly recommend using components (dw_i_/dt = …). It would make it much more clear, with very little cost.

Throughout the paper, we chose to use vectors for easier reading and have consistently used bold notation for vectors. In particular, it greatly reduces the number of indices, which is already rather high, due to the complexity of the problem, as you have pointed out above. We fear that a switch of notation around one particular equation might be confusing to the reader. Alternatively, using component-wise notation everywhere would make the equations significantly more difficult to parse. We would therefore kindly ask you to consider keeping the current notation.

24. Along the same line, the phrase "unweighted synaptic potential (USP) train" is pretty distracting. We would strongly suggest dropping it, since everybody agrees what a spike train is. Plus, we searched, and "weighted dendritic potentials" was used only once, in the sentence before unweighted synaptic potentials were defined. So "unweighted" seems a bit redundant.

We understand the possible confusion and hope that writing out the voltage equation (see changes in main text, Equation 5 of the revised manuscript, line 84) helps clarify this point. The spike trains are denoted as x_i_, whereas the filtered spike trains (with the PSP kernel \ε) are denoted by x_iε_. These are what we call USPs, and they are quite important, as they appear explicitly in the learning rule (and its derivation).

25. Important equations should, in our opinion, be displayed. That's because readers (OK, some of these readers) always go back and search for important quantities, and they're easier to see if displayed. (This was triggered by the expression for V, a few lines up from Equation 2.)

We absolutely agree and have displayed more equations inline in the revised manuscript, both in the main text and in the supplement. Being aware that different readers have different preferences regarding equations, we tried to reduce the number of displayed equations to a minimum, but your comments encouraged us to increase the value of this minimum.

26. The setup should be made clear up front: the neuron is receiving a teacher spike train, y_t_, and using those spike trains to update its synapses. The statement before Equation 3, "Assuming that the neuron strives to reproduce a target firing distribution p*(y|x)" was not helpful (I couldn't really make sense of it, so I kind of ignored it). And so it took me forever to figure out what was going on.

Agreed. We modified the text accordingly (lines 93-97).

27. Along the same lines, wouldn't it make sense to just say the neuron is trying to maximize the log probability of the teacher spikes? It gives the same cost function, and it's a lot less obscure than saying the cost function is the KL distance. Clearly a matter of taste, but log probability seems more sensible.

We agree, and added a sentence about the equivalence of minimizing the KL and maximizing the log-likelihood of teacher spikes (lines 99-100).

28. On the top of page 9, it says that learning rates are inversely correlated to the variance of σ^2^(x^ε^). It needs to be clear why the equations say that; right now it's pretty much out of the blue.

Indeed, this statement referred more to the empirical observation provided in Figure 5. The more precise mathematical reasoning is given in the next sentence (line 262): “In particular, for the homosynaptic component, the scaling is exactly [inversely proportional].”

29. End of discussion (page 12): "error-correcting plasticity rule". Why is it error correcting?

This is due to the multiplicative error term [Y* – phi(V)] in the learning rule. We address this term explicitly in Section 2.2, as well as in Section 2.6 and Figure 7, where plasticity switches sign at the zero-error line.

30. Very end of discussion:"Explicitly and exactly applying natural gradient at the network level does not appear biologically feasible due to the existence of cross-unit terms in the Fisher information matrix G. However, methods such as the unit-wise natural-gradient approach (Ollivier, 2015) could be employed to approximate the natural gradient using a block-diagonal form of G. For spiking networks, this would reduce global natural-gradient descent to our local rule for single neurons."We didn't understand that -- it should be unpacked.

We have revised and extended this paragraph to hopefully improve its clarity (lines 407-414). In particular, it now explicitly addresses the non-localities induced by applying natural-gradient descent at the network level and explains the benefits of the unit-wise solution more clearly.

31. Given that this is eLife, and there's no page limit, we would strongly urge you to get rid of the Appendix and put all the analysis in Methods. We suspect anybody willing to read Methods will want to see the algebra in the Appendix. In any case, all relevant quantities (e.g., g_1_, …) should be in Methods; the reader shouldn't have to hunt them down in what will potentially be another document. And certainly, the gradient, which is central to the whole endeavor, should be in Methods. As should Equation S18, so that the reader knows where G(w) comes from.

Agreed, done, and thank you for the suggestion! We struggled with this decision ourselves and tried to make the manuscript more reader-friendly by shifting some of the more detailed content into an SI, but are definitely happy with the new structure.

32. There's a much easier derivation of G(w), Equation 31:int dx exp(h dot t) N(x; mu, Σ) f(a dot x)is easy to evaluate (here x is a vector and N(x; mu, Σ) is a Gaussian with mean mu and covariance Σ). Take two derivatives, and, voila, you have. Same answer, but easier on the reader.

We really appreciate the detailed feedback, especially given how complicated some of our derivations are and we are certainly welcoming suggestions for simplification. In this case, however, we must apologize but the derivation of Equation 31 (now Equation 48) already seems quite straightforward to us – which, admittedly, might very much be a matter of taste.

In addition, the derivation of its inverse can be more concise. G(w) is a sum of a diagonal matrix and a rank-2 modulation. DenotingV = (r w') and C = [[c1 e^2^, c2 e], [c2 e, c3]],we may writeG(w) = c1 \Σ + V C V^T^.Applying the Woodbury identity,G{^-1^}(w) = \Σ{^-1^}/c1 – V' (C{^-1^} + V \Σ{^-1^} V^T^) {^-1^} V'^T^,whereV' = (r'/e w).Thus,g = [[g1 g3], [g2 g4]]is given asg = – (C{^-1^} + V \Σ{^-1^} VT) {^-1^}.

Thank you for this suggestion! Indeed, the idea of working directly with a rank-2 correction is more elegant and concise than our iterative application of Sherman-Morrison, and certainly more appealing to a reader interested in understanding the principle behind our method. We therefore added this idea to our derivation in the Methods (line 779-780). However, since the result remains the same, we have conserved our original derivation in order to avoid an extensive and error-prone change in notation.

33. Aren't Equations S12-13 a repeat of arguments made above, on the same page? Or is this something new? Either way, it should be clear (and if it's a repeat, maybe it can be dropped).

Assuming you are referring to either Equation S3 or Equation S4 (now Equations 51 and 52, lines 733-735), both of them are similar to the arguments made in Eq. S12 and Eq. S13 -(now Equations 61 and 62, line 742), but not exactly the same.

1. In Equation S3 (now Equation 51), the density over the USPs vanishes, since it appears in both the counter and the denominator of the fraction.

2. In Equation S4 (now Equation 52), the density of the target firing distribution vanishes when taking the derivative, since it does not depend on w.

3. In contrast, in EquationS12 and S13 (now Equations 61 and 62), the density over the USPs vanishes when taking the derivative, since it also does not depend on w.

We have adapted the text to make this difference more clear.

[Editors' note: further revisions were suggested prior to acceptance, as described below.]

Reviewer #1:The manuscript is much improved, although it's still a bit hard to read. Some of which we think could be easily fixed. Specific comments follow.1. p 1: "It certainly could be the case that evolution has favored one particular parametrization over all others during its gradual tuning of synaptic plasticity, but this would necessarily imply sub-optimal convergence for all but a narrow set of neuron morphologies and connectome configurations."We found this confusing, since parametrization and learning rules are not directly coupled. Did you mean "evolution has favored one particular parametrization over all others, and updates weights using Euclidean gradient descent ……"?

We adapted the sentence to address this issue.

2. One more attempt to kill USP: why not call x^ε^ filtered spike trains, since that's what they are? "Unweighted synaptic potential" is completely unfamiliar, and with a paper this complicated, that's the last thing one needs -- every single time we read that phrase, we had to mentally translate it to filtered spike trains.

We believe that USP is a more exact description for x^ε^, because it implicitly specifies the filter (same filter as the membrane, difference of exponentials) and the lack of synaptic weighting (see our reply to #24) whereas “filtered spike trains” does not (it could just be the synaptic current, for example). The first paragraph of Sec. ‘The natural-gradient plasticity rule’ and especially Equation 5 should leave no room for interpretation.

3. Technically, x_iε_ should depend on distance to the soma: because of dendritic filtering, the farther a psp travels along a dendrite the more it spreads out (in time). You should mention this, and speculate on whether it will make much difference to your analysis. It seems like it might, since c_ε_ scales with the timescale of the psp.

It is true that our model does not capture all the details of biological neurons, but our observations regarding the effects of weight reparametrization hold in general. We clarified this in a footnote above Equation 1.

4. In both places where you mention spikes (x_i_, above Equation 4 and Y^*^, above Equation 9) you should point out that they are a sum of δ functions. "Spikes" isn't so well defined in this field.

We agree, which is why we stated this explicitly in Equation 27.

5. If you want anybody besides hard core theorists do understand what you're doing, you'll need to be explicit about what p* is: it's Equation 7 with phi replaced by phi*, the target firing rate as a function of voltage. We strongly suggest you do that. In particular, the description "a target firing distribution p*(y∣x^ε^)" doesn't help much, since a firing rate distribution is usually a distribution over firing rates. What you mean, though, is that p* gives you the true probability of a spike in a time interval.

We agree, which is why we stated this explicitly in Sec. ‘Sketch for the derivation of the somatic natural-gradient learning rule‘, Equations 32 and 33.

5. It would be very helpful, in Equation 9, to add\approx dt[ (phi – – phi*) – – phi* log(phi/phi*) ](and point out that it's exact in the limit dt  0). Then Equation 10 is easy -- just say that phi* is replaced by Y*.

We agree, but have decided – in line with suggestions from the reviewers – to keep the main manuscript as light as possible on mathematical details/derivations and rather focus on the key arguments and insights. (In this particular case, since Equation 10 is not our result and appears often in other literature, we point to Pfister et al.et al. (2006) for more details.)

6. Starting in Equation 10, you switch to a weight without a superscript. We found this very confusing, since the learning rule you wrote down in Equation 10 is exactly the one for w^s^. We strongly suggest you never use w. But if there is a reason to do so, you should be crystal clear about what it is. As far as we could tell, this switch to w is completely unsignalled. So maybe it's a typo?

We agree, which is why we have stated this explicitly in the text above Equation 11: “In the following, we therefore drop the index from the synaptic weights w to emphasize the parametrization-invariant nature of the natural gradient.” In particular, this means that Equation 13 holds in a more general setting than only for the somatic parametrization. If we use w_s_ instead of w, we lose this generality.

7. p 4: "which is likely to harm the convergence speed towards an optimal weight configuration."The word "likely" is not all that convincing. Can you be a lot stronger? Are there relevant refs? Or you could point to the simulations in Figure 3.

We provide an explanation under Equation 3, which is explicitly referenced in the sentence you mention: “However, from a functional perspective, the opposite should be true: dendritic weights should experience a larger change than somatic weights in order to elicit the same effect on the cost.” We used “likely” simply because we want to be cautious. It is possible to imagine (arguably pathological) cases where the somatic learning rate is so large that it prevents convergence, so a dendritic parametrization can help by decreasing the learning rate.

8. p 4: "In the following, we therefore drop the index from the synaptic weights w to emphasize the parametrization-invariant nature of the natural gradient."In fact, what's really going on is that you're deriving the natural gradient learning rule (Equation 13) with V given byV = sum_i_ f(w_i_) x_iε_which is actually very confusing, since f was used previously to relate w^s^ to w^d^ (Equation 4). Is the f in Eq 14 the same as the one in Equation 4? If so, it should be w^d^ on the right hand side of Equation 14. If not, you should use a different symbol for the function.

The f here is a generalization of the f in Equation 4. It describes the relationship of an arbitrary weight parametrization to the somatic amplitude parametrization. Since Equation 4 is actually a special case for the dendritic amplitudes, we feel that using f in both cases is justified. We added a sentence below Equation 14 that states this explicitly.

As you can see, your notation is confusing us. Our suggestion would be to always use superscripts s or d, and never use w by itself.9. Figure 2:"(A) During supervised learning, the error between the current and the target state is measured in terms of a cost function defined on the neuron's output space; in our case, this is the manifold formed by the neuronal output distributions p(y, x)."What does "defined on the neuron's output space" mean? Doesn't the cost function depend only on the weights, since it's an average over the input-output function? Which is more or less what you say in the next sentence,"As the output of a neuron is determined by the strength of incoming synapses, the cost C is an implicit function of the afferent weight vector w.!Although we're not sure why you say "implicit" function; isn't it a very explicit function (see point 5 above).

The cost function does indeed depend on the weights, but only indirectly, via the output. If two particular weight configurations give the same output, the cost function will be the same. This is illustrated in Figure 2A. We agree that “implicit” was not the right word here and replaced it with “indirectly”. C is defined on the space of p, but the p themselves depend on w, so C depends indirectly on w.

"If, instead, we follow the gradient on the output manifold itself, it becomes independent of the underlying parametrization."Since we don't know what the output manifold is (presumably the statistical manifold in panel A? not that that helps), this sentence doesn't make sense to us.

In the paragraphs directly above Equation 11 we discuss the output manifold in detail. In particular, we say explicitly that “the set of our neuron’s realizable output distributions[,] forms a Riemannian manifold” and, in the first sentence under Figure 2, that “[the] cost function [is] defined on the neuron's output space; in our case, this is the manifold formed by the neuronal output distributions p(y,x)”. These distributions are, in turn, defined in Equation 7 and 8.

"In contrast, natural-gradient learning will locally correct for distortions arising from non-optimal parametrizations."This is a highly nontrivial statement, and there's nothing in the manuscript so far that makes this obvious. I think a reference is needed here. Or point the reader to Figure 3 for an example?

Since natural gradient multiplies the Euclidean gradient by the inverse Fisher information matrix G^-1^ (and G is a Riemannian metric), provides the relationship between distances on the parameter space and the output manifold, we believe this statement is justified. We added a reference to Figure 3 as suggested.

10. Equation 13: You may not like components, but it would be extremely helpful to put them in Equation 13, or maybe add an equation with components to make it clear. There is a precedent; you use components in Equation 4. And let's face it, anybody who isn't mathematically competent will have been lost long ago, and anybody who is mathematically competent will want to be sure what's going on. And things are actually ambiguous, since it's really hard to tell what's a vector and what's a scalar. In particular, γ_u_ should be treated as a vector, even though it's not bold, so it's not immediately clear whether γ_s_ should also be treated as a vector. If nothing else, you should use γ_u_ {\bf 1}; that would help a little. Also, the convention these days when writing f(w) is for f not to be bold, but instead define it as a pointwise nonlinearity. You don't have to follow conventions, but it does make it easier on the reader.The same comment applies to Equation 17.

Thank you for pointing this out – yes, we should have multiplied γ with the unit vector explicitly, as we did in other parts of the document. The vector notation was a conscious choice on our part, with one important reason being the avoidance of stacked indices. Given this decision, we used bold notation for vector-valued functions such as **f** to help the reader differentiate them from scalar-valued ones such as C.

11. In Equation 14, presumably w^s^ on the left hand side should be bold? And why not use indices, as in Equation 4? Much more clear, and fewer symbols.

Good catch, thank you!

12. p 6, typo: "inactive inactive".

Good catch, thank you!

13. Two questions and one comment about Figure 3F. First, why don't the orange and blue lines start in the same place? Second, the blue line doesn't appear to be saturating. Can the Euclidean gradient do better than the natural gradient? If so, that's important to point out. But maybe it's not true, and is a by-product of your optimization method?

We describe this in the simulation details (section 6.4.1), but we added a few clarifying words: “For the learning curves in Figure 3F, initial and target weight components were chosen randomly (but identically for the two curves) from a uniform distribution on U(-1/n,1/n), corresponding to maximal PSP amplitudes between -600 µV and 600 µV for the simulation with n=100 input neurons. Learning curves were averaged over 1000 initial and target weight configurations.” This implies that the orange and blue line do indeed start at the same point, but it is just a bit difficult to see since they partly overlap.

Furthermore, the Euclidean gradient descent does converge, which is a bit more obvious in Figure 9A. However, we wanted to highlight the difference between natural-gradient and Euclidean-gradient descent and therefore chose to focus on the most relevant part of the learning curves. Please note also Figure 12C-F, which shows the convergence behavior displayed in Figure 3F is consistent over multiple input scenarios.

Since we did not provide a formal proof that natural-gradient-descent learning outperforms Euclidean-gradient-descent learning, there could be special scenarios where Euclidean-gradient-descent learning exhibits, on average, a more favorable convergence behavior than natural gradient. However, we did not observe such a case in our simulations, which is consistent with the findings from other studies of natural-gradient descent.

And the comment: presumably the D_KL_ term on the y-axis has a factor of dt. That makes it pretty much impossible to interpret, since we don't know what dt is (probably it's somewhere, but it's not in an obvious place). We suggest removing the factor of dt, and making it clear that when phi is close to phi*, what's being plotted is <(phi-phi*)^2/2 phi*>. That will make the plot much easier to interpret.

We see how this relates to you point #5, but we think that Figure 9C (distance of firing rates) should provide additional context to interpret Figure 3F, should the need arise.

14. p 8: "As a result, the effect of synaptic plasticity on the neuron's output is independent of the synapse location, since dendritic attenuation is precisely counterbalanced by weight update amplification."Because of the term γ_w_ w_d_, this is true only if w^d^ \propto 1/α(d). But, as you point out later, this doesn't have to be the case. So this statement needs to be modified. Maybe it's approximately true?

This might be a misunderstanding. Equation 17 is a special case of Equation 13 for exactly the case where Equation 16 holds. Please note the general form of the last term in Equation 13 and refer to Figure 8B for a detailed explanation on how attenuation is canceled out for natural-gradient-descent learning.

15. Figure 5, several comments:In the equation in panel H, r should be in the numerator, not the denominator, and a factor of ε_02_ is missing. In addition, the term on the right hand side can be simplified:σ^2^(x^ε^) = r ε_02_/2(tau_1_ + tau_2_).Also it would be a good idea to switch to tau_s_ and tau_m_ here rather than tau_1_ and tau_2_, to be consistent with Methods, and to not clash with the tau's in panels A-B.And in panels A-B, presumably it should be tau{_si_} rather than tau_i_?. Also, tau_m_ should be reported here.

Good ideas, thank you, we have incorporated all of them into Figure 5.

It would make it a lot easier on the reader if you wrote down an equation for cε, which isn't so complicated: c_ε_/r_i_ = 2 (tau_m_ + tau_s_)/ε_02_ r_i_. Equation 19 is a natural place to do that.

Agreed, we added this expression to Equation 31 (see also # 23).

Because of the other two terms in Equation 17, it's not true that "Natural-gradient learning scales inversely with input variance." It should be clear that this is approximate.

You are certainly right, which is why we said “inversely” rather than “inversely proportional”. We added an “approximately” to make this unequivocal.

16. In Equation 19, it should be x_iε_ on the left hand side, not x^ε^.

Good catch, thank you!

17. p 10: "Furthermore, it is also consistent with data from Aitchison and Latham (2014) and Aitchison et al.et al. (2021), as well as with their observation of an inverse dependence on presynaptic firing rates, although our interpretation is different from theirs."There is no justification for this statement: in Aitchison et al.et al. a plot of Δ w/w versus r showed 1/sqrt(r) scaling. That would be consistent with the theory in this paper only if the weight scales as 1/sqrt(r). It might, but without checking, you can't make that claim.That data is weak, so the fact that you don't fit it is hardly the end of the world. However, what's more important is to point out that the theory here makes very different predictions that the theory in Aitchison et al.et al. That way, experimentalists will have something to do.Finally, the inverse scaling with firing rate must eventually saturate, since there's a limit to how big weight changes can be. You should comment on this, if briefly.

We suspect that this is simply a matter of wording, as we also pointed out above, under #15. We used “inverse” in an approximate, qualitative sense, and not as “inversely proportional”. It is meant to be taken as “if the presynaptic rate becomes larger, the weight changes become smaller”. We believe the statement to be correct and in line with Aitchison et al.et al., who write “the learning rate […] increases as the synapse’s uncertainty […] increases” and “the synapse’s uncertainty should fall as the presynaptic firing rate increases”. We adapted the corresponding sentence in our manuscript to mitigate the potential misunderstanding that our model makes the same quantitative predictions.

18. Figure 7, we're pretty lost, for several reasons:a. Because homosynaptic scales as 1/r, it's not possible to have unstimulated synapses (for which r=0); at least not with the current derivation. If you want to have unstimulated synapses, you need to show that it is indeed possible, by computing G when r_i_=0 for some of the i's.

We already addressed this in our response to point 8b of the previous review: “You are right, and we addressed this misunderstanding in the revised caption of Figure 7 and Sec. ‘Comparison of homo- and heterosynaptic plasticity’ (lines 677-679). In the simulation, the synapses received no input spikes at all. The assumption of infinitesimally small input rates [in the simulated learning rule] was simply an argument that allowed us to avoid division by zero in the homosynaptic term. Biologically, this would simply correspond to zero homosynaptic plasticity for zero input.” Please note that line numbers and highlighted text changes are now, necessarily, different; the referenced sentence is the second one in the first paragraph of Sec. ‘Comparison of homo- and heterosynaptic plasticity’.

b. From the bottom of page 27, γ_u_ \approx ε_0_ c_ε_. Thus, the first two plasticity terms arec_ε_ (x^ε^/r – ε_0_).Because = r ε_0_, these two terms approximately cancel. Which means everything should be driven by the last term, c_w_ V f(w). This doesn't seem consistent with the explanation in Figure 7.

If we understand correctly, you are looking at the average weight change <\Δ w>. This is an average over a product between the postsynaptic error and the parenthesis containing the sum over the three terms (cf. Equation 13). The average of this product is not the product of the two averages, because the error is correlated with the input, and plasticity is driven by this very correlation (and of course the last term c_w_ V f(w)).

c. Because of the term c_w_ V f(w), shouldn't there be a strong dependence on the unstimulated weights in panels B-D?

The purpose of these panels is to address this exact question. Figure 7B shows the change in the stimulated weights, so there is no dependence on the unstimulated weights (since the influence of the term c_w_ V f(w) is component-wise). In contrast, for the unstimulated weights (Figure 7C), we see a clear effect of this term which particularly manifests itself in the switch from potentiation to depression in the upper right corner of the panel. This dependence on the unstimulated weight is also clearly visible in the upper right corner of Figure 7D.

All this should be clarified.19. p 13: "We further note an interesting property of our learning rule, which it inherits directly from the Fisher information metric that underlies natural gradient descent, namely invariance under sufficient statistics (Cencov, 1972)." What does "invariance under sufficient statistics" refer to?

A parameter of a probability distribution can often be described by a statistic (such as the sample mean) which is sufficient if it contains all necessary information about the parameter. The distributions of a sufficient statistic form themselves a parametric model and therefore a Riemannian manifold, with a one-to-one relationship between the original parametric model and the one for the sufficient statistic. Due to the invariance property of the Fisher metric, the distances of corresponding points will be equal up to a constant factor. We felt the need to mention this distinguishing property of the Fisher metric, since it makes it stand out from other Riemannian metrics on statistical manifolds and because it is relevant for biological neurons, as explained in the paragraph. Since we believe that the discussion is not a good place to elaborate on mathematical details – again, to not distract the reader from the main message – we refer to the original literature, in this case Cencov, 1972.

20. p 13: "A further prediction that follows from our plasticity rule is the normalization of weight changes by the presynaptic variance. We would thus anticipate that increasing the jitter in presynaptic spike trains should reduce LTP in standard plasticity induction protocols."Presumably this statement comes from the fact that the learning rate scales inversely with the variance of the filtered spike train. However, it's not clear that jittering spike trains increases the variance. What would definitely increase the variance, though, is temporal correlations. Maybe this could be swapped in for jittering? Eiththeyer that, or explain why jittering increases variance. And also it would be nice to refer the reader to the dependence of the learning rate on variance.

We adapted the sentence as requested.

21. You should comment on where you think the teacher spike train comes from. Presumably it's delivered by PSPs at the soma, but those don't propagate back to synapses. How do you envision the teacher spike trains communicating with the synapses? Even if the answer is "we don't know", it's important to inform the reader - this may simply be an avenue for future research.

We did, directly below Equation 10, where the teacher spike train first appears in the postsynaptic error term: “On the single-neuron level, a possible biological implementation has been suggested by Urbanczik and Senn (2014), who demonstrated how a neuron may exploit its morphology to store errors, an idea that was recently extended to multilayer networks (Sacramento et al.et al., 2017; Haider et al.et al., 2021a).”

22. In Equations 29 and 30, you should explain why you use \approx. Presumably because that's because you assumed a constant firing rate?

Exactly, “\approx” becomes “=” for constant input rates. We added an explanatory sentence below Equation 31.

23. Equation 31: it would be extremely useful to point out that c_ε_ = 2(tau_m_ + tau_s_)/ε_0_^2, along with a reference (or calculate it yourself somewhere).

Done.

24. After Equation 31, "Unless indicated otherwise, simulations were performed with a membrane time constant tau_m_ = 10 ms and a synaptic time constant τs = 3 ms. Hence, USPs had an amplitude of 60 mV".Doesn't the amplitude of the USPs depend on ε_0_? And 60 mV seems pretty big. Is that a typo?

Yes, it does, and we explicitly say that ε_0_ = 1 mV ms just a few lines above. The size of the USP does not matter – the U stands for unweighted, hence our insistence on keeping the acronym to make this explicit (see our reply to #2). Just one sentence further down from the one you cite, we discuss the actual size of PSPs. (But also in other places, for example Sec. ‘Distance dependence of amplitude changes’.)

25. Equation 34: Missing parentheses around Σ_USP_ w^s^ in the second term in parentheses.

Good catch, thank you!

26. Equation 24 and the line above it: should w_i_ be w_id_? If not, you shouldn't use f, since that was used to translate somatic to dendritic amplitude.

We are not sure what you mean, since Equation 24 does not contain a w_i_. Could this be a typo? Maybe our reply to # 8 regarding the use of f is helpful in this context.

27. Equation 115: Should it be Theta(V-theta)?

Good catch, thank you!

28. Equation 117 is identical to Equation 67. Did you mean to drop the term (phi')_2_/phi?

Yes, thank you!

Also, (phi')^2^/phi = 1 when V > theta. But presumably it's equal to 0 when V < theta. If so, Equation 117 should beG(w) = E(dt Theta(V-theta) x x^T^).if that's correct, then the integrals I_1_-I_3_ do not reduce to the values given in Equations 118-120.

You are right and thank you for pointing this out. We think that everything should work out fine if we assume that the mass of the membrane potential distribution is above theta. While this is not necessarily biologically realistic, we think that this result is still instructive due to its simplicity. We adapted the text for more clarity.

Reviewer #2:The authors have done a good job in the revision.A number of small suggestions remaining:– Equation 13. Wouldn't it be easier to write this for a single component wi)˙=……

We agree that sometimes this can be easier, but in this case, we decided to use vector notation to avoid index clutter, see also our reply to #10 of Reviewer 1.

– Figure 3 shows temporal structure in the teacher signal but this is nowhere explained in the main text.

We added a few clarifying sentences to the caption of Figure 3: “During learning, the firing patterns of the student neuron align to those of the teacher neuron. The structure in these patterns comes from autocorrelations in this instantaneous rate. These, in turn, are due to mechanisms such as the membrane filter (as seen in the voltage traces) and

the nonlinear activation function.”

– Fig3D+E perhaps the axis or caption can indicate whether w1,2 is 10 or 50Hz.

Good idea, thank you! We added a corresponding sentence to the caption.

– Figure 4A: Shouldn't the purple top solid curve (dendritic voltage) be taller than the solid orange curve?

Yes, absolutely, we adapted the sketch to make this clearer.

– Fig5D+E+F might look better with the position of the y-axis left instead of right.

We chose the axes on the left so the histograms look more familiar and just rotated by 90° CCW.